# Stress Testing Byzantine Robustness in Distributed Learning

## Abstract

Byzantine robustness in distributed learning consists in ensuring that distributed optimization algorithms, such as distributed SGD, are robust to arbitrarily malicious participants, also called Byzantine workers. Essentially, such workers *attack* the algorithm to prevent it from delivering a good model, by sharing erroneous information. Several defenses have been proposed so far, typically with theoretical worst-case robustness analyses. Yet, these analyses only show convergence to critical points up to large constants, which provides a false sense of security in the absence of a strong attack benchmark. We contribute to addressing this shortcoming by modeling an optimal Byzantine adversary in distributed learning, from which we derive Jump, a long-term attack strategy aiming at circumventing the training loss's minima. Interestingly, even if Jump is a solution to a simplified form of the optimal adversary's problem, it is very powerful: even the greedy version of Jump can satisfactorily break existing defenses. We systematically evaluate state-of-the-art attacks and defenses on MNIST and CIFAR-10 under data heterogeneity, and show that Jump consistently performs better or comparably to other attacks. For example, on CIFAR-10, Jump *doubles* the accuracy damage from $66\%$ accuracy, across existing attacks, to $50\%$ on average, compared to $81\%$ without attack under moderate data heterogeneity. Hence, we encourage the usage of Jump as a stress test of Byzantine robustness in distributed learning.

## 1 Introduction

Modern machine learning systems owe much of their success to the emergence of a new generation of models, which are trained using an ever-increasing amount of data and computing resources. Training at such a large scale has become more accessible due to the advent of distributed machine learning algorithms (Bertsekas & Tsitsiklis, 2015; McMahan et al., 2017; Kairouz et al., 2021). These algorithms involve several machines (or workers) collaboratively train a common model. This not only reduces the computational burden on any single worker but also allows the workers to retain control over their local data, which is quite desirable when dealing with sensitive personal data, e.g., in healthcare (Sheller et al., 2020). In a typical distributed learning algorithm, such as distributed SGD (D-SGD), the workers seek an optimal model using stochastic gradients computed on their respective local data. At each iteration, the workers send their local gradients to a central server, which updates the current model using the average of the gradients.

However, in practical scenarios, distributing the learning procedure among several workers causes safety and reliability issues. Some workers may crash, i.e., suddenly stop responding, while others may communicate incorrect information, e.g., due to local data poisoning. In the worst case, when an adversary takes control of a worker, this information may even be maliciously crafted in order to disrupt the learning procedure. To cover all such possibilities, it is common to assume that a fraction of the workers are *Byzantine* and can arbitrarily deviate from their prescribed algorithm.[1] Standard distributed algorithms, however, are not reliable in such an adversarial setting. Indeed, if left unprotected, the learning procedure can be critically influenced by a handful of Byzantine workers, as demonstrated in the past by several *attacks* executed by Byzantine workers (Baruch et al., 2019; Xie et al., 2020; Shejwalkar & Houmansadr, 2021). This vulnerability could have severe societal repercussions in public-domain applications, such as healthcare or banking.

---

[1]The term Byzantine is borrowed from the distributed computing literature, where the issue of misbehaving machines in distributed systems was first studied (Lamport et al., 1982).

A plethora of solutions has been proposed to protect distributed learning against Byzantine workers, e.g., see (Blanchard et al., 2017; Guerraoui et al., 2018; Yin et al., 2018; Allen-Zhu et al., 2021; Karimireddy et al., 2022; Allouah et al., 2023). These methods replace the simple average operation at the server in D-SGD by a robust aggregation rule. The resulting algorithm provably outputs a critical point of the training loss function, despite the presence of some Byzantine workers. However, the theoretical guarantees of these methods do not preclude an adversary from forcing the algorithm to a critical point of their choice (Guerraoui et al., 2018), which could yield poor generalization. Furthermore, Byzantine-robust convergence rates feature large multiplicative constants, which means that in practice an adversary can significantly slow down training, despite theoretical robustness guarantees. It is thus necessary to exhaustively test the proposed defenses on practical learning tasks. Several *attacks*, i.e., simulations for Byzantine workers, have been developed to break the defenses (Baruch et al., 2019; Xie et al., 2020; Shejwalkar & Houmansadr, 2021). However, these attacks often target specific defense schemes. Consequently, testing a new defense scheme against these attacks could provide a *false sense of security*. Clearly, existing attacks often fail to simulate the worst-case adversarial behavior. We aim to address this shortcoming by proposing an adaptive Byzantine attack called JUMP. Our key contributions can be summarized as follows.

## 1.1 KEY CONTRIBUTIONS

We consider the general class of defenses based on adapting D-SGD with a robust aggregation rule (Guerraoui et al., 2023). We design our attack to stress test the robustness of defenses within the standard Byzantine threat model, where an adversary has open-box access to the learning scheme and can corrupt a subset of workers. Such workers may share arbitrary vectors, in place of their true local gradients, in each training step. In this context, the adversary's objective is to maximize the total training loss of the algorithm. We observe, however, that this problem belongs to a class of non-convex optimization problems that are hard to solve in general. To overcome this challenge, we propose JUMP, a novel Byzantine attack that solves a simplified version of the problem by restricting the adversary's search space in the following manner. First, all the Byzantine vectors are constrained to be identical, as a natural collusion strategy, and be colinear with the average of the honest (non-Byzantine) local gradients. Second, the problem is decomposed into smaller subproblems, by partitioning the iterative learning procedure into several segments of fewer iterations.

We demonstrate, through experiments on low-dimensional non-convex learning tasks, that while a Byzantine-robust adaptation of D-SGD can reach a global minimum under state-of-the-art attacks, JUMP consistently circumvents it to worse regions of the loss landscape, where it leverages data heterogeneity to converge to poor models. Surprisingly, this phenomenon holds regardless of the segment length that JUMP uses for decomposing the aforementioned optimization problem of the optimal adversary. This means that even a greedy version of JUMP, which tries to maximize the training loss at each training step independently, yields an effective attack that can significantly diminish the accuracy of Byzantine-robust schemes. To verify this, we conduct a series of experiments on heterogeneous versions of CIFAR-10 and MNIST classification tasks, with state-of-the-art Byzantine-robust schemes that have been proven to be optimal (Karimireddy et al., 2022; Allouah et al., 2023). Our findings unequivocally suggest that the existing attack benchmarks grossly overestimate the Byzantine robustness property of the defenses. In particular, while state-of-the-art defenses yield good accuracy under existing attacks, they fail under JUMP in most scenarios.

## 1.2 RELATED WORK

Many Byzantine-robustness schemes use known aggregation methods from robust statistics, such as *geometric median* (Chen et al., 2017; Pillutla et al., 2022), *minimum diameter averaging* (Rousseeuw, 1985), *coordinate-wise trimmed mean* and *median* (Yin et al., 2018). Researchers have also proposed original schemes specifically designed for distributed learning, e.g., *Krum* (Blanchard et al., 2017), outlier detection (Alistarh et al., 2018; Allen-Zhu et al., 2021), and *comparative gradient elimination* (Gupta et al., 2021). However, several works on attacks have demonstrated the vulnerability of these defenses by exploiting either the stochasticity (Baruch et al., 2019; Xie et al., 2020; Karimireddy et al., 2021) or the heterogeneity (Karimireddy et al., 2022) aspects of distributed learning. To circumvent these shortcomings, recent works improved upon these defenses by adding pre-aggregation schemes (Karimireddy et al., 2022; Allouah et al., 2023) and variance reduction operations (Karimireddy et al., 2021; Farhadkhani et al., 2022; Gorbunov et al., 2023). An overview on Byzantine-robust distributed learning methods can be found in Guerraoui et al. (2023).

**Closely related work.** The landscape of Byzantine attacks in distributed learning encompasses a wide spectrum of strategies, with varying degrees of attacker knowledge and adaptability. These attacks can be categorized into two types: *non-adaptive* and *adaptive* attacks (Shejwalkar & Houmansadr, 2021). On the one hand, non-adaptive attacks are executed by adversaries with limited knowledge, which do not adapt their strategies to the behavior of other workers or the server. These attacks include *a little is enough* (Baruch et al., 2019), *fall of empires* (Xie et al., 2020), *label-flipping*, *sign-flipping* (Allen-Zhu et al., 2021), and *mimic* (Karimireddy et al., 2022). Naturally, because these techniques do not use the full power of an open-box adversary, they fall short of the worst-case Byzantine model and can be mitigated by advanced defenses (Allouah et al., 2023).

On the other hand, adaptive attacks are executed by adversaries who are able to adapt their strategies according to the overall behavior of the learning procedure. Our methodology belongs to this line of research. A recent study (Shejwalkar & Houmansadr, 2021) provides a framework for crafting aggregation-tailored attacks against existing defenses, while it also proposes two principled aggregation-agnostic adaptive attacks, *MinMax* and *MinSum*. However, we show that these attacks often perform poorly compared to JUMP mainly because they do not maximize the loss directly, and cannot circumvent local minima. Lastly, the attack presented in (Li et al., 2022) follows a reinforcement learning approach, where corrupted workers attempt to learn the global data distribution as well as the attack strategy. However, this attack is only effective at later stages of the training, hence could be mitigated by early stopping. In contrast, JUMP is designed to damage the performance during the entire training, and is significantly cheaper computationally than reinforcement learning.

## 2 PROBLEM STATEMENT

We consider the classical server-based architecture with $n$ workers $w_1, \ldots, w_n$, and a trusted central server. The workers only communicate with the server, without any inter-worker communication. For a given learning task, we denote by $\mathcal{D}_i$ the data distribution of $w_i$. For a given model parameter $\theta \in \mathbb{R}^d$, data point $z \sim \mathcal{D}_i$ has real-valued loss $\ell(\theta; z)$. The local loss at worker $w_i$ is defined as:

$$\mathcal{L}_i(\theta) := \mathbb{E}_{z \sim \mathcal{D}_i}[\ell(\theta; z)], \ \forall \theta \in \mathbb{R}^d. \tag{1}$$

**Goal of distributed learning.** We consider the scenario where $f$ workers of unknown identity are *Byzantine*. These workers do not follow the prescribed algorithm, and may send arbitrary information to the server. Following the Byzantine threat model (Lamport et al., 1982), we assume that there exists an external *omniscient* adversary controlling these workers. This means that the adversary has full knowledge of the protocol and the workers' identities, and can coordinate the actions of the Byzantine workers. In this context, the server aims to compute, by collaborating with the workers, a parameter $\theta^*$ that minimizes the global *honest* (non-Byzantine) loss function $\mathcal{L}_{\mathcal{H}}$ defined as:

$$\mathcal{L}_{\mathcal{H}}(\theta) := \frac{1}{|\mathcal{H}|} \sum_{i \in \mathcal{H}} \mathcal{L}_i(\theta), \ \forall \theta \in \mathbb{R}^d, \tag{2}$$

where $\mathcal{H}$ is the set of honest workers, i.e., those following the prescribed instructions correctly.

**Robust distributed SGD.** When all workers are honest, the above optimization problem can be solved using the distributed variant of SGD (D-SGD) (Bertsekas & Tsitsiklis, 2015). D-SGD is an iterative method where, in each step $t \geq 1$, the server maintains a parameter vector $\theta_t$ broadcasted to all workers. Each worker $w_i$ then returns a stochastic estimate $g_t^{(i)}$ of its local gradient $\nabla \mathcal{L}_i(\theta_t)$. The server then updates $\theta_t$ by using the average of the received gradients, i.e., $\theta_{t+1} = \theta_t - \gamma_t \frac{1}{n} \sum_{i=1}^n g_t^{(i)}$. In the presence of Byzantine workers, however, the server cannot simply average gradients (Blanchard et al., 2017). Instead, a standard approach is to augment D-SGD with *robust aggregation* in place of averaging, so the update of D-SGD is modified to:

$$\theta_{t+1} = \theta_t - \gamma_t F\left(g_t^{(1)}, \ldots, g_t^{(n)}\right), \tag{3}$$

where $F$ denotes a robust aggregation rule seeking to filter out the malicious vectors sent by Byzantine workers. Prominent robust aggregations include *Krum* (Blanchard et al., 2017), *Coordinate-wise Median* (Yin et al., 2018), and *Coordinate-wise Trimmed mean* (Yin et al., 2018).

**Adversary's goal.** While the server and the honest workers seek to minimize the global loss function defined in Equation (2) using Robust D-SGD, the adversary's goal is the exact opposite, i.e., to

maximize the loss function. We consider an external adversary that has open-box access to the learning procedure and can arbitrarily corrupt the stochastic gradients sent by the Byzantine workers at each iteration of Robust D-SGD. Specifically, the adversary seeks a solution to the following optimization problem:

$$
\max_{A_1,\ldots,A_T \in \mathbb{R}^{d \times f}} \quad \sum_{t=1}^{T} \mathcal{L}_{\mathcal{H}}(\theta_t)
$$

$$
\text{s.t.} \quad \theta_{t+1} = \theta_t - \gamma_t F\left(g_t^{(1)}, \ldots, g_t^{(n)}\right), \tag{P}
$$

$$
g_t^{(i)} = \nabla \mathcal{L}_i(\theta_t) + u_t^{(i)}, \quad \forall i \in \mathcal{H},
$$

$$
\left[g_t^{(i)}\right]_{i \notin \mathcal{H}} = A_t, \quad \forall t \in [T],
$$

where $u_t^{(i)}$ is the realization of a random vector $U(\theta_t)$ characterizing the *noise* in the gradient computation of the honest workers at $\theta_t$. An optimal adversarial strategy to Problem (P) is a set of stacked vectors $(A_1^*, \ldots, A_T^*)$ that ensures the global honest loss across the training trajectory is the largest possible. However, note that Problem (P) has *non-linear* equality constraints. Indeed, a robust aggregation $F$ must be non-linear, otherwise it can be arbitrarily manipulated by a single malicious vector (Blanchard et al., 2017). Moreover, in many cases, the aggregation $F$ is also non-differentiable, e.g., Krum and coordinate-wise median. Consequently, Problem (P) is hard to solve and does not admit an efficient solution in general (Boyd & Vandenberghe, 2004).

## 3 DESIGN OF THE JUMP ATTACK

In this section, we present our main contribution: the JUMP attack. First, in Section 3.1, we reduce the complexity of the adversary's problem by restricting its search space. Then, building upon this simplification, we introduce JUMP and describe it in Section 3.2.

### 3.1 SIMPLIFYING PROBLEM (P)

The adversary's Problem (P) has three axes of complexity: (i) the number of Byzantine workers $f$, (ii) the model dimension $d$, and (iii) the number of iterations $T$. Thus, for most modern-day machine learning tasks, solving the adversary's problem is highly challenging computationally. To reduce the hardness of Problem (P), we apply simplifications across each of the three aforementioned axes:

**(i) Number of Byzantine vectors.** First, we constrain all Byzantine vectors at any given iteration to be equal. This is a quite natural collusion strategy and is in fact how existing attacks are implemented (Karimireddy et al., 2022; Farhadkhani et al., 2022; Allouah et al., 2023). Indeed, robust aggregations generally produce a direction dictated by the "majority" of gradients. Accordingly, the Byzantine workers need to coordinate their actions to bias the output. This first simplification divides the number of variables in the Problem (P) by $f$. Now, at each iteration $t \geq 1$, the adversary only seeks a vector $a_t \in \mathbb{R}^d$ and sets $A_t = a_t \times \mathbf{1}^\top$, where $\mathbf{1} \in \mathbb{R}^f$ is the vector filled with ones.

**(ii) Model dimension.** Second, we constrain the Byzantine vector $a_t$, at each iteration $t \geq 1$, to be colinear with the average honest gradient, i.e., we set

$$
a_t = \lambda_t \cdot \overline{g}_t^{\mathcal{H}}, \ \forall t \in [T], \tag{4}
$$

where $\lambda_t \in \mathbb{R}$ and $\overline{g}_t^{\mathcal{H}} := \frac{1}{|\mathcal{H}|} \sum_{i \in \mathcal{H}} g_t^{(i)}$ is the average of honest gradients at iteration $t$. This constraint allows the adversary to bias the model in the ascent direction, which is a natural strategy for loss maximization, but also in the descent direction to provoke "overshooting" the minimum. This second simplification divides the number of variables of Problem (P) by $d$, which represents the main bottleneck for our problem using modern machine learning models.

**(iii) Time length.** Finally, given the integer parameter $\tau \in [T]$ we call *segment length*, we divide Problem (P) into $\lceil \frac{T}{\tau} \rceil$ smaller subproblems. Each subproblem, denoted $(P_\ell)$, is as follows:

$$\max_{\lambda_{(\ell-1)\tau+1}, \ldots, \lambda_{\ell\tau} \in \mathbb{R}} \quad \sum_{t=(\ell-1)\tau+1}^{\min\{\ell\tau, T\}} \mathcal{L}_\mathcal{H}(\theta_t)$$

$$\text{s.t.} \quad \theta_{t+1} = \theta_t - \gamma_t F\left(g_t^{(1)}, \ldots, g_t^{(n)}\right), \tag{$P_\ell$}$$

$$g_t^{(i)} = \nabla \mathcal{L}_i(\theta_t) + u_t^{(i)}, \quad \forall i \in \mathcal{H},$$

$$g_t^{(i)} = \lambda_t \cdot \overline{g}_t^\mathcal{H}, \, \forall i \notin \mathcal{H}, \quad \forall t \in \{(\ell-1)\tau+1, \ldots, \ell\tau\},$$

where $u_t^{(i)}$ above is as defined in Section 2. In words, we break down the time length of the initial Problem (P) into several segments of length $\tau$ each, which is also the number of variables of the subproblem $(P_\ell)$. Parameter $\tau$ controls the greediness of our attack strategy. For example, when $\tau = 1$, JUMP maximizes each iteration's loss independently.

## 3.2 ATTACK DESCRIPTION

We now introduce the JUMP attack strategy by building upon the above simplifications. JUMP works with the following inputs: initial model $\theta_1$, number of training iterations $T$, robust aggregation $F$, honest loss function $\mathcal{L}_\mathcal{H}$, local honest loss functions $\mathcal{L}_i$, and non-linear programming solver $S$. Regarding the latter, recall that the constraints of Problem $(P_\ell)$ are non-smooth in general due to robust aggregation. This motivates the use of derivative-free non-linear solvers such as Powell's (Powell, 1964), Nelder-Mead's (Nelder & Mead, 1965), and Differential Evolution (Storn & Price, 1997).

Given segment length parameter $\tau$, the JUMP attack planifies across $\lceil \frac{T}{\tau} \rceil$ segments. For each segment $\ell \in \left\{1, \ldots, \lceil \frac{T}{\tau} \rceil\right\}$, JUMP obtains $\lambda_{(\ell-1)\tau+1}, \ldots, \lambda_{\ell\tau}$ by applying solver $S$ to Problem $(P_\ell)$ initialized at model $\theta_{(\ell-1)\tau+1}$. The latter is $\theta_1$ if the segment is the first one ($\ell = 1$). Otherwise, it is obtained by using the solutions to Problem $(P_{\ell-1})$ and simulating the update rule in (3) with the (gradients of) honest local loss functions and robust aggregation $F$. After obtaining the solutions $\lambda_1, \ldots, \lambda_T$ across all subproblems, JUMP returns the malicious gradients following Equation (4). The full JUMP procedure is summarized in Algorithm 1.

---

**Algorithm 1** JUMP attack

---

**Input:** segment length $\tau$, solver $S$, initial model $\theta_1$
1: **for** $\ell$ **in** $1 \ldots \lceil \frac{T}{\tau} \rceil$ **do**
2:  Get $\lambda_{(\ell-1)\tau+1}, \ldots, \lambda_{\ell\tau}$ by solving Problem $(P_\ell)$ with initial model $\theta_{(\ell-1)\tau+1}$ using solver $S$
3: **end for**
4: **for** $t \in [T], i \notin \mathcal{H}$ **do**
5:  Set malicious gradient $a_t^{(i)} = \lambda_t \cdot \overline{g}_t^\mathcal{H}$
6: **end for**
7: **return** stack of malicious gradients $a$

---

Note that JUMP is readily applicable to any variant of Robust D-SGD, e.g. using local momentum (Karimireddy et al., 2021; Farhadkhani et al., 2022) or pre-aggregations (Karimireddy et al., 2022; Allouah et al., 2023). Also, for the greedy case $\tau = 1$, which does not require planning, the JUMP procedure can be conducted online during training by observing the honest gradients directly.

## 4 JUMPING OVER MINIMA: INSIGHTS IN TWO DIMENSIONS

In this section, we provide insights into JUMP and compare it with existing attacks on a two-dimensional non-convex toy problem. We first analyze the effect of the simplifications made in Section 3, and then explain the success of JUMP compared to existing attacks. We consider the setting with $n = 3$, $f = 1$ with two honest workers $w_1$ and $w_2$. For each honest worker $w_i$, the local loss function is set to:

$$\mathcal{L}_i(\theta) = 1 - b_i \cdot e^{-\|\theta - z_i\|_2^2}, \, \forall \theta \in \mathbb{R}^2, \tag{5}$$

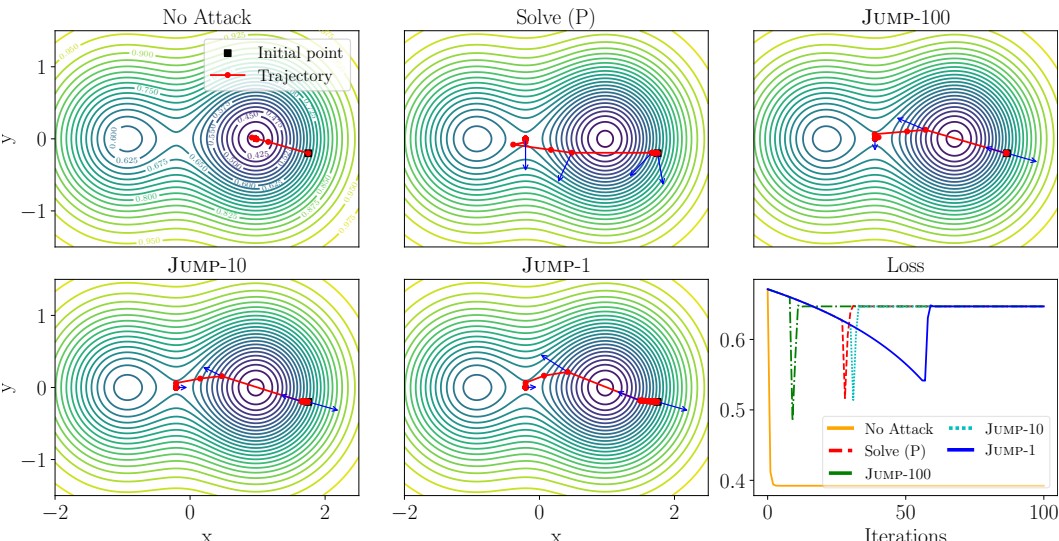

Figure 1: Training trajectories and loss across $T = 100$ steps of Robust D-SGD with the coordinate-wise median aggregation, initial point $\theta_1 = (1.75, -0.2)$, and learning rate $\gamma = 1.2$. JUMP-$\tau$ is the JUMP attack with segment length $\tau$. Solve (P) is the attack strategy obtained by solving Problem (P). For all attacks, we use Powell's method as solver. Blue arrows are the opposite of Byzantine gradients at key training steps.

where $\|\cdot\|_2$ denotes the Euclidean norm. Furthermore, we set $b_1 = 0.8$, $z_1 = (-1, 0)$, and $b_2 = 1.2$, $z_2 = (1, 0)$. Thus, the global honest loss $\mathcal{L}_\mathcal{H} = \frac{1}{2}(\mathcal{L}_1 + \mathcal{L}_2)$ has one local minimum at $(-1, 0)$ and one global minimum at $(1, 0)$ approximately. We show the landscape of the global loss in Figure 1.

### 4.1 IMPACT OF SIMPLIFICATIONS

We show the training trajectories of Robust D-SGD[2] in Figure 1, using JUMP with various segment lengths or by solving (P) directly, where we use the Powell solver. Our observations on coordinate-wise median also hold for other aggregations, hence we defer experiments on these to Appendix C.1.

**Common strategy.** We first highlight a phenomenon common to all attacks in Figure 1: the training trajectories (1) *resist* descent, then (2) *jump* over the global minimum, and finally (3) *escape* to a bad region of the landscape. This is in fact a natural strategy, as the honest workers are in majority and both have the same gradient direction, the global minimum inevitably attracts the trajectory. However, by sending the largest possible gradient in the same direction as the honest workers, the Byzantine worker can make the trajectory jump over the global minimum. After the jump, the honest gradients point in opposite directions because of their different minima, so it is easy from that point on to escape to a bad region of the landscape, e.g., a saddle point.

While all the considered JUMP instantiations follow the same strategy as that obtained by solving (P) directly, they have a slightly suboptimal performance due to the simplifications made in the design of JUMP. We first turn our attention to the constraint in Equation (4), where Byzantine gradients need to be colinear with the honest average gradient. Then, we analyze the impact of the segment length in JUMP, corresponding to the last simplification made in Section 3.

**Collinearity constraint.** We compare the attack obtained by solving (P) ("Solve (P)" in Figure 1) to that obtained using JUMP with full segment length $\tau = T = 100$ (JUMP-100). The main difference is that JUMP-100 performs a jump earlier in time but of lesser quality: the model reached after jumping is closer to the global minimum than "Solve (P)". This is due to the collinearity constraint, which precludes the trajectory obtained with the latter attack. Overall, JUMP-100 achieves a slightly lower total loss, with the benefit of significantly reducing the search space.

**Segment length constraint.** As shown in Figure 1, the larger the segment length, the earlier the jump, and the faster is the attack in escaping to the saddle point. This can be explained by the fact that JUMP tries to maximize the average loss over the segment as per Problem ($P_\ell$), so an earlier jump means a large loss for more training steps. Importantly, the greedy attack JUMP-1 shows equal

---

[2]Honest workers have one data point each, so they actually compute exact local gradients.

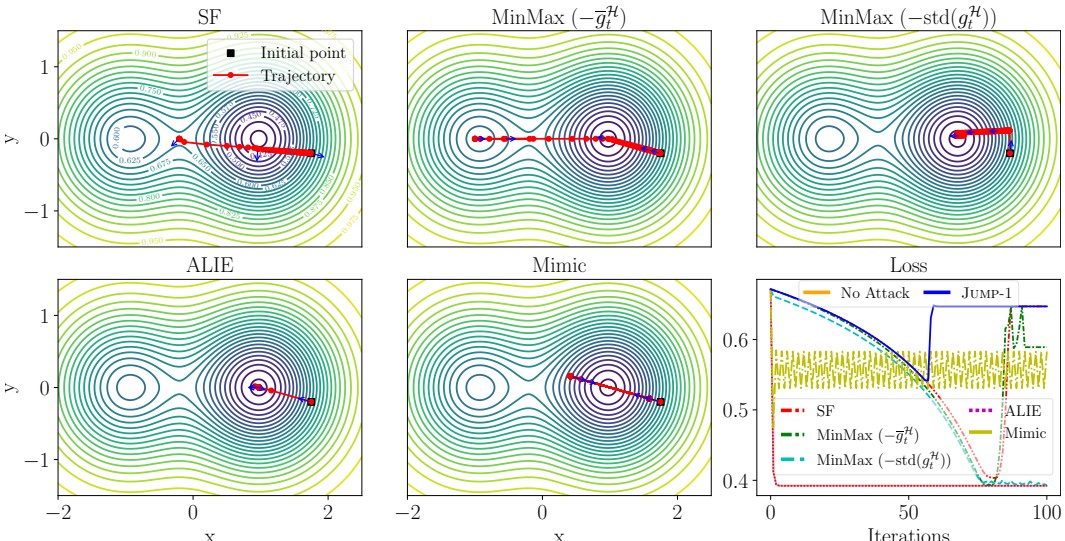

Figure 2: Training trajectories and loss across $T = 100$ steps of Robust D-SGD with the coordinate-wise median aggregation, initial point $\theta_1 = (1.75, -0.2)$, and learning rate $\gamma = 1.2$. We simulate the sign-flipping (SF), MinMax, Mimic and ALIE attacks, which are recalled in Section 4. We denote by $\mathrm{MinMax}\,(p_t)$ the use of perturbation vector $p_t$ in MinMax as in Equation (6). Blue arrows are the opposite of Byzantine gradients at key training steps.

asymptotic performance and thus appears as the natural candidate for larger scale learning tasks given its cheaper computational cost. We also confirm this observation on MNIST in Appendix A, where we quantify how $\tau = 1$ achieves the best (time) efficiency-power trade-off in JUMP.

### 4.2 COMPARISON WITH EXISTING ATTACKS

In general, existing attacks produce a Byzantine vector of the form:

$$a_t = \overline{g}_t^{\mathcal{H}} + \eta_t \cdot p_t, \tag{6}$$

where $p_t \in \mathbb{R}^d$ is a "perturbation" vector and $\eta_t > 0$ is a scale factor. For example, sign-flipping (SF) (Allen-Zhu et al., 2021) corresponds to $p_t = -\overline{g}_t^{\mathcal{H}}$ and $\eta_t = 2$. MinMax (Shejwalkar & Houmansadr, 2021) corresponds to maximizing $\eta_t$ via binary search under the constraint that the distance between the Byzantine vector and honest gradients is smaller than the maximum pairwise distance between the honest gradients. MinMax works with three possible perturbation vectors: $-\overline{g}_t^{\mathcal{H}}, -\mathrm{std}(g_t^{\mathcal{H}}), -\mathrm{sign}(g_t^{\mathcal{H}})$, where std and sign denote the coordinate-wise standard deviation and sign, respectively. Mimic (Karimireddy et al., 2022) sets $p_t = g_t^{(i)} - \overline{g}_t^{\mathcal{H}}$, where $i \in \mathcal{H}$ is heuristically chosen to maximize variance, and $\eta_t = 1$. For A Little Is Enough (ALIE) (Baruch et al., 2019), $p_t = -\mathrm{std}(g_t^{\mathcal{H}})$ and $\eta_t$ is a tunable hyperparameter.

In Figure 2, we show the training trajectories of Robust D-SGD using the coordinate-wise median aggregation (Yin et al., 2018) under the SF, MinMax, ALIE and Mimic attacks recalled above. Observe that both SF and MinMax with $p_t = -\overline{g}_t^{\mathcal{H}}$ follow a strategy where they (1) resist descent, (2) cross the global minimum, and (3) reach a bad region of the landscape. Unlike JUMP, they do not circumvent the global minimum, which is undesirable, e.g., if the defender uses early stopping in training. It is also worth mentioning that MinMax with $p_t = -\overline{g}_t^{\mathcal{H}}$ converges to the local minimum, and SF to the saddle point. Besides, ALIE and MinMax with $p_t = -\mathrm{std}(g_t^{\mathcal{H}})$ both resist descent, only to converge to the global minimum. Interestingly, Mimic periodically jumps around the global minimum, but these jumps are too small and result in a loss constantly lower than that of JUMP.

**Key reasons of superiority.** We can explain the superior performance of JUMP by two key design features: (a) JUMP maximizes the global *loss* directly, and (b) it can send a large vector in the *same direction* as the honest gradient. Regarding (a), recall that MinMax also optimizes the scale factor, but it maximizes the distance of the aggregation output to honest gradient and thus cannot leverage the landscape information. As per (b), this is a crucial feature of the design of JUMP which enables long jumps over minima, as explained earlier. In comparison, the scale factor in MinMax is positive, and so the attack cannot both resist descent and jump for any choice of perturbation vector. Evidently, other attacks such as SF, Mimic, and ALIE have the same limitation.

| Agg. | $f$ | ALIE | SF | LF | Mimic | MinMax | AGRT | JUMP (ours) |
|------|-----|------|-----|-----|-------|--------|------|-------------|
| CM   | 1 | $80.6 \pm 0.8$ | $79.9 \pm 0.3$ | $80.6 \pm 0.6$ | $79.5 \pm 0.3$ | $77.2 \pm 2.6$ | $75.0 \pm 1.6$ | $76.5 \pm 2.5$ |
|      | 3 | $78.4 \pm 0.5$ | $76.1 \pm 0.8$ | $75.9 \pm 0.4$ | $77.6 \pm 0.8$ | $70.9 \pm 3.1$ | $56.2 \pm 6.9$ | $43.2 \pm 5.2$ |
|      | 5 | $58.9 \pm 4.3$ | $51.8 \pm 9.2$ | $51.7 \pm 3.0$ | $73.4 \pm 0.9$ | $65.1 \pm 6.9$ | $37.3 \pm 5.7$ | $28.5 \pm 3.0$ |
| Krum | 1 | $79.3 \pm 1.9$ | $78.8 \pm 1.4$ | $78.4 \pm 0.8$ | $78.1 \pm 1.1$ | $76.9 \pm 3.9$ | $70.8 \pm 2.6$ | $72.3 \pm 1.8$ |
|      | 3 | $76.8 \pm 1.6$ | $70.7 \pm 3.7$ | $71.7 \pm 1.9$ | $75.3 \pm 1.7$ | $65.4 \pm 10.8$ | $47.5 \pm 4.0$ | $43.1 \pm 4.4$ |
|      | 5 | $58.9 \pm 3.4$ | $43.0 \pm 8.7$ | $46.4 \pm 1.4$ | $72.1 \pm 1.5$ | $61.9 \pm 7.2$ | $33.9 \pm 8.2$ | $25.4 \pm 0.4$ |
| GM   | 1 | $81.0 \pm 0.4$ | $80.8 \pm 0.2$ | $81.0 \pm 0.3$ | $80.5 \pm 0.2$ | $79.2 \pm 1.0$ | $77.3 \pm 0.5$ | $78.2 \pm 0.9$ |
|      | 3 | $78.7 \pm 0.5$ | $76.0 \pm 1.1$ | $76.4 \pm 0.4$ | $78.3 \pm 0.5$ | $71.7 \pm 4.0$ | $64.3 \pm 3.9$ | $45.5 \pm 3.3$ |
|      | 5 | $60.7 \pm 2.0$ | $55.6 \pm 4.8$ | $52.9 \pm 2.3$ | $74.0 \pm 1.0$ | $68.5 \pm 4.1$ | $40.7 \pm 5.7$ | $28.5 \pm 4.1$ |
| TM   | 1 | $81.0 \pm 0.3$ | $81.1 \pm 0.2$ | $81.1 \pm 0.4$ | $80.8 \pm 0.3$ | $79.5 \pm 1.0$ | $79.1 \pm 0.5$ | $78.9 \pm 0.5$ |
|      | 3 | $78.5 \pm 0.3$ | $76.6 \pm 0.6$ | $76.4 \pm 0.6$ | $78.2 \pm 0.6$ | $73.6 \pm 3.1$ | $65.8 \pm 2.6$ | $50.2 \pm 3.1$ |
|      | 5 | $60.5 \pm 3.9$ | $59.4 \pm 4.6$ | $51.3 \pm 1.8$ | $74.0 \pm 0.7$ | $68.4 \pm 3.0$ | $46.3 \pm 4.4$ | $31.9 \pm 5.6$ |

Table 1: Comparison of JUMP with state-of-the-art attacks on CIFAR-10 under moderate heterogeneity ($\alpha = 1$). In green , we emphasize cases where JUMP is significantly more powerful than any other attack. In light green , we show cases where JUMP is comparable to the best among other attacks: JUMP's accuracy is the lowest and its confidence interval overlaps with some other attack's. The baseline accuracy is $81.3 \pm 0.3$ in the absence of attacks ($f = 0$).

# 5 EXPERIMENTS

In this section, we evaluate the performance of JUMP in comparison with current state-of-the-art attacks on heterogeneous versions of two benchmark image classification datasets: MNIST (LeCun & Cortes, 2005) and CIFAR-10 (Krizhevsky et al., 2009).

## 5.1 SETUP

**Data heterogeneity, models, and defenses.** We set the number of honest workers to 12, i.e. $n - f = 12$, and vary the number of Byzantine workers $f$ in $\{1, 3, 5\}$. We simulate data heterogeneity across honest workers using the Dirichlet distribution as in (Hsu et al., 2019; Lin et al., 2020). This distribution is parameterized by a real-value $\alpha$, which basically controls the level of heterogeneity. We consider three values for $\alpha$, specifically 0.1, 1 and 10, simulating high, moderate, and low heterogeneity, respectively. We use convolutional neural networks (CNNs) for each dataset, architectural details deferred to Appendix B.1. We train the models using Robust D-SGD with local momentum using the same hyperparameters as in (Karimireddy et al., 2022; Allouah et al., 2023). We test several robust aggregations: coordinate-wise median (CM) and trimmed mean (TM) (Yin et al., 2018), Krum (Blanchard et al., 2017), and geometric median (GM) (Pillutla et al., 2022). We also augment these aggregation rules using the state-of-the-art NNM pre-aggregation method, proven to impart optimal robustness under heterogeneity (Allouah et al., 2023).

**Byzantine attacks.** We simulate the following Byzantine attacks: A Little Is Enough (ALIE) (Baruch et al., 2019), Fall Of Empires (FOE) (Xie et al., 2020), Sign-Flipping (SF) and Label Flipping (LF) (Allen-Zhu et al., 2021), MinMax and MinSum (Shejwalkar & Houmansadr, 2021), and Mimic (Karimireddy et al., 2022). Unless stated otherwise, we implement JUMP with segment length $\tau = 1$, which we argued is both sufficiently powerful and computationally cheap. More details on the implementation of attacks are given in Appendix B.3.

**Reproducibility.** All experiments are run with five seeds from 1 to 5 for reproducibility and 95% confidence intervals are shown on plots and tables. Our code will be made public for reusability.

## 5.2 RESULTS

In Table 1, we show a representative comparison of all attacks on the CIFAR-10 dataset with moderate heterogeneity ($\alpha = 1$). We also show the accuracy across training steps in Figure 3 for $f \in \{1, 3\}$ using GM on MNIST and CIFAR-10 with high heterogeneity ($\alpha = 0.1$). We defer the full results on all attacks to appendices C.2 and C.3.

**Consistent superiority.** Our results on CIFAR-10, in Table 1, show that JUMP is significantly more powerful in most cases and achieves the lowest (best) accuracy, especially when the number of

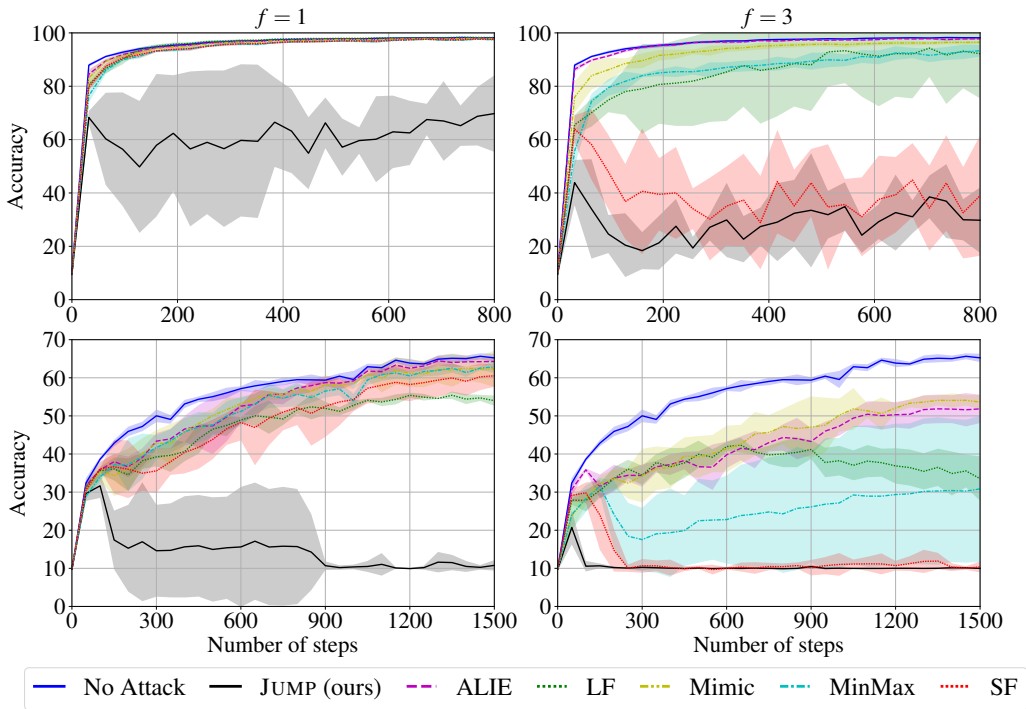

Figure 3: Accuracy evolution of Robust D-SGD using Geometric Median on MNIST (1st row) and CIFAR-10 (2nd row), with high data heterogeneity ($\alpha = 0.1$), under JUMP and other representative attacks for $f = 1$ (1st column) and $f = 3$ (2nd column) Byzantine workers.

Byzantine workers is large enough, i.e. $f \in \{3, 5\}$, and is *always* at least as powerful as any other attack, regardless of the number of Byzantine workers. On average, JUMP decreases the baseline accuracy by 31.1 accuracy points, and the lowest accuracy across existing attacks by 15.9. As such, JUMP is almost *two times* more damaging than any existing attack. In contrast, there is no clear winner among existing attacks: for example, MinMax is the most damaging for $f = 3$ on CM, LF is the most damaging for $f = 5$ on TM, among considered attacks in Table 1. Finally, we remark that JUMP inflicts the most damage regardless of the aggregation rule used by the defense.

**Breakdown point shift.** Under high data heterogeneity, i.e., $\alpha = 0.1$, we observe in Figure 3 that JUMP significantly amplifies the impact of even a single Byzantine worker (i.e., $f = 1$). While the defense scheme in the case of CIFAR-10 can tackle other attacks, it is rendered completely ineffective against JUMP. The detrimental impact of JUMP is also consistently larger than other attacks in the case of MNIST. We make similar observations when $f = 3$, where we note that the SF attack also achieves a comparable performance asymptotically. Importantly, in the absence of a strong stress test like JUMP, one could make false conclusions about the strength of existing defenses. Our results show that, if the threat model is strong enough to include worst-case adversaries, the breakdown point of Robust D-SGD might actually be lower than was previously believed.

## 6 CONCLUSION AND FUTURE WORK

Byzantine robustness is critical when using distributed learning in data-sensitive applications. In this work, we present the first holistic approach to designing a powerful attack truly embodying the omniscient Byzantine adversary, ultimately leading to the design of the JUMP attack. Our findings show a clear gap between the empirical performance of state-of-the-art defenses and their theoretical robustness claims. Moreover, we extensively demonstrate the superiority of JUMP over state-of-the-art attacks across various data heterogeneity levels and Byzantine fractions. We believe that more benchmarks are needed to evaluate robustness on real-world tasks, for which we strongly encourage the usage of JUMP as a reliable stress test. Future work can also build upon the minima circumvention strategy of JUMP to design powerful and realistic attacks for weaker threat models.

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

ORGANIZATION OF THE APPENDIX

Appendix A contains an analysis of the trade-off between time efficiency and attack strength in JUMP. Appendix B contains the detailed experimental setups. Finally, Appendix C contains the comprehensive experimental results.

## A   EFFICIENCY-POWER TRADE-OFF IN JUMP

In this section, we evaluate the dependence of the time efficiency and attack strength of JUMP on segment length $\tau$, by experimenting on the MNIST (LeCun & Cortes, 2005) image classification task with a logistic regression model. More details on the experimental setup are given in Appendix B.1. We show the maximal accuracy under the JUMP attack as well as the runtime ratio of JUMP compared to the SF attack across different segment lengths in Figure 4.

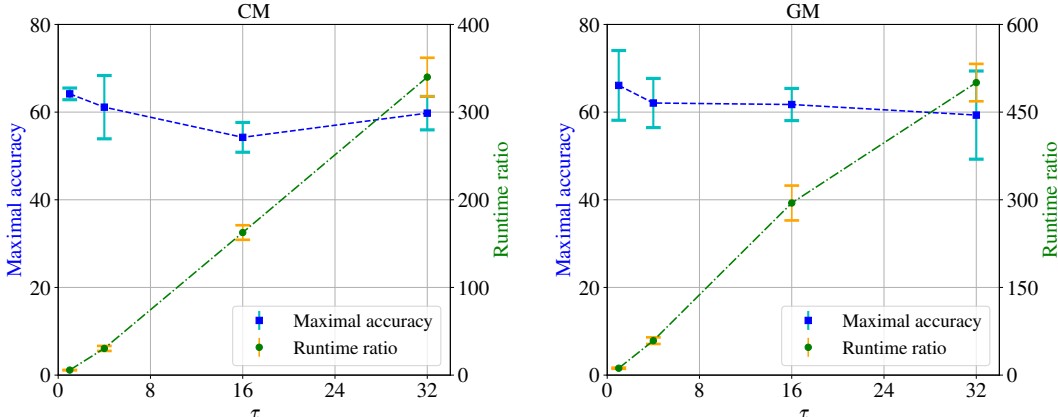

Figure 4: Evolution of maximal accuracy (corresponding to left y-axis) and runtime ratio to the SF attack's (corresponding to right y-axis) for the CM (1st column) and GM (2nd column) aggregations by varying segment length $\tau \in \{1, 4, 16, 32\}$ on MNIST with logistic regression. Each experiment is run 5 times and we show the 95% confidence interval.

**Reasonable time efficiency when $\tau = 1$.** The runtime ratios of JUMP with $\tau = 1$ are $5.7\times$ for CM and $11.8\times$ for GM. However, given that SF's time efficiency approaches the baseline when there is no attack, deploying JUMP takes reasonable time for stress testing robust aggregation rules.

**The longer $\tau$, the stronger the attack.** In Figure 4, we observe a general trend where the maximal accuracy under the JUMP attack decreases as $\tau$ increases on both CM and GM. This is expected, as a longer segment length allows for better trajectory planning. However, an exception is noted at $\tau = 32$ for CM, where the maximal accuracy shows an increase. We attribute this to the Powell solver starting to lose its effectiveness in higher dimensions and complex landscapes, a characteristic inherent to non-linear programming solvers.

**Greedy ($\tau = 1$) achieves the best efficiency-power trade-off.** In Figure 4, the runtime of JUMP is roughly proportional to the segment length $\tau$, with slope 15 approximately. Such a trend is problematic, especially since the maximal accuracy does not significantly decrease when increasing $\tau$. The gain in reducing maximal accuracy provided by a longer segment length is *marginal* compared to the significant increase in runtime. Therefore, we argue that $\tau = 1$ offers the most favorable efficiency-power trade-off. Nonetheless, if the aim is to construct the most powerful attack regardless of computational resources, we still furnish the option to conduct the JUMP attack with $\tau > 1$.

## B   EXPERIMENTAL SETUPS

In this section, we presents the detailed experimental setups. Appendix B.1 presents training settings for the classification task on the MNIST and CIFAR10 datasets, Appendix B.2 shows settings of robust aggregation rules, and Appendix B.3 contains settings of attacks.

## B.1 Experimental Setup for the Classification Task on MNIST and CIFAR-10

We present the architecture of the CNN models and additional details on the experimental setup in Table 2, in which the setup of MNIST follows Karimireddy et al. (2022) and CIFAR-10's follows Allouah et al. (2023). All our experiments were conducted on the following hardware: 2 NVIDIA A10 24GB GPUs and 8 NVIDIA GeForce GTX TITAN X 12GB GPUs. We use a compact notation to define the models as in El Mhamdi et al. (2021); Allouah et al. (2023).

L(#*outputs*) for a fully-connected linear layer with #*outputs* dimensions of output, R for ReLU activation, C(#*channels*) for a fully-connected 2D-convolutional layer with #*channels* channels of output, M for 2D-maxpool (kernel size 2), B for batch-normalization, and D(#*probability*) for dropout with probability #*probability*.

| Dataset | MNIST | CIFAR-10 |
|---|---|---|
| Architecture | (1,28,28)-C(32)-R-C(64)-M-D(0.25)-L(128)-R-D(0.5)-L(10) | (3,32×32)-C(64)-R-B-C(64)-R-B-M-D(0.25)-C(128)-R-B-C(128)-R-B-M-D(0.25)-L(128)-R-D(0.25)-L(10) |
| Loss | Cross Entropy | Cross Entropy |
| Preprocessing | Normalization | Random Horizontal Flip, Normalization |
| Pre-aggregation rules | NNM | NNM |
| Aggregation rules | CM, TM, GM, Krum | CM, TM, GM, Krum |
| Gradient clipping | 2 | 5 |
| $l_2$-regularization | $10^{-4}$ | $10^{-2}$ |
| Number of steps | $T = 800$ | $T = 1500$ |
| Batch size | $b = 32$ | $b = 64$ |
| Learning rate | $\gamma_t = 0.1$ | $\begin{cases} \gamma_t = 0.25, & t \leq 1000 \\ \gamma_t = 0.025, & 1000 < t \leq 1500 \end{cases}$ |
| Momentum parameter | $\beta \in \{0, 0.9\}$ | $\beta \in \{0, 0.9, 0.99\}$ |
| Number of honest workers | $n - f = 12$ | $n - f = 12$ |
| Number of Byzantine workers | $f \in \{1, 3, 5\}$ | $f \in \{1, 3, 5\}$ |
| Random seeds | $1, 2, 3, 4, 5$ | $1, 2, 3, 4, 5$ |
| Heterogeneity | $\alpha \in \{0.1, 1, 10\}$ | $\alpha \in \{0.1, 1, 10\}$ |

Table 2: Model Architectures, hyperparameters, and distributed settings for MNIST and CIFAR-10 experiments.

We present the architecture of the MNIST logistic regression model and additional details on the experimental setup used in efficiency-power trade-off experiments in Table 3.

## B.2 Aggregation Rules

On all datasets, we implement the pre-aggregation method Nearest Neighbor Mixing (NNM) (Allouah et al., 2023) with four robust aggregation rules namely Geometric Median (GM) (Pillutla et al., 2022), Coordinate-wise Median (CM) (Yin et al., 2018), Coordinate-wise Trimmed Mean (TM) (Yin et al., 2018), and Krum (Blanchard et al., 2017), following Allouah et al. (2023). The hyperparameters of these four aggregation rules are listed in Table 4 below. Note that in the two-dimensional landscape experiments, we use a slight adaption of Krum as in (Farhadkhani et al., 2022; Allouah et al., 2023), in which Krum outputs the vector that is the nearest to its neighbors upon discarding $f$, as opposed to $f + 1$ in the original version (Blanchard et al., 2017), which allows using Krum for $n = 3, f = 1$. In other experiments, we use the original version.

## B.3 Byzantine Attacks

We use seven state-of-the-art attacks – LF (Allen-Zhu et al., 2021), SF (Allen-Zhu et al., 2021), ALIE (Baruch et al., 2019), FOE (Xie et al., 2020), Mimic (Karimireddy et al., 2022), MinMax (Shejwalkar & Houmansadr, 2021), and MinSum (Shejwalkar & Houmansadr, 2021) – as references for

evaluating the JUMP attack. The hyperparameter settings of these seven attacks as well as our JUMP attack follow the formalization in Section 4.2 and are listed in Table 5.

| Dataset | MNIST (Logistic Regression) |
|---|---|
| Architecture | (1,28,28)-L(10) |
| Loss | Cross Entropy |
| Preprocessing | Normalization |
| Pre-aggregation rules | NNM |
| Aggregation rules | CM, GM |
| Gradient clipping | 2 |
| $l_2$-regularization | N/A |
| Number of steps | $T = 640$ |
| Batch size | $b = 32$ |
| Learning rate | $\gamma_t = 0.7$ |
| Momentum parameter | $\beta = 0.9$ |
| Number of honest workers | $n - f = 12$ |
| Number of Byzantine workers | $f = 3$ |
| Random seeds | $1, 2, 3, 4, 5$ |
| Heterogeneity | $\alpha = 0.1$ |

Table 3: Model architectures, hyperparameters, and distributed settings for efficiency-power trade-off experiments. N/A indicates that we do not set the respective parameter.

| Agg. | Hyperparameters |
|---|---|
| Krum | N/A |
| CM | N/A |
| GM | $T = 8$ |
| TM | $b = f$ |

Table 4: Hyperparameters of aggregation rules considered. N/A indicates that no parameter is required for the respective method. $T$ is the smoothing parameter in Smoothed Weiszfeld (Pillutla et al., 2022) method to calculate geometric median. $b$ is the number of the largest and smallest gradients to be trimmed in trimmed mean method.

| Attacks | Hyperparameters |
|---|---|
| LF | $p_t = a_t - \overline{g}_t^{\mathcal{H}}$, $a_t$ is the gradient on flipped labels $(9 - l)$, $\eta_t = 1$ |
| SF | $p_t = -\overline{g}_t^{\mathcal{H}}$, $\eta_t = 2$ |
| ALIE | $p_t = -\text{std}(g_t^{\mathcal{H}})$, $\eta_t = argmax_z \left( \phi(z) < \frac{n-f-s}{n-f} \right), s = \lfloor \frac{n}{2} + 1 \rfloor - f$ |
| FOE | $p_t = -\overline{g}_t^{\mathcal{H}}$, $\eta_t = 1.1$ |
| Mimic | $p_t = g_t^{(i)} - \overline{g}_t^{\mathcal{H}}$, $i \in \mathcal{H}$ is heuristically chosen to maximize variance, $\eta_t = 1$ |
| MinMax | $p_t = -\frac{\overline{g}_t^{\mathcal{H}}}{\left\| \overline{g}_t^{\mathcal{H}} \right\|_2}$, $\eta_t$ is obtained via binary search in $[0, 50]$ with threshold $10^{-5}$ |
| MinSum | $p_t = -\frac{\overline{g}_t^{\mathcal{H}}}{\left\| \overline{g}_t^{\mathcal{H}} \right\|_2}$, $\eta_t$ is obtained via binary search in $[0, 50]$ with threshold $10^{-5}$ |
| JUMP | $p_t = -\overline{g}_t^{\mathcal{H}}$, $\eta_t$ with initialization $\eta_{\text{init}} = 0$ is obtained by the Powell solver without bound constraints. |

Table 5: The hyperparameter settings of considered attacks. In the LF attack, $l \in \{0, 1, \cdots, 9\}$ denotes the original label of a sample. All other notations are in accordance with Equation (6). For JUMP, utilizing the notation from Section 3, we optimize $\lambda_t$ with initialization $\lambda_{\text{init}} = 1$ by the Powell solver without bound constraints.

## C    FULL RESULTS

This section presents our full experimental results. Appendix C.1 presents our two-dimensional toy problem results, Appendix C.2 contains the full tabular numerical results on MNIST and CIFAR-10, and Appendix C.3 presents the accuracy evolution plots on MNIST and CIFAR-10.

### C.1    TWO-DIMENSIONAL NON-CONVEX TOY PROBLEM

In this section, we present the entirety of our results on the two-dimensional toy problem. Compared to Section 4, we additionally include the MinSum and MinMax attacks with all possible perturbation vectors (Shejwalkar & Houmansadr, 2021). We also include the same experiments for the GM and Krum aggregations, TM being exactly equal to CM in our setting $n = 3, f = 1$. Overall, the same observations made in Section 4 continue to hold. That is, the collinearity constraint introduces a minor suboptimality in the performance of JUMP compared to solving the original adversary's problem. Moreover, a larger segment length also means a marginally better performance, with the general strategy being the same independently of the segment length. Besides, the MinSum and MinMax attacks behave similarly to the versions of MinMax analyzed in Section 4. Finally, the FOE attack generally behaves like SF, which is expected given that they use the same perturbation vector (Shejwalkar & Houmansadr, 2021).

**Results on CM and TM.**    The results with the coordinate-wise median aggregation are in figures 5, 6, and 7.

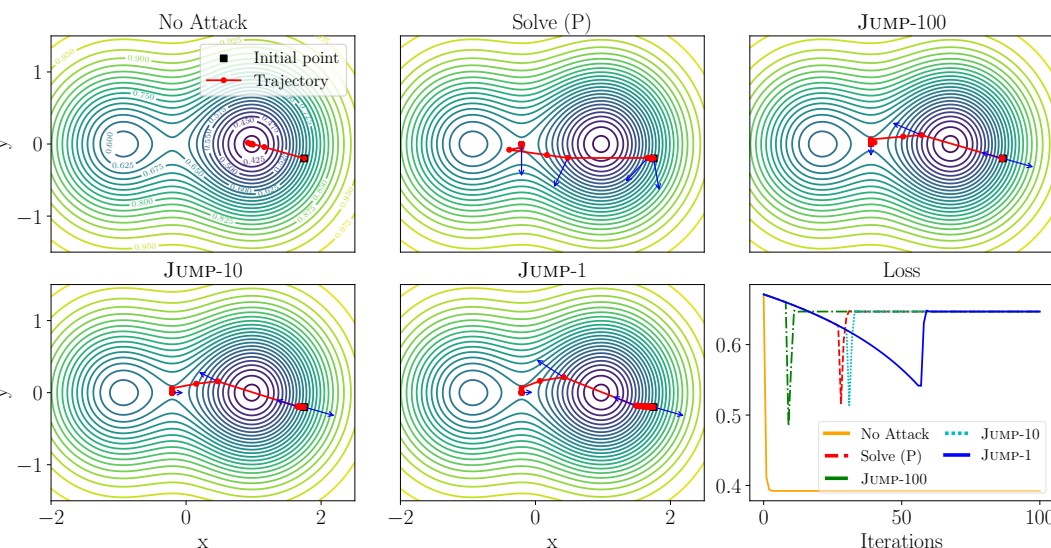

Figure 5: Training trajectories and loss across $T = 100$ steps of Robust D-SGD with the coordinate-wise median aggregation, initial point $\theta_1 = (1.75, -0.2)$, and learning rate $\gamma = 1.2$. JUMP-$\tau$ is the JUMP attack with segment length $\tau$. Solve (P) is the attack strategy obtained by solving Problem (P). For all attacks, we use Powell's method as solver. Blue arrows are the opposite of Byzantine gradients at key training steps.

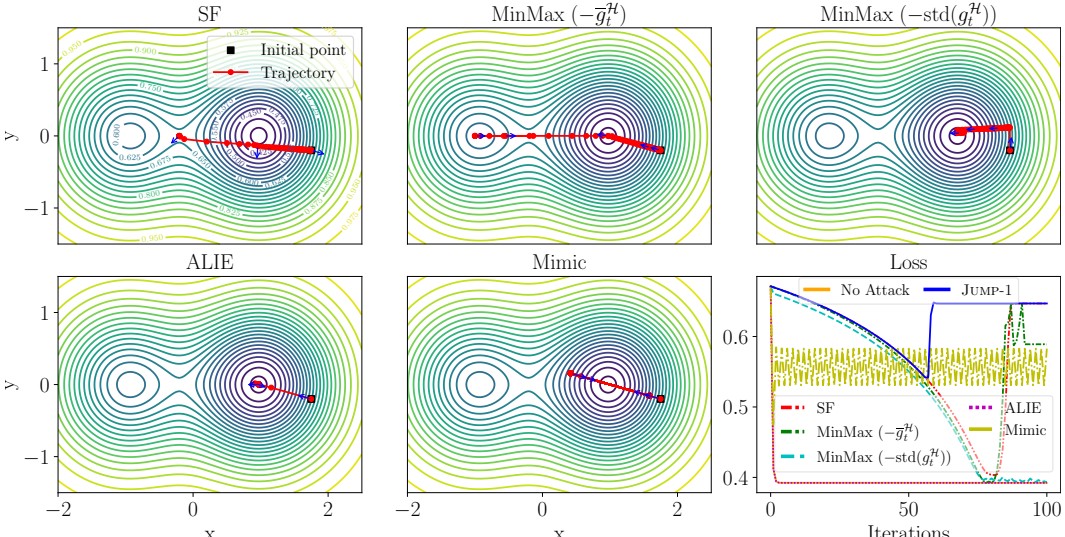

Figure 6: Training trajectories and loss across $T = 100$ steps of Robust D-SGD with the coordinate-wise median aggregation, initial point $\theta_1 = (1.75, -0.2)$, and learning rate $\gamma = 1.2$. We simulate the sign-flipping (SF), MinMax, Mimic and ALIE attacks, which are recalled in Section 4. We denote by MinMax $(p_t)$ the use of perturbation vector $p_t$ in MinMax as in Equation (6). Blue arrows are the opposite of Byzantine gradients at key training steps.

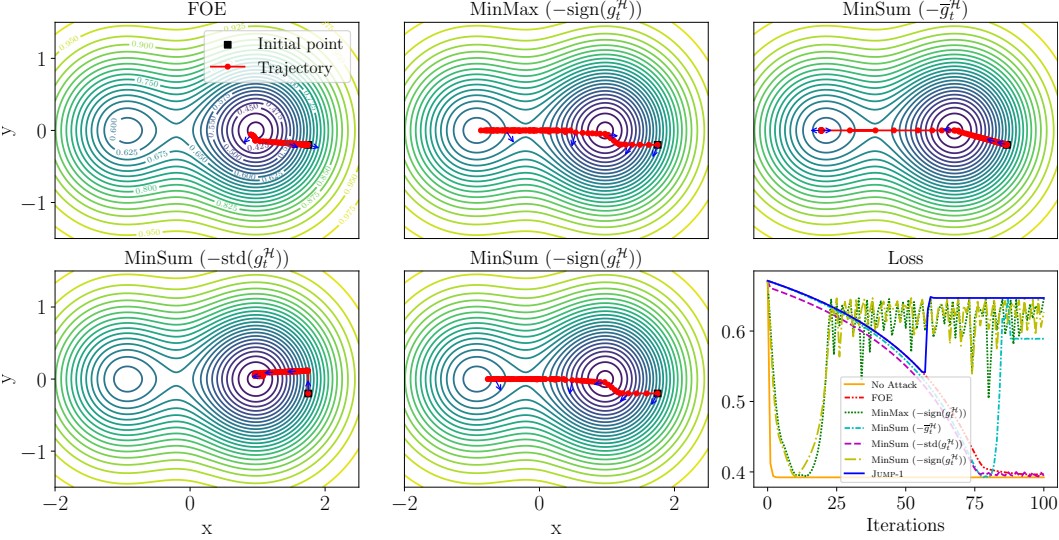

Figure 7: Training trajectories and loss across $T = 100$ steps of Robust D-SGD with the coordinate-wise median aggregation, initial point $\theta_1 = (1.75, -0.2)$, and learning rate $\gamma = 1.2$. We simulate the Fall Of Empire (FOE), MinMax, and MinSum attacks. We denote by MinMax $(p_t)$ and MinSum $(p_t)$ the use of perturbation vector $p_t$ in MinMax and MinSum as in Equation (6). Blue arrows are the opposite of Byzantine gradients at key training steps.

**Results on GM.**    The results with the geometric median aggregation are in figures 8, 9, and 10.

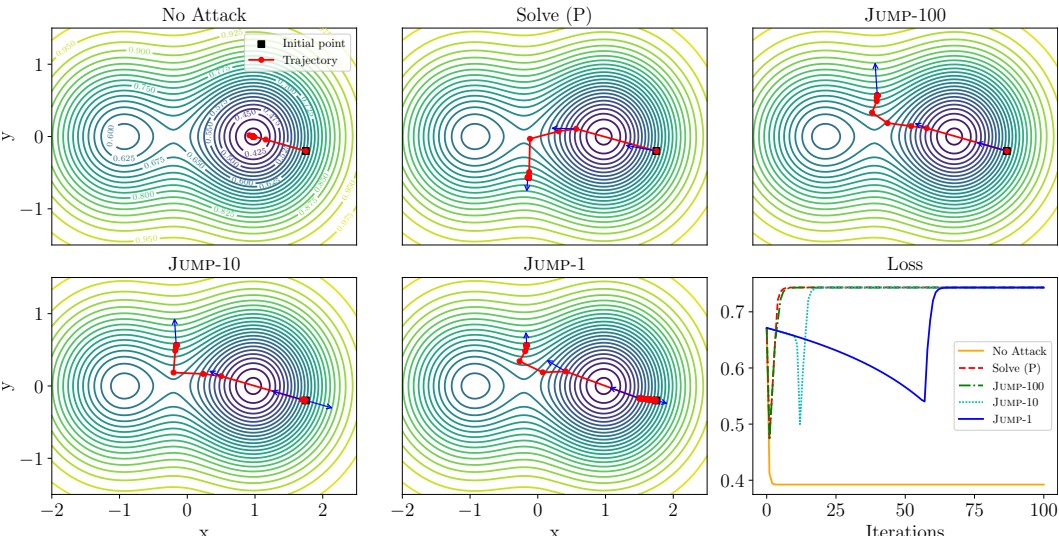

Figure 8: Training trajectories and loss across $T = 100$ steps of Robust D-SGD with the geometric median aggregation, initial point $\theta_1 = (1.75, -0.2)$, and learning rate $\gamma = 1.2$. JUMP-$\tau$ is the JUMP attack with segment length $\tau$. Solve (P) is the attack strategy obtained by solving Problem (P). For all attacks, we use Powell's method as solver. Blue arrows are the opposite of Byzantine gradients at key training steps.

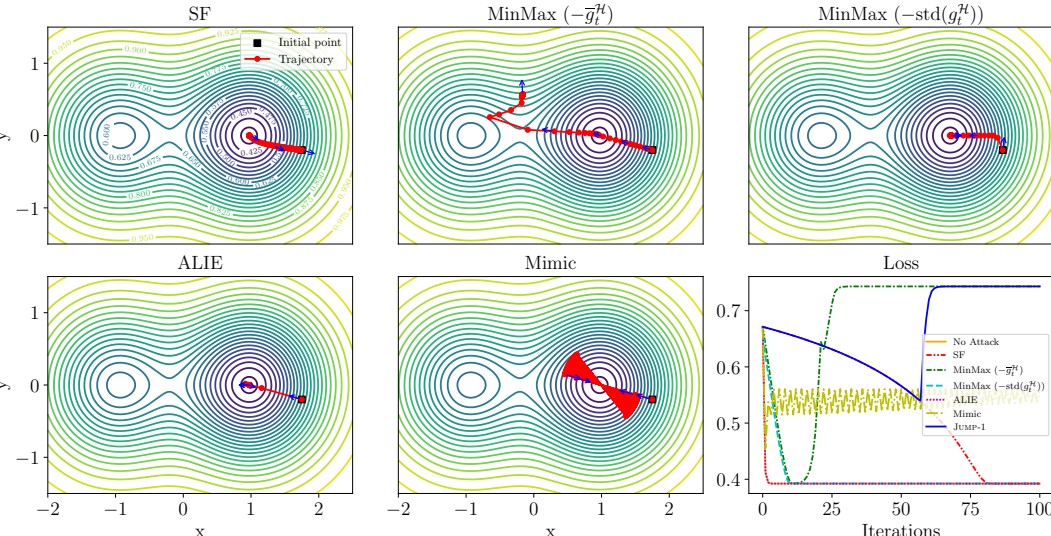

Figure 9: Training trajectories and loss across $T = 100$ steps of Robust D-SGD with the geometric median aggregation, initial point $\theta_1 = (1.75, -0.2)$, and learning rate $\gamma = 1.2$. We simulate the sign-flipping (SF), MinMax, Mimic and ALIE attacks, which are recalled in Section 4. We denote by MinMax $(p_t)$ the use of perturbation vector $p_t$ in MinMax as in Equation (6). Blue arrows are the opposite of Byzantine gradients at key training steps.

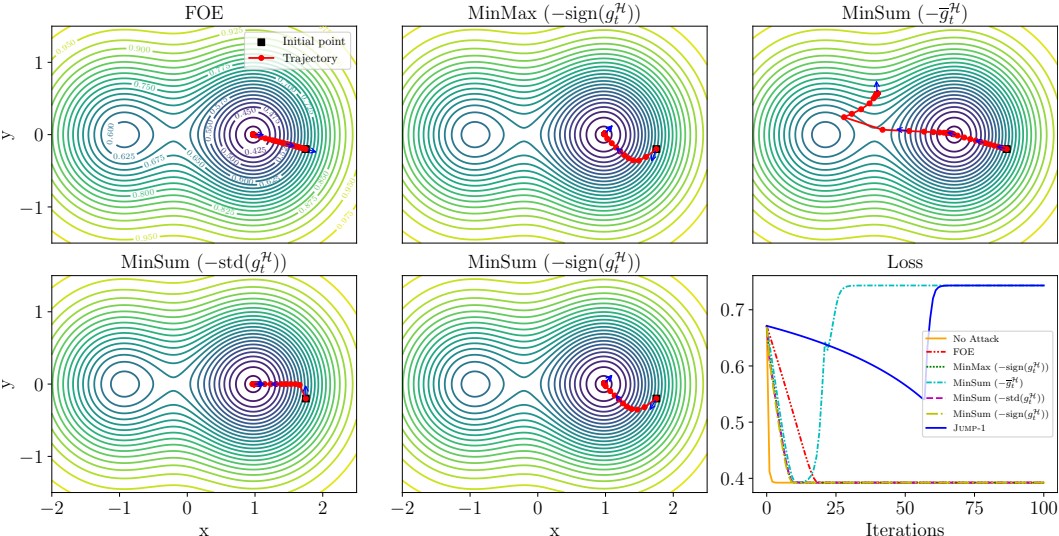

Figure 10: Training trajectories and loss across $T = 100$ steps of Robust D-SGD with the geometric median aggregation, initial point $\theta_1 = (1.75, -0.2)$, and learning rate $\gamma = 1.2$. We simulate the Fall Of Empire (FOE), MinMax, and MinSum attacks. We denote by $\mathrm{MinMax}\ (p_t)$ and $\mathrm{MinSum}\ (p_t)$ the use of perturbation vector $p_t$ in MinMax and MinSum as in Equation (6). Blue arrows are the opposite of Byzantine gradients at key training steps.

**Results on Krum.** The results with the Krum aggregation are in figures 11, 12, and 13.

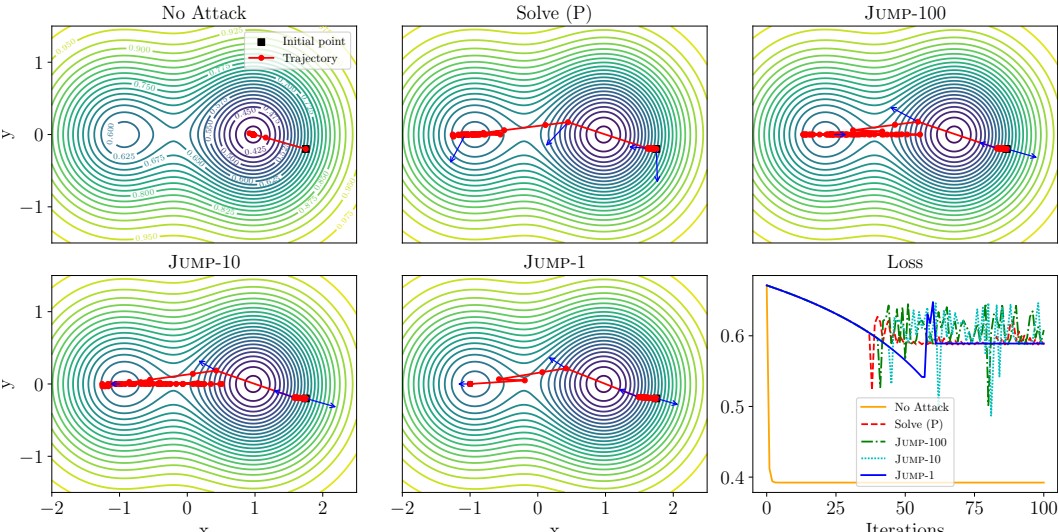

Figure 11: Training trajectories and loss across $T = 100$ steps of Robust D-SGD with the Krum aggregation, initial point $\theta_1 = (1.75, -0.2)$, and learning rate $\gamma = 1.2$. JUMP-$\tau$ is the JUMP attack with segment length $\tau$. Solve (P) is the attack strategy obtained by solving Problem (P). For all attacks, we use Powell's method as solver. Blue arrows are the opposite of Byzantine gradients at key training steps.

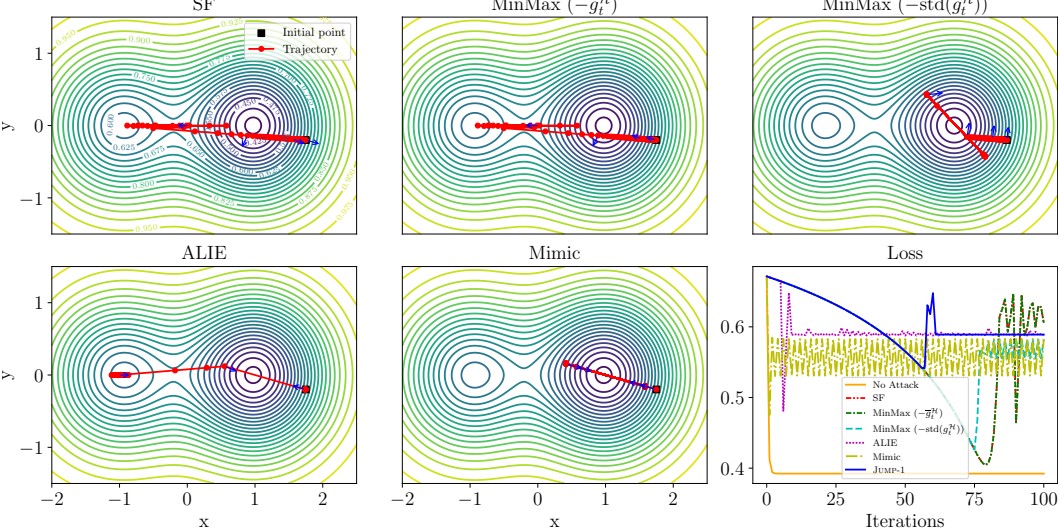

Figure 12: Training trajectories and loss across $T = 100$ steps of Robust D-SGD with the Krum aggregation, initial point $\theta_1 = (1.75, -0.2)$, and learning rate $\gamma = 1.2$. We simulate the sign-flipping (SF), MinMax, Mimic and ALIE attacks, which are recalled in Section 4. We denote by MinMax $(p_t)$ the use of perturbation vector $p_t$ in MinMax as in Equation (6). Blue arrows are the opposite of Byzantine gradients at key training steps.

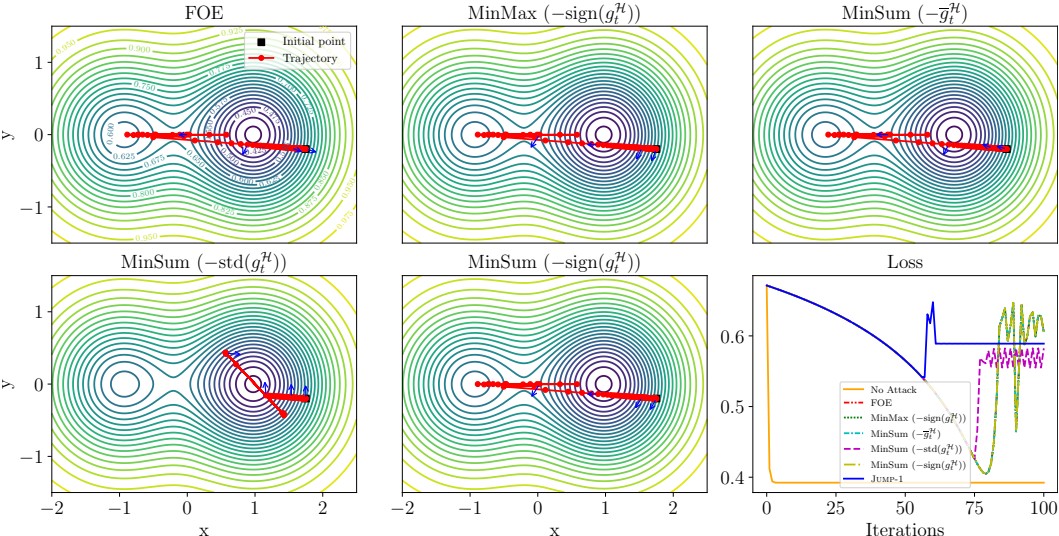

Figure 13: Training trajectories and loss across $T = 100$ steps of Robust D-SGD with the Krum aggregation, initial point $\theta_1 = (1.75, -0.2)$, and learning rate $\gamma = 1.2$. We simulate the Fall Of Empire (FOE), MinMax, and MinSum attacks. We denote by $\mathrm{MinMax}\ (p_t)$ and $\mathrm{MinSum}\ (p_t)$ the use of perturbation vector $p_t$ in MinMax and MinSum as in Equation (6). Blue arrows are the opposite of Byzantine gradients at key training steps.

## C.2 FULL NUMERICAL RESULTS ON MNIST AND CIFAR-10

Here we provide the full numerical results on MNIST (Table 6) and CIFAR-10 (Table 7) when heterogeneity is high ($\alpha = 0.1$), moderate ($\alpha = 1$), and low ($\alpha = 10$). The reader can focus on the color code detailed in the caption of each table. The results essentially convey the same message as in Section 5, where JUMP outperforms existing attacks across the vast majority of scenarios. It is worth mentioning that the results for $\alpha = 10$ are slightly different than those of Section 5 because of the low data heterogeneity. Indeed, in the latter regime, the considered defenses all have excellent accuracies across all attacks, which is predicted by the theory in i.i.d. settings when using local momentum (Farhadkhani et al., 2022).

| agg | alpha | f | momentum | ALIE | FOE | LF | Mimic | MinMax | MinSum | SF | No Attack | JUMP | Superiority |
|---|---|---|---|---|---|---|---|---|---|---|---|---|---|
| AVG | 0.1 | 0 | 0.0 | - | - | - | - | - | - | - | 98.4 ± 0.1 | - | NaN |
| | | | 0.9 | - | - | - | - | - | - | - | 98.3 ± 0.2 | - | NaN |
| | 1.0 | 0 | 0.0 | - | - | - | - | - | - | - | 98.4 ± 0.1 | - | NaN |
| | | | 0.9 | - | - | - | - | - | - | - | 98.4 ± 0.1 | - | NaN |
| | 10.0 | 0 | 0.0 | - | - | - | - | - | - | - | 98.5 ± 0.1 | - | NaN |
| | | | 0.9 | - | - | - | - | - | - | - | 98.5 ± 0.1 | - | NaN |
| CM | 0.1 | 1 | 0.0 | 98.3 ± 0.2 | 98.2 ± 0.2 | 98.4 ± 0.1 | 98.2 ± 0.1 | 97.7 ± 0.1 | 98.2 ± 0.1 | 98.1 ± 0.1 | - | 93.0 ± 13.9 | -4.7 |
| | | | 0.9 | 98.1 ± 0.1 | 98.0 ± 0.1 | 98.3 ± 0.1 | 98.0 ± 0.2 | 97.6 ± 0.2 | 97.8 ± 0.2 | 97.7 ± 0.1 | - | 76.9 ± 12.1 | -20.7 |
| | | 3 | 0.0 | 97.3 ± 0.3 | 94.4 ± 1.0 | 98.5 ± 0.1 | 97.5 ± 0.2 | 87.9 ± 4.8 | 96.5 ± 0.7 | 66.2 ± 5.6 | - | 58.5 ± 5.7 | -7.7 |
| | | | 0.9 | 97.9 ± 0.2 | 93.2 ± 3.3 | 94.1 ± 11.6 | 96.0 ± 1.1 | 92.1 ± 3.0 | 95.2 ± 2.2 | 65.7 ± 4.9 | - | 45.9 ± 5.9 | -19.8 |
| | | 5 | 0.0 | 90.8 ± 1.4 | 77.0 ± 2.1 | 63.0 ± 3.5 | 93.6 ± 2.7 | 84.8 ± 2.6 | 89.6 ± 1.5 | 53.2 ± 9.5 | - | 55.2 ± 2.3 | 2.0 |
| | | | 0.9 | 95.6 ± 0.5 | 84.4 ± 14.9 | 58.3 ± 6.1 | 92.4 ± 3.2 | 83.9 ± 4.7 | 95.0 ± 3.7 | 42.9 ± 23.1 | - | 33.3 ± 8.7 | -9.6 |
| | 1.0 | 1 | 0.0 | 98.4 ± 0.1 | 98.3 ± 0.1 | 98.5 ± 0.1 | 98.4 ± 0.1 | 98.0 ± 0.1 | 98.3 ± 0.1 | 98.2 ± 0.2 | - | 98.1 ± 0.1 | 0.1 |
| | | | 0.9 | 98.1 ± 0.1 | 98.4 ± 0.1 | 98.4 ± 0.2 | 98.4 ± 0.0 | 98.1 ± 0.1 | 98.2 ± 0.1 | 98.3 ± 0.1 | - | 98.3 ± 0.2 | 0.2 |
| | | 3 | 0.0 | 97.7 ± 0.1 | 97.8 ± 0.2 | 98.5 ± 0.1 | 98.2 ± 0.2 | 94.5 ± 0.3 | 97.7 ± 0.1 | 97.0 ± 0.3 | - | 94.0 ± 0.7 | -0.5 |
| | | | 0.9 | 98.4 ± 0.1 | 98.1 ± 0.2 | 98.4 ± 0.1 | 98.3 ± 0.1 | 97.0 ± 0.4 | 97.5 ± 0.3 | 97.4 ± 0.3 | - | 98.3 ± 0.1 | 1.3 |
| | | 5 | 0.0 | 94.2 ± 1.0 | 97.0 ± 0.3 | 98.5 ± 0.1 | 97.7 ± 0.2 | 93.1 ± 0.7 | 95.9 ± 0.6 | 83.6 ± 3.7 | - | 82.1 ± 3.2 | -1.5 |
| | | | 0.9 | 97.8 ± 0.2 | 97.5 ± 0.2 | 98.5 ± 0.1 | 98.0 ± 0.1 | 96.2 ± 0.3 | 96.6 ± 0.5 | 95.2 ± 0.6 | - | 97.7 ± 0.3 | 2.5 |
| | 10.0 | 1 | 0.0 | 98.4 ± 0.1 | 98.4 ± 0.1 | 98.5 ± 0.2 | 98.4 ± 0.1 | 98.1 ± 0.2 | 98.3 ± 0.1 | 98.3 ± 0.2 | - | 98.2 ± 0.2 | 0.1 |
| | | | 0.9 | 98.5 ± 0.1 | 98.4 ± 0.2 | 98.5 ± 0.1 | 98.5 ± 0.2 | 98.3 ± 0.2 | 98.3 ± 0.2 | 98.3 ± 0.1 | - | 98.3 ± 0.1 | 0.1 |
| | | 3 | 0.0 | 97.7 ± 0.3 | 97.9 ± 0.2 | 98.5 ± 0.2 | 98.2 ± 0.1 | 95.0 ± 1.0 | 97.7 ± 0.3 | 97.0 ± 0.4 | - | 95.2 ± 0.4 | 0.2 |
| | | | 0.9 | 98.4 ± 0.1 | 98.2 ± 0.2 | 98.5 ± 0.1 | 98.3 ± 0.1 | 97.1 ± 0.3 | 97.5 ± 0.3 | 97.7 ± 0.2 | - | 98.2 ± 0.2 | 1.1 |
| | | 5 | 0.0 | 94.5 ± 0.8 | 97.1 ± 0.2 | 98.5 ± 0.1 | 97.8 ± 0.1 | 93.5 ± 0.7 | 95.8 ± 0.6 | 89.9 ± 1.7 | - | 88.8 ± 0.9 | -1.1 |
| | | | 0.9 | 98.2 ± 0.1 | 97.7 ± 0.3 | 98.5 ± 0.1 | 98.1 ± 0.1 | 95.7 ± 0.6 | 96.1 ± 0.5 | 95.5 ± 0.6 | - | 98.0 ± 0.4 | 2.5 |
| GM | 0.1 | 1 | 0.0 | 98.2 ± 0.1 | 98.2 ± 0.1 | 98.4 ± 0.1 | 98.2 ± 0.1 | 97.8 ± 0.2 | 98.2 ± 0.2 | 98.1 ± 0.1 | - | 90.1 ± 13.4 | -7.7 |
| | | | 0.9 | 98.2 ± 0.1 | 98.0 ± 0.1 | 98.4 ± 0.2 | 98.0 ± 0.1 | 97.7 ± 0.2 | 97.8 ± 0.2 | 97.7 ± 0.1 | - | 80.4 ± 11.2 | -17.3 |
| | | 3 | 0.0 | 97.2 ± 0.4 | 95.7 ± 0.3 | 98.5 ± 0.1 | 97.6 ± 0.2 | 90.1 ± 3.3 | 96.8 ± 0.5 | 66.5 ± 3.7 | - | 61.9 ± 5.6 | -4.6 |
| | | | 0.9 | 98.0 ± 0.2 | 93.6 ± 3.4 | 94.4 ± 10.9 | 96.7 ± 0.6 | 93.4 ± 2.3 | 95.2 ± 0.7 | 66.2 ± 4.5 | - | 52.5 ± 5.0 | -13.7 |
| | | 5 | 0.0 | 90.9 ± 2.2 | 80.3 ± 2.5 | 67.5 ± 6.0 | 95.0 ± 2.0 | 86.1 ± 0.7 | 91.3 ± 1.4 | 54.3 ± 8.9 | - | 56.6 ± 3.3 | 2.3 |
| | | | 0.9 | 95.0 ± 1.1 | 85.0 ± 13.7 | 59.0 ± 6.3 | 94.7 ± 1.9 | 86.0 ± 0.8 | 94.8 ± 3.3 | 47.1 ± 14.0 | - | 30.9 ± 10.8 | -16.2 |
| | 1.0 | 1 | 0.0 | 98.4 ± 0.1 | 98.4 ± 0.1 | 98.5 ± 0.1 | 98.4 ± 0.1 | 98.0 ± 0.1 | 98.4 ± 0.1 | 98.2 ± 0.1 | - | 98.2 ± 0.2 | 0.2 |
| | | | 0.9 | 98.4 ± 0.1 | 98.3 ± 0.1 | 98.5 ± 0.2 | 98.4 ± 0.1 | 98.1 ± 0.2 | 98.2 ± 0.1 | 98.3 ± 0.1 | - | 98.3 ± 0.1 | 0.2 |
| | | 3 | 0.0 | 97.7 ± 0.1 | 97.9 ± 0.2 | 98.5 ± 0.1 | 98.2 ± 0.1 | 94.8 ± 0.7 | 97.7 ± 0.3 | 96.9 ± 0.4 | - | 94.3 ± 0.4 | -0.5 |
| | | | 0.9 | 98.4 ± 0.2 | 98.0 ± 0.2 | 98.5 ± 0.1 | 98.3 ± 0.1 | 97.0 ± 0.4 | 97.4 ± 0.3 | 97.4 ± 0.2 | - | 98.2 ± 0.2 | 1.2 |
| | | 5 | 0.0 | 94.2 ± 1.4 | 97.1 ± 0.3 | 98.5 ± 0.1 | 97.8 ± 0.2 | 92.4 ± 0.5 | 95.9 ± 0.6 | 83.8 ± 1.7 | - | 83.7 ± 7.4 | -2.1 |
| | | | 0.9 | 98.0 ± 0.2 | 97.6 ± 0.2 | 98.5 ± 0.1 | 98.1 ± 0.1 | 96.2 ± 0.4 | 96.7 ± 0.4 | 95.1 ± 0.8 | - | 97.9 ± 0.1 | 2.8 |
| | 10.0 | 1 | 0.0 | 98.4 ± 0.1 | 98.3 ± 0.1 | 98.5 ± 0.1 | 98.4 ± 0.1 | 98.1 ± 0.1 | 98.3 ± 0.2 | 98.3 ± 0.1 | - | 98.2 ± 0.2 | 0.1 |
| | | | 0.9 | 98.5 ± 0.1 | 98.4 ± 0.2 | 98.5 ± 0.1 | 98.5 ± 0.1 | 98.2 ± 0.2 | 98.3 ± 0.2 | 98.3 ± 0.2 | - | 98.3 ± 0.1 | 0.1 |
| | | 3 | 0.0 | 97.8 ± 0.2 | 97.9 ± 0.2 | 98.5 ± 0.1 | 98.2 ± 0.1 | 94.7 ± 0.5 | 97.7 ± 0.2 | 97.0 ± 0.3 | - | 95.2 ± 0.5 | 0.5 |
| | | | 0.9 | 98.4 ± 0.1 | 98.2 ± 0.2 | 98.5 ± 0.2 | 98.3 ± 0.2 | 97.1 ± 0.4 | 97.4 ± 0.3 | 97.7 ± 0.2 | - | 98.2 ± 0.2 | 1.1 |
| | | 5 | 0.0 | 94.6 ± 0.6 | 97.2 ± 0.2 | 98.6 ± 0.6 | 97.8 ± 0.1 | 93.5 ± 0.7 | 95.9 ± 0.6 | 90.0 ± 1.3 | - | 88.4 ± 1.7 | -1.6 |
| | | | 0.9 | 98.2 ± 0.1 | 97.7 ± 0.3 | 98.5 ± 0.1 | 98.1 ± 0.2 | 95.7 ± 0.4 | 96.1 ± 0.5 | 95.5 ± 0.6 | - | 98.0 ± 0.2 | 2.5 |
| Krum | 0.1 | 1 | 0.0 | 98.3 ± 0.1 | 98.1 ± 0.2 | 98.4 ± 0.1 | 98.2 ± 0.1 | 97.8 ± 0.1 | 98.1 ± 0.1 | 98.1 ± 0.2 | - | 82.0 ± 11.0 | -15.8 |
| | | | 0.9 | 98.1 ± 0.2 | 97.9 ± 0.2 | 98.4 ± 0.1 | 98.0 ± 0.2 | 97.7 ± 0.3 | 97.9 ± 0.3 | 97.8 ± 0.4 | - | 80.4 ± 12.3 | -17.3 |
| | | 3 | 0.0 | 97.2 ± 0.2 | 94.3 ± 0.4 | 98.5 ± 0.1 | 97.3 ± 0.3 | 90.1 ± 3.4 | 96.4 ± 0.5 | 66.7 ± 8.3 | - | 59.1 ± 4.8 | -7.6 |
| | | | 0.9 | 97.9 ± 0.2 | 92.8 ± 4.0 | 95.7 ± 7.1 | 96.2 ± 0.7 | 91.7 ± 3.1 | 94.2 ± 1.0 | 67.0 ± 8.6 | - | 46.8 ± 3.0 | -20.2 |
| | | 5 | 0.0 | 90.9 ± 1.5 | 78.0 ± 3.3 | 63.3 ± 2.1 | 93.7 ± 2.9 | 86.5 ± 0.4 | 88.9 ± 1.6 | 45.1 ± 7.1 | - | 47.6 ± 4.5 | 2.5 |
| | | | 0.9 | 93.2 ± 3.3 | 85.5 ± 13.0 | 60.0 ± 8.3 | 92.2 ± 3.6 | 86.3 ± 2.1 | 93.7 ± 3.9 | 38.1 ± 20.5 | - | 23.6 ± 4.0 | -14.5 |
| | 1.0 | 1 | 0.0 | 98.4 ± 0.1 | 98.3 ± 0.1 | 98.5 ± 0.1 | 98.4 ± 0.2 | 98.0 ± 0.1 | 98.3 ± 0.2 | 98.3 ± 0.1 | - | 98.2 ± 0.2 | 0.2 |
| | | | 0.9 | 98.4 ± 0.1 | 98.4 ± 0.1 | 98.4 ± 0.2 | 98.4 ± 0.1 | 98.1 ± 0.1 | 98.2 ± 0.2 | 98.3 ± 0.1 | - | 98.3 ± 0.2 | 0.2 |
| | | 3 | 0.0 | 97.7 ± 0.1 | 97.8 ± 0.2 | 98.5 ± 0.1 | 98.1 ± 0.1 | 94.7 ± 0.6 | 97.7 ± 0.2 | 97.0 ± 0.3 | - | 93.2 ± 0.4 | -1.5 |
| | | | 0.9 | 98.4 ± 0.2 | 98.0 ± 0.3 | 98.5 ± 0.1 | 98.3 ± 0.2 | 97.0 ± 0.4 | 97.5 ± 0.2 | 97.4 ± 0.3 | - | 98.2 ± 0.1 | 1.2 |
| | | 5 | 0.0 | 94.4 ± 0.6 | 97.1 ± 0.4 | 98.4 ± 0.1 | 97.7 ± 0.1 | 92.9 ± 1.7 | 95.9 ± 0.6 | 84.2 ± 2.4 | - | 78.7 ± 5.6 | -5.5 |
| | | | 0.9 | 98.1 ± 0.1 | 97.6 ± 0.3 | 98.5 ± 0.1 | 98.1 ± 0.1 | 96.0 ± 0.6 | 96.7 ± 0.3 | 94.9 ± 0.7 | - | 97.8 ± 0.2 | 2.9 |
| | 10.0 | 1 | 0.0 | 98.4 ± 0.1 | 98.3 ± 0.2 | 98.5 ± 0.2 | 98.4 ± 0.1 | 98.1 ± 0.2 | 98.3 ± 0.1 | 98.3 ± 0.1 | - | 98.3 ± 0.2 | 0.2 |
| | | | 0.9 | 98.5 ± 0.2 | 98.4 ± 0.2 | 98.5 ± 0.1 | 98.5 ± 0.1 | 98.2 ± 0.2 | 98.3 ± 0.2 | 98.3 ± 0.2 | - | 98.3 ± 0.1 | 0.1 |
| | | 3 | 0.0 | 97.8 ± 0.2 | 97.9 ± 0.1 | 98.5 ± 0.1 | 98.2 ± 0.1 | 94.9 ± 0.4 | 97.7 ± 0.2 | 97.0 ± 0.3 | - | 94.3 ± 0.4 | -0.6 |
| | | | 0.9 | 98.4 ± 0.1 | 98.2 ± 0.2 | 98.5 ± 0.1 | 98.4 ± 0.1 | 97.1 ± 0.3 | 97.5 ± 0.3 | 97.6 ± 0.2 | - | 98.2 ± 0.1 | 1.1 |
| | | 5 | 0.0 | 93.8 ± 1.1 | 97.1 ± 0.2 | 98.5 ± 0.1 | 97.9 ± 0.4 | 93.3 ± 1.0 | 95.9 ± 0.6 | 89.1 ± 0.8 | - | 87.3 ± 1.2 | -1.8 |
| | | | 0.9 | 98.2 ± 0.1 | 97.7 ± 0.3 | 98.5 ± 0.1 | 98.0 ± 0.2 | 95.6 ± 0.8 | 96.3 ± 0.6 | 95.5 ± 0.7 | - | 98.1 ± 0.1 | 2.6 |
| TM | 0.1 | 1 | 0.0 | 98.3 ± 0.1 | 98.2 ± 0.2 | 98.4 ± 0.1 | 98.2 ± 0.1 | 97.8 ± 0.2 | 98.1 ± 0.2 | 98.1 ± 0.1 | - | 98.0 ± 0.2 | 0.2 |
| | | | 0.9 | 98.1 ± 0.2 | 97.9 ± 0.2 | 98.3 ± 0.2 | 98.0 ± 0.1 | 97.6 ± 0.2 | 97.8 ± 0.2 | 97.7 ± 0.2 | - | 78.7 ± 0.9 | -18.9 |
| | | 3 | 0.0 | 97.3 ± 0.2 | 95.1 ± 0.8 | 98.5 ± 0.1 | 97.4 ± 0.3 | 89.1 ± 2.8 | 96.6 ± 0.8 | 65.8 ± 3.5 | - | 66.7 ± 2.1 | 0.9 |
| | | | 0.9 | 98.0 ± 0.1 | 91.5 ± 2.5 | 90.6 ± 13.1 | 96.3 ± 0.6 | 91.8 ± 3.0 | 94.4 ± 0.9 | 68.8 ± 4.4 | - | 53.9 ± 5.2 | -14.9 |
| | | 5 | 0.0 | 90.5 ± 1.6 | 78.1 ± 2.7 | 64.8 ± 1.0 | 93.9 ± 2.8 | 84.5 ± 2.6 | 89.1 ± 1.6 | 52.6 ± 3.5 | - | 59.1 ± 2.2 | 6.5 |
| | | | 0.9 | 95.7 ± 0.8 | 83.8 ± 14.8 | 58.0 ± 6.9 | 92.7 ± 3.3 | 84.0 ± 3.0 | 94.3 ± 3.9 | 37.4 ± 10.5 | - | 37.1 ± 4.0 | -0.3 |
| | 1.0 | 1 | 0.0 | 98.4 ± 0.1 | 98.4 ± 0.1 | 98.4 ± 0.1 | 98.4 ± 0.1 | 98.1 ± 0.1 | 98.4 ± 0.1 | 98.3 ± 0.1 | - | 98.3 ± 0.2 | 0.2 |
| | | | 0.9 | 98.4 ± 0.0 | 98.3 ± 0.1 | 98.4 ± 0.1 | 98.4 ± 0.1 | 98.0 ± 0.2 | 98.2 ± 0.1 | 98.3 ± 0.1 | - | 98.4 ± 0.1 | 0.4 |
| | | 3 | 0.0 | 97.6 ± 0.2 | 97.8 ± 0.2 | 98.5 ± 0.1 | 98.2 ± 0.1 | 94.8 ± 0.5 | 97.7 ± 0.3 | 97.0 ± 0.3 | - | 95.1 ± 0.4 | 0.3 |
| | | | 0.9 | 98.4 ± 0.2 | 98.1 ± 0.2 | 98.5 ± 0.1 | 98.3 ± 0.1 | 97.0 ± 0.4 | 97.4 ± 0.3 | 97.4 ± 0.3 | - | 98.2 ± 0.2 | 1.2 |

Continued on next page

| agg | alpha | f | momentum | ALIE | FOE | LF | Mimic | MinMax | MinSum | SF | No Attack | JUMP | Superiority |
|---|---|---|---|---|---|---|---|---|---|---|---|---|---|
| | | 5 | 0.0 | 93.5 ± 0.4 | 97.1 ± 0.3 | 98.4 ± 0.1 | 97.7 ± 0.2 | 93.2 ± 1.8 | 95.9 ± 0.5 | 88.2 ± 2.8 | - | 87.4 ± 2.3 | -0.8 |
| | | | 0.9 | 98.0 ± 0.3 | 97.6 ± 0.2 | 98.5 ± 0.1 | 98.0 ± 0.1 | 96.1 ± 0.5 | 96.6 ± 0.4 | 95.4 ± 0.7 | - | 97.9 ± 0.2 | 2.5 |
| | 10.0 | 1 | 0.0 | 98.4 ± 0.1 | 98.3 ± 0.1 | 98.5 ± 0.2 | 98.4 ± 0.1 | 98.1 ± 0.2 | 98.3 ± 0.1 | 98.3 ± 0.1 | - | 98.3 ± 0.2 | 0.2 |
| | | | 0.9 | 98.5 ± 0.2 | 98.4 ± 0.2 | 98.5 ± 0.1 | 98.4 ± 0.2 | 98.2 ± 0.2 | 98.2 ± 0.3 | 98.3 ± 0.2 | - | 98.3 ± 0.2 | 0.1 |
| | | 3 | 0.0 | 97.7 ± 0.2 | 97.9 ± 0.2 | 98.5 ± 0.2 | 98.2 ± 0.2 | 94.9 ± 0.9 | 97.7 ± 0.2 | 97.0 ± 0.3 | - | 95.7 ± 0.3 | 0.8 |
| | | | 0.9 | 98.4 ± 0.1 | 98.2 ± 0.2 | 98.5 ± 0.1 | 98.3 ± 0.1 | 97.1 ± 0.4 | 97.4 ± 0.3 | 97.7 ± 0.2 | - | 98.2 ± 0.2 | 1.1 |
| | | 5 | 0.0 | 94.1 ± 1.3 | 97.0 ± 0.3 | 98.5 ± 0.1 | 97.8 ± 0.2 | 93.7 ± 0.6 | 95.9 ± 0.7 | 91.7 ± 1.3 | - | 90.3 ± 0.9 | -1.4 |
| | | | 0.9 | 98.2 ± 0.1 | 97.7 ± 0.3 | 98.5 ± 0.1 | 98.1 ± 0.1 | 95.7 ± 0.5 | 96.0 ± 0.6 | 95.6 ± 0.6 | - | 98.1 ± 0.2 | 2.5 |

Table 6: Comparison of JUMP with state-of-the-art attacks on MNIST under heterogeneity $\alpha \in \{0.1, 1, 10\}$. In green, we emphasize cases where JUMP is significantly more powerful than any other attack. Under the column "Superiority", we report the difference in accuracy between the best existing attack and JUMP. In light green, we show cases where JUMP is comparable to the best among other attacks: JUMP's accuracy is one of the lowest and its confidence interval overlaps with some other attack's. In red, we show cases where JUMP is weaker than some attack.

| agg | alpha | f | momentum | ALIE | FOE | LF | Mimic | MinMax | MinSum | SF | No Attack | JUMP | Superiority |
|---|---|---|---|---|---|---|---|---|---|---|---|---|---|
| AVG | 0.1 | 0 | 0.0 | - | - | - | - | - | - | - | 66.5 ± 0.4 | - | NaN |
| | | | 0.9 | - | - | - | - | - | - | - | 67.1 ± 0.7 | - | NaN |
| | | | 0.99 | - | - | - | - | - | - | - | 65.8 ± 0.7 | - | NaN |
| | 1.0 | 0 | 0.0 | - | - | - | - | - | - | - | 81.3 ± 0.4 | - | NaN |
| | | | 0.9 | - | - | - | - | - | - | - | 82.4 ± 0.3 | - | NaN |
| | | | 0.99 | - | - | - | - | - | - | - | 81.3 ± 0.3 | - | NaN |
| | 10.0 | 0 | 0.0 | - | - | - | - | - | - | - | 82.4 ± 0.2 | - | NaN |
| | | | 0.9 | - | - | - | - | - | - | - | 83.5 ± 0.2 | - | NaN |
| | | | 0.99 | - | - | - | - | - | - | - | 82.3 ± 0.5 | - | NaN |
| CM | 0.1 | 1 | 0.0 | 65.1 ± 0.4 | 65.3 ± 0.7 | 55.2 ± 7.7 | 64.9 ± 0.7 | 64.0 ± 0.8 | 65.1 ± 0.9 | 63.2 ± 2.6 | - | 40.5 ± 20.2 | -14.7 |
| | | | 0.9 | 66.5 ± 0.4 | 66.2 ± 0.6 | 51.6 ± 2.1 | 63.3 ± 5.0 | 64.6 ± 0.5 | 66.4 ± 0.8 | 62.1 ± 6.3 | - | 22.7 ± 11.2 | -28.9 |
| | | | 0.99 | 64.3 ± 1.1 | 62.4 ± 3.3 | 54.6 ± 1.9 | 61.4 ± 5.2 | 62.7 ± 1.5 | 59.6 ± 4.2 | | - | 30.6 ± 5.1 | -24.0 |
| | | 3 | 0.0 | 57.1 ± 0.9 | 51.7 ± 0.3 | 40.0 ± 2.2 | 55.6 ± 6.1 | 42.0 ± 5.6 | 57.3 ± 5.9 | 22.3 ± 5.0 | - | 13.3 ± 2.6 | -9.0 |
| | | | 0.9 | 62.2 ± 1.1 | 59.8 ± 2.2 | 39.3 ± 1.7 | 63.1 ± 3.1 | 53.9 ± 10.1 | 64.4 ± 3.5 | 29.1 ± 3.7 | - | 11.3 ± 1.3 | -17.8 |
| | | | 0.99 | 47.8 ± 5.0 | 55.6 ± 3.9 | 45.5 ± 3.1 | 49.4 ± 6.7 | 36.6 ± 6.6 | 58.3 ± 6.0 | 30.2 ± 2.2 | - | 19.8 ± 4.4 | -10.4 |
| | | 5 | 0.0 | 21.4 ± 1.7 | 48.1 ± 0.9 | 27.9 ± 1.2 | 49.5 ± 8.7 | 28.6 ± 8.6 | 51.7 ± 5.7 | 14.7 ± 2.9 | - | 13.2 ± 2.3 | -1.5 |
| | | | 0.9 | 32.1 ± 1.9 | 45.4 ± 1.9 | 30.6 ± 1.2 | 53.7 ± 5.6 | 37.5 ± 7.0 | 59.8 ± 7.1 | 10.4 ± 0.4 | - | 11.1 ± 1.6 | 0.7 |
| | | | 0.99 | 33.2 ± 1.8 | 49.4 ± 4.1 | 37.1 ± 1.6 | 46.2 ± 6.8 | 38.9 ± 11.8 | 46.1 ± 2.6 | 21.8 ± 2.0 | - | 15.9 ± 5.1 | -5.9 |
| | 1.0 | 1 | 0.0 | 81.1 ± 0.2 | 80.9 ± 0.2 | 81.3 ± 0.2 | 80.7 ± 0.4 | 80.6 ± 0.4 | 81.0 ± 0.4 | 80.9 ± 0.2 | - | 80.4 ± 0.3 | -0.2 |
| | | | 0.9 | 82.0 ± 0.2 | 82.3 ± 0.4 | 82.3 ± 0.4 | 81.7 ± 0.4 | 81.7 ± 0.3 | 81.6 ± 0.3 | 82.0 ± 0.2 | - | 81.6 ± 0.1 | 0.0 |
| | | | 0.99 | 80.6 ± 0.8 | 79.7 ± 0.2 | 80.6 ± 0.6 | 79.5 ± 0.3 | 77.2 ± 2.6 | 78.6 ± 1.4 | 79.9 ± 0.3 | - | 76.5 ± 2.5 | -0.7 |
| | | 3 | 0.0 | 73.8 ± 0.8 | 80.3 ± 0.4 | 77.6 ± 0.2 | 79.0 ± 0.5 | 74.0 ± 0.6 | 76.9 ± 0.4 | 79.5 ± 0.5 | - | 67.0 ± 1.3 | -6.8 |
| | | | 0.9 | 80.7 ± 0.3 | 81.1 ± 0.5 | 77.8 ± 0.3 | 79.4 ± 0.5 | 77.6 ± 0.7 | 78.2 ± 0.3 | 77.2 ± 0.7 | - | 74.1 ± 0.5 | -3.1 |
| | | | 0.99 | 78.4 ± 0.5 | 79.6 ± 0.3 | 75.9 ± 0.4 | 77.6 ± 0.8 | 70.9 ± 3.1 | 77.3 ± 0.4 | 76.1 ± 0.8 | - | 43.2 ± 5.2 | -27.7 |
| | | 5 | 0.0 | 39.8 ± 1.2 | 79.4 ± 0.4 | 71.1 ± 0.7 | 75.5 ± 0.8 | 42.7 ± 2.4 | 70.8 ± 2.5 | 36.1 ± 11.0 | - | 23.6 ± 9.3 | -12.5 |
| | | | 0.9 | 63.2 ± 1.1 | 78.6 ± 0.5 | 61.5 ± 0.8 | 75.7 ± 0.6 | 74.4 ± 0.6 | 75.1 ± 0.6 | 48.7 ± 9.2 | - | 38.4 ± 14.9 | -10.3 |
| | | | 0.99 | 58.9 ± 4.3 | 75.5 ± 0.8 | 51.7 ± 3.0 | 73.4 ± 0.9 | 65.1 ± 6.9 | 73.1 ± 1.7 | 51.8 ± 9.2 | - | 28.5 ± 3.0 | -23.2 |
| | 10.0 | 1 | 0.0 | 82.4 ± 0.2 | 82.3 ± 0.1 | 82.6 ± 0.2 | 82.3 ± 0.4 | 81.8 ± 0.4 | 82.1 ± 0.2 | 82.3 ± 0.2 | - | 81.6 ± 0.3 | -0.2 |
| | | | 0.9 | 83.1 ± 0.3 | 83.0 ± 0.2 | 83.4 ± 0.3 | 82.8 ± 0.3 | 82.8 ± 0.1 | 82.5 ± 0.3 | 82.8 ± 0.6 | - | 82.9 ± 0.2 | 0.4 |
| | | | 0.99 | 81.9 ± 0.4 | 82.0 ± 0.2 | 82.2 ± 0.5 | 81.6 ± 0.4 | 81.6 ± 0.2 | 81.6 ± 0.2 | 81.7 ± 0.2 | - | 81.0 ± 0.3 | -0.3 |
| | | 3 | 0.0 | 75.7 ± 0.4 | 81.7 ± 0.3 | 82.6 ± 0.3 | 80.2 ± 0.5 | 76.2 ± 0.7 | 79.0 ± 0.3 | 80.2 ± 0.5 | - | 71.3 ± 0.7 | -4.4 |
| | | | 0.9 | 82.4 ± 0.4 | 82.3 ± 0.5 | 83.4 ± 0.4 | 80.8 ± 0.2 | 78.1 ± 0.2 | 79.2 ± 0.5 | 80.8 ± 0.9 | - | 80.2 ± 0.3 | 2.1 |
| | | | 0.99 | 81.3 ± 0.4 | 80.6 ± 0.5 | 82.2 ± 0.4 | 80.2 ± 0.4 | 77.4 ± 0.2 | 78.2 ± 0.4 | 78.3 ± 0.2 | - | 75.1 ± 0.7 | -2.3 |
| | | 5 | 0.0 | 47.9 ± 1.6 | 80.7 ± 0.2 | 82.5 ± 0.4 | 77.2 ± 0.7 | 61.1 ± 1.4 | 64.8 ± 1.3 | 61.3 ± 1.4 | - | 49.5 ± 1.2 | 1.6 |
| | | | 0.9 | 74.4 ± 0.7 | 80.7 ± 0.1 | 83.4 ± 0.2 | 77.6 ± 0.6 | 70.7 ± 0.4 | 73.8 ± 0.9 | 69.0 ± 1.0 | - | 74.2 ± 1.1 | 5.2 |
| | | | 0.99 | 76.4 ± 1.0 | 78.7 ± 0.5 | 82.2 ± 0.4 | 76.2 ± 0.4 | 69.7 ± 0.6 | 74.5 ± 0.5 | 68.6 ± 0.4 | - | 64.8 ± 2.4 | -3.8 |
| GM | 0.1 | 1 | 0.0 | 65.4 ± 0.2 | 65.4 ± 1.0 | 58.4 ± 9.5 | 65.1 ± 0.6 | 64.3 ± 0.6 | 65.1 ± 0.5 | 63.7 ± 1.6 | - | 53.4 ± 3.7 | -5.0 |
| | | | 0.9 | 66.6 ± 0.6 | 66.2 ± 0.4 | 56.6 ± 8.6 | 66.0 ± 0.6 | 65.3 ± 0.3 | 66.2 ± 1.0 | 65.2 ± 0.6 | - | 22.8 ± 11.0 | -33.8 |
| | | | 0.99 | 64.7 ± 1.2 | 63.0 ± 3.3 | 55.8 ± 0.9 | 63.6 ± 1.4 | 63.2 ± 1.2 | 64.2 ± 2.2 | 60.9 ± 2.9 | - | 32.6 ± 5.0 | -23.2 |
| | | 3 | 0.0 | 57.1 ± 1.5 | 51.8 ± 0.7 | 38.8 ± 2.3 | 55.9 ± 5.6 | 43.1 ± 6.6 | 57.2 ± 6.1 | 21.1 ± 2.7 | - | 16.2 ± 3.6 | -4.9 |
| | | | 0.9 | 61.9 ± 1.6 | 59.1 ± 1.7 | 40.1 ± 2.2 | 58.8 ± 5.9 | 54.2 ± 11.3 | 65.3 ± 2.5 | 29.5 ± 3.5 | - | 12.9 ± 2.5 | -16.6 |
| | | | 0.99 | 52.4 ± 3.7 | 56.4 ± 4.9 | 43.8 ± 0.9 | 55.4 ± 2.9 | 37.6 ± 11.6 | 60.4 ± 10.6 | 30.7 ± 2.5 | - | 20.7 ± 4.5 | -10.0 |
| | | 5 | 0.0 | 22.7 ± 1.3 | 51.8 ± 4.8 | 27.3 ± 1.1 | 51.5 ± 3.6 | 31.2 ± 10.4 | 53.2 ± 4.6 | 15.3 ± 1.9 | - | 12.9 ± 3.8 | -2.4 |
| | | | 0.9 | 31.9 ± 1.4 | 49.0 ± 0.8 | 29.4 ± 1.0 | 52.7 ± 3.0 | 36.9 ± 5.0 | 60.8 ± 3.0 | 10.4 ± 0.3 | - | 10.8 ± 0.6 | 0.4 |
| | | | 0.99 | 32.5 ± 1.0 | 44.9 ± 2.5 | 36.1 ± 3.5 | 47.2 ± 7.5 | 35.8 ± 7.0 | 42.9 ± 4.6 | 21.3 ± 3.5 | - | 19.4 ± 4.2 | -1.9 |
| | 1.0 | 1 | 0.0 | 81.1 ± 0.3 | 81.1 ± 0.4 | 81.2 ± 0.3 | 80.8 ± 0.3 | 80.8 ± 0.1 | 80.9 ± 0.2 | 81.1 ± 0.3 | - | 80.7 ± 0.3 | -0.1 |
| | | | 0.9 | 82.2 ± 0.4 | 82.1 ± 0.3 | 82.2 ± 0.3 | 81.8 ± 0.3 | 81.8 ± 0.3 | 81.8 ± 0.5 | 82.0 ± 0.4 | - | 81.8 ± 0.1 | 0.0 |
| | | | 0.99 | 81.0 ± 0.4 | 80.7 ± 0.3 | 81.0 ± 0.3 | 81.0 ± 0.1 | 79.2 ± 1.0 | 80.3 ± 0.1 | 80.8 ± 0.2 | - | 78.2 ± 0.9 | -1.0 |
| | | 3 | 0.0 | 73.8 ± 0.6 | 80.4 ± 0.4 | 77.6 ± 0.3 | 78.9 ± 0.6 | 74.0 ± 1.3 | 76.9 ± 0.5 | 79.5 ± 0.3 | - | 68.1 ± 0.4 | -5.7 |
| | | | 0.9 | 80.9 ± 0.3 | 81.0 ± 0.4 | 77.8 ± 0.5 | 79.8 ± 0.5 | 78.0 ± 0.3 | 78.8 ± 0.4 | 77.6 ± 0.6 | - | 74.4 ± 1.1 | -3.2 |
| | | | 0.99 | 78.7 ± 0.5 | 79.8 ± 0.3 | 76.4 ± 0.4 | 78.3 ± 0.5 | 71.7 ± 4.0 | 77.1 ± 1.1 | 76.0 ± 1.1 | - | 45.5 ± 3.3 | -26.2 |
| | | 5 | 0.0 | 39.4 ± 1.1 | 79.5 ± 0.4 | 71.0 ± 1.0 | 75.3 ± 0.6 | 41.8 ± 4.1 | 71.0 ± 3.1 | 28.6 ± 15.3 | - | 28.0 ± 12.3 | -0.6 |
| | | | 0.9 | 62.9 ± 0.8 | 79.1 ± 0.2 | 59.8 ± 3.1 | 76.0 ± 0.5 | 74.5 ± 0.3 | 75.3 ± 0.3 | 48.0 ± 8.3 | - | 50.3 ± 7.6 | 2.3 |
| | | | 0.99 | 60.7 ± 2.0 | 76.9 ± 0.4 | 52.9 ± 2.3 | 74.0 ± 1.0 | 68.5 ± 4.1 | 74.3 ± 1.2 | 55.6 ± 4.8 | - | 28.5 ± 4.1 | -24.4 |
| | 10.0 | 1 | 0.0 | 82.3 ± 0.3 | 82.2 ± 0.4 | 82.7 ± 0.3 | 82.1 ± 0.3 | 81.8 ± 0.3 | 82.1 ± 0.3 | 82.2 ± 0.2 | - | 81.8 ± 0.5 | 0.0 |
| | | | 0.9 | 83.2 ± 0.4 | 83.1 ± 0.2 | 83.3 ± 0.3 | 82.9 ± 0.4 | 82.9 ± 0.3 | 82.8 ± 0.4 | 82.9 ± 0.4 | - | 83.2 ± 0.3 | 0.4 |
| | | | 0.99 | 82.0 ± 0.4 | 81.9 ± 0.4 | 82.2 ± 0.1 | 81.6 ± 0.5 | 81.6 ± 0.5 | 81.8 ± 0.4 | 82.0 ± 0.3 | - | 81.8 ± 0.3 | 0.2 |
| | | 3 | 0.0 | 75.7 ± 0.2 | 81.6 ± 0.3 | 82.5 ± 0.4 | 80.2 ± 0.6 | 76.1 ± 0.4 | 79.0 ± 0.4 | 80.2 ± 0.2 | - | 71.2 ± 0.5 | -4.5 |
| | | | 0.9 | 82.8 ± 0.4 | 82.3 ± 0.3 | 83.2 ± 0.4 | 81.1 ± 0.2 | 78.3 ± 0.4 | 79.5 ± 0.4 | 80.8 ± 0.5 | - | 80.4 ± 0.3 | 2.1 |
| | | | 0.99 | 81.5 ± 0.3 | 80.9 ± 0.3 | 82.4 ± 0.3 | 79.6 ± 0.5 | 77.9 ± 0.5 | 78.4 ± 0.5 | 79.0 ± 0.5 | - | 75.8 ± 1.0 | -2.1 |

| agg | alpha | f | momentum | ALIE | FOE | LF | Mimic | MinMax | MinSum | SF | No Attack | JUMP | Superiority |
|---|---|---|---|---|---|---|---|---|---|---|---|---|---|
| | | 5 | 0.0 | 48.2 ± 1.3 | 80.7 ± 0.3 | 82.3 ± 0.1 | 77.4 ± 0.4 | 60.8 ± 0.8 | 64.1 ± 2.3 | 62.5 ± 1.4 | - | 51.1 ± 1.5 | 2.9 |
| | | | 0.9 | 74.3 ± 0.5 | 81.3 ± 0.3 | 83.3 ± 0.3 | 78.2 ± 0.3 | 71.7 ± 1.4 | 74.0 ± 0.6 | 68.4 ± 1.3 | - | 74.5 ± 0.6 | 6.1 |
| | | | 0.99 | 76.5 ± 0.7 | 79.4 ± 0.4 | 82.2 ± 0.2 | 76.5 ± 0.6 | 72.7 ± 0.7 | 75.2 ± 0.6 | 70.0 ± 0.8 | - | 65.4 ± 2.3 | -4.6 |
| Krum | 0.1 | 1 | 0.0 | 65.4 ± 0.6 | 65.4 ± 0.7 | 52.4 ± 1.0 | 65.3 ± 0.6 | 64.3 ± 0.6 | 64.9 ± 0.5 | 62.8 ± 2.1 | - | 44.1 ± 20.9 | -8.3 |
| | | | 0.9 | 66.7 ± 0.5 | 66.3 ± 0.7 | 54.8 ± 7.5 | 65.4 ± 0.4 | 64.8 ± 0.7 | 65.8 ± 0.6 | 61.6 ± 5.6 | - | 23.5 ± 9.9 | -31.3 |
| | | | 0.99 | 55.5 ± 9.6 | 58.4 ± 7.6 | 48.6 ± 5.7 | 60.4 ± 8.0 | 61.6 ± 2.4 | 64.3 ± 1.6 | 60.5 ± 3.2 | - | 28.1 ± 2.4 | -20.5 |
| | | 3 | 0.0 | 56.7 ± 0.7 | 51.6 ± 0.5 | 40.5 ± 2.0 | 57.1 ± 6.9 | 42.5 ± 4.0 | 60.1 ± 2.5 | 22.3 ± 6.3 | - | 13.4 ± 2.5 | -8.9 |
| | | | 0.9 | 63.3 ± 0.7 | 60.2 ± 1.0 | 40.2 ± 2.0 | 61.8 ± 2.6 | 44.7 ± 10.6 | 65.1 ± 3.0 | 25.4 ± 3.9 | - | 13.0 ± 2.2 | -12.4 |
| | | | 0.99 | 46.6 ± 5.4 | 44.9 ± 6.7 | 40.4 ± 1.2 | 49.8 ± 6.8 | 36.2 ± 6.0 | 56.1 ± 15.3 | 29.7 ± 5.1 | - | 22.5 ± 4.2 | -7.2 |
| | | 5 | 0.0 | 20.6 ± 2.9 | 47.5 ± 0.7 | 28.6 ± 2.2 | 49.7 ± 1.6 | 30.8 ± 11.1 | 52.3 ± 4.0 | 13.3 ± 1.5 | - | 13.0 ± 2.8 | -0.3 |
| | | | 0.9 | 31.5 ± 1.2 | 48.3 ± 5.9 | 30.1 ± 1.5 | 51.0 ± 3.5 | 32.5 ± 5.2 | 59.4 ± 4.4 | 11.3 ± 2.4 | - | 11.2 ± 1.8 | -0.1 |
| | | | 0.99 | 32.8 ± 1.4 | 43.3 ± 7.3 | 34.2 ± 3.3 | 40.8 ± 11.6 | 33.4 ± 10.3 | 43.6 ± 9.6 | 21.5 ± 3.6 | - | 16.0 ± 3.6 | -5.5 |
| | 1.0 | 1 | 0.0 | 81.1 ± 0.1 | 81.0 ± 0.1 | 81.0 ± 0.3 | 80.9 ± 0.5 | 80.8 ± 0.4 | 80.7 ± 0.2 | 80.8 ± 0.3 | - | 80.5 ± 0.3 | -0.2 |
| | | | 0.9 | 82.1 ± 0.4 | 82.0 ± 0.5 | 82.3 ± 0.3 | 81.6 ± 0.3 | 81.6 ± 0.4 | 81.8 ± 0.4 | 81.7 ± 0.2 | - | 81.6 ± 0.3 | 0.0 |
| | | | 0.99 | 79.3 ± 1.9 | 78.7 ± 2.7 | 78.4 ± 0.8 | 78.1 ± 1.1 | 76.9 ± 3.9 | 75.9 ± 2.2 | 78.8 ± 1.4 | - | 72.3 ± 1.8 | -3.6 |
| | | 3 | 0.0 | 73.7 ± 0.2 | 80.4 ± 0.5 | 77.5 ± 0.3 | 78.8 ± 0.3 | 74.6 ± 0.4 | 76.6 ± 0.2 | 79.1 ± 0.4 | - | 62.3 ± 2.7 | -11.4 |
| | | | 0.9 | 80.6 ± 0.4 | 80.8 ± 0.4 | 77.6 ± 0.3 | 79.5 ± 0.6 | 77.7 ± 0.7 | 78.2 ± 0.4 | 76.9 ± 0.6 | - | 73.0 ± 1.0 | -3.9 |
| | | | 0.99 | 76.8 ± 1.6 | 77.6 ± 2.8 | 71.7 ± 1.9 | 75.3 ± 1.7 | 65.4 ± 10.8 | 70.8 ± 3.0 | 70.7 ± 3.7 | - | 43.1 ± 4.4 | -22.3 |
| | | 5 | 0.0 | 39.6 ± 1.1 | 79.3 ± 0.3 | 70.5 ± 0.7 | 75.5 ± 0.4 | 40.8 ± 1.6 | 71.2 ± 2.3 | 31.4 ± 12.3 | - | 25.2 ± 7.3 | -6.2 |
| | | | 0.9 | 61.1 ± 3.7 | 78.5 ± 0.5 | 61.2 ± 0.8 | 75.6 ± 0.8 | 74.2 ± 0.6 | 75.1 ± 0.8 | 45.0 ± 3.5 | - | 44.5 ± 6.0 | -0.5 |
| | | | 0.99 | 58.9 ± 3.4 | 75.5 ± 1.4 | 46.4 ± 1.4 | 72.1 ± 1.5 | 61.9 ± 7.2 | 72.5 ± 1.1 | 43.0 ± 8.7 | - | 25.4 ± 0.4 | -17.6 |
| | 10.0 | 1 | 0.0 | 82.5 ± 0.4 | 82.1 ± 0.3 | 82.6 ± 0.5 | 82.1 ± 0.4 | 81.8 ± 0.4 | 82.0 ± 0.2 | 82.3 ± 0.3 | - | 81.7 ± 0.4 | -0.1 |
| | | | 0.9 | 83.1 ± 0.4 | 83.0 ± 0.6 | 83.4 ± 0.4 | 82.7 ± 0.5 | 82.5 ± 0.2 | 82.6 ± 0.3 | 82.8 ± 0.2 | - | 83.0 ± 0.1 | 0.5 |
| | | | 0.99 | 81.9 ± 0.4 | 81.6 ± 0.3 | 82.4 ± 0.1 | 81.7 ± 0.5 | 81.5 ± 0.2 | 81.3 ± 0.4 | 81.6 ± 0.4 | - | 81.0 ± 0.3 | -0.3 |
| | | 3 | 0.0 | 75.6 ± 0.4 | 81.5 ± 0.3 | 82.3 ± 0.6 | 80.2 ± 0.5 | 75.3 ± 0.6 | 78.8 ± 0.4 | 80.0 ± 0.6 | - | 69.7 ± 0.6 | -5.6 |
| | | | 0.9 | 82.3 ± 0.3 | 81.9 ± 0.4 | 83.3 ± 0.4 | 80.9 ± 0.5 | 77.6 ± 0.3 | 78.7 ± 0.2 | 80.0 ± 0.2 | - | 79.3 ± 0.2 | 1.7 |
| | | | 0.99 | 80.7 ± 0.4 | 79.9 ± 0.2 | 82.2 ± 0.1 | 78.4 ± 0.4 | 74.7 ± 1.2 | 77.0 ± 0.9 | 76.6 ± 0.3 | - | 72.7 ± 0.6 | -2.0 |
| | | 5 | 0.0 | 47.4 ± 1.3 | 80.5 ± 0.3 | 82.6 ± 0.3 | 77.1 ± 0.6 | 58.8 ± 1.4 | 65.6 ± 1.7 | 60.7 ± 1.3 | - | 47.4 ± 0.8 | 0.0 |
| | | | 0.9 | 74.3 ± 0.5 | 80.5 ± 0.3 | 83.3 ± 0.2 | 77.2 ± 0.8 | 70.4 ± 1.4 | 73.4 ± 0.6 | 67.3 ± 0.6 | - | 73.5 ± 1.0 | 6.2 |
| | | | 0.99 | 75.5 ± 0.4 | 77.6 ± 0.7 | 82.2 ± 0.2 | 74.7 ± 0.7 | 70.4 ± 0.9 | 73.3 ± 0.5 | 63.1 ± 0.5 | - | 61.7 ± 0.8 | -1.4 |
| TM | 0.1 | 1 | 0.0 | 65.4 ± 0.3 | 65.2 ± 0.7 | 63.7 ± 7.3 | 65.4 ± 0.6 | 64.4 ± 0.7 | 65.3 ± 0.6 | 64.1 ± 1.2 | - | 56.6 ± 2.6 | -7.1 |
| | | | 0.9 | 66.0 ± 0.9 | 65.4 ± 0.6 | 50.7 ± 0.9 | 66.0 ± 0.4 | 65.0 ± 0.9 | 66.4 ± 0.3 | 65.3 ± 0.8 | - | 31.6 ± 6.3 | -19.1 |
| | | | 0.99 | 65.4 ± 0.6 | 65.0 ± 0.7 | 61.5 ± 3.7 | 63.4 ± 5.0 | 62.1 ± 2.8 | 64.8 ± 1.5 | 64.4 ± 0.9 | - | 37.6 ± 13.8 | -23.9 |
| | | 3 | 0.0 | 57.6 ± 1.1 | 51.9 ± 0.5 | 38.8 ± 0.9 | 54.3 ± 3.2 | 41.7 ± 4.4 | 57.2 ± 6.1 | 19.3 ± 4.5 | - | 17.5 ± 2.2 | -1.8 |
| | | | 0.9 | 62.2 ± 0.7 | 57.2 ± 0.6 | 39.6 ± 1.5 | 59.8 ± 4.7 | 48.2 ± 11.3 | 63.3 ± 3.2 | 30.6 ± 5.8 | - | 12.6 ± 2.0 | -18.0 |
| | | | 0.99 | 57.2 ± 2.3 | 54.0 ± 4.8 | 42.6 ± 1.2 | 52.9 ± 1.9 | 38.2 ± 11.3 | 57.0 ± 4.8 | 30.4 ± 2.5 | - | 23.9 ± 2.6 | -6.5 |
| | | 5 | 0.0 | 21.8 ± 2.3 | 48.7 ± 6.8 | 28.4 ± 2.0 | 52.6 ± 5.7 | 24.8 ± 3.4 | 53.0 ± 4.5 | 14.1 ± 1.6 | - | 13.4 ± 2.8 | -0.7 |
| | | | 0.9 | 32.5 ± 2.8 | 46.9 ± 0.8 | 30.2 ± 1.7 | 53.9 ± 4.9 | 35.5 ± 7.6 | 61.8 ± 2.4 | 10.6 ± 0.7 | - | 11.5 ± 1.2 | 0.9 |
| | | | 0.99 | 32.9 ± 1.8 | 46.3 ± 3.1 | 33.9 ± 1.4 | 47.9 ± 5.7 | 36.1 ± 6.2 | 45.1 ± 9.1 | 23.9 ± 1.6 | - | 18.8 ± 3.8 | -5.1 |
| | 1.0 | 1 | 0.0 | 80.8 ± 0.2 | 81.0 ± 0.3 | 81.1 ± 0.3 | 80.8 ± 0.3 | 80.6 ± 0.4 | 80.9 ± 0.3 | 81.1 ± 0.4 | - | 80.5 ± 0.2 | -0.1 |
| | | | 0.9 | 82.2 ± 0.5 | 82.3 ± 0.3 | 82.3 ± 0.4 | 82.1 ± 0.3 | 81.7 ± 0.2 | 81.9 ± 0.5 | 82.2 ± 0.5 | - | 82.0 ± 0.4 | 0.3 |
| | | | 0.99 | 81.0 ± 0.3 | 80.7 ± 0.4 | 81.1 ± 0.4 | 80.8 ± 0.3 | 79.5 ± 1.0 | 80.3 ± 0.3 | 81.1 ± 0.2 | - | 78.9 ± 0.5 | -0.6 |
| | | 3 | 0.0 | 73.8 ± 0.5 | 80.3 ± 0.5 | 77.7 ± 0.4 | 78.9 ± 0.6 | 74.7 ± 1.0 | 77.2 ± 0.6 | 79.5 ± 0.1 | - | 68.6 ± 1.5 | -5.2 |
| | | | 0.9 | 80.9 ± 0.1 | 81.2 ± 0.2 | 78.1 ± 0.3 | 79.6 ± 0.4 | 77.9 ± 0.5 | 78.7 ± 0.4 | 77.8 ± 0.7 | - | 75.5 ± 0.7 | -2.3 |
| | | | 0.99 | 78.5 ± 0.3 | 79.7 ± 0.5 | 76.4 ± 0.6 | 78.2 ± 0.6 | 73.6 ± 3.1 | 77.6 ± 0.5 | 76.6 ± 0.6 | - | 50.2 ± 3.1 | -23.4 |
| | | 5 | 0.0 | 40.0 ± 1.2 | 79.5 ± 0.2 | 70.8 ± 0.6 | 75.2 ± 0.6 | 42.3 ± 2.8 | 71.1 ± 2.9 | 36.1 ± 4.2 | - | 31.1 ± 12.6 | -5.0 |
| | | | 0.9 | 62.8 ± 0.8 | 78.6 ± 0.2 | 61.2 ± 0.9 | 76.0 ± 0.7 | 74.4 ± 0.5 | 75.5 ± 0.6 | 49.6 ± 9.5 | - | 58.8 ± 1.8 | 9.2 |
| | | | 0.99 | 60.5 ± 3.9 | 76.6 ± 0.6 | 51.3 ± 1.8 | 74.0 ± 0.7 | 68.4 ± 3.0 | 73.6 ± 1.5 | 59.4 ± 4.6 | - | 31.9 ± 5.6 | -19.4 |
| | 10.0 | 1 | 0.0 | 82.4 ± 0.2 | 82.2 ± 0.2 | 82.5 ± 0.4 | 82.0 ± 0.2 | 82.0 ± 0.4 | 82.1 ± 0.3 | 82.4 ± 0.3 | - | 81.7 ± 0.2 | -0.3 |
| | | | 0.9 | 83.2 ± 0.5 | 83.0 ± 0.2 | 83.5 ± 0.3 | 82.9 ± 0.3 | 82.7 ± 0.3 | 83.0 ± 0.3 | 82.9 ± 0.4 | - | 83.1 ± 0.1 | 0.4 |
| | | | 0.99 | 82.2 ± 0.2 | 82.0 ± 0.5 | 82.0 ± 0.5 | 82.3 ± 0.3 | 81.9 ± 0.2 | 81.6 ± 0.4 | 82.2 ± 0.1 | - | 81.7 ± 0.3 | 0.1 |
| | | 3 | 0.0 | 75.8 ± 0.3 | 81.7 ± 0.3 | 82.4 ± 0.4 | 80.2 ± 0.5 | 76.9 ± 0.2 | 78.8 ± 0.7 | 80.6 ± 0.3 | - | 72.0 ± 0.3 | -3.8 |
| | | | 0.9 | 82.4 ± 0.2 | 82.2 ± 0.5 | 83.3 ± 0.2 | 81.1 ± 0.4 | 78.6 ± 0.2 | 79.4 ± 0.5 | 81.0 ± 0.2 | - | 80.3 ± 0.3 | 1.7 |
| | | | 0.99 | 81.4 ± 0.4 | 80.9 ± 0.6 | 82.3 ± 0.1 | 79.4 ± 0.5 | 77.7 ± 0.5 | 78.4 ± 0.7 | 78.8 ± 0.2 | - | 75.6 ± 0.7 | -2.1 |
| | | 5 | 0.0 | 48.4 ± 2.5 | 80.8 ± 0.2 | 82.3 ± 0.4 | 77.3 ± 1.0 | 61.4 ± 1.0 | 65.0 ± 0.9 | 62.0 ± 1.2 | - | 49.7 ± 2.1 | 1.3 |
| | | | 0.9 | 74.4 ± 0.7 | 80.9 ± 0.2 | 83.2 ± 0.4 | 77.6 ± 0.7 | 72.0 ± 1.0 | 73.8 ± 0.5 | 69.7 ± 0.9 | - | 74.7 ± 1.1 | 5.0 |
| | | | 0.99 | 76.5 ± 0.9 | 79.0 ± 0.6 | 82.5 ± 0.5 | 76.5 ± 0.9 | 71.0 ± 1.8 | 74.6 ± 0.6 | 70.5 ± 0.1 | - | 65.7 ± 1.9 | -4.8 |

Table 7: Comparison of JUMP with state-of-the-art attacks on CIFAR-10 under heterogeneity $\alpha \in \{0.1, 1, 10\}$. Under the column "Superiority", we report the difference in accuracy between the best existing attack and JUMP. In green, we emphasize cases where JUMP is significantly more powerful than any other attack. In light green, we show cases where JUMP is comparable to the best among other attacks: JUMP's accuracy is one of the lowest and its confidence interval overlaps with some other attack's. In red, we show cases where JUMP is weaker than some attack.

## C.3 ACCURACY EVOLUTIONS ON MNIST AND CIFAR-10

In this section, we show all accuracy evolutions on MNIST and CIFAR-10 with the same setting as Figure 3 in Section 5. Here, we additionally show the same results on all aggregation rules, heterogeneity levels, and momentum parameters considered. The observations made in Section 5 and Appendix C.2, generally continue to hold, i.e., that JUMP is always a clear winner for small $f$ and is also the most damaging for larger $f$, on par with some attacks, especially during late training stages. Once again, under low data heterogeneity, all defenses perform well as predicted by theory in i.i.d. settings using local momentum (Farhadkhani et al., 2022).

**Results on GM.** The results with the geometric median aggregation are in figures 14, 15, and 16.

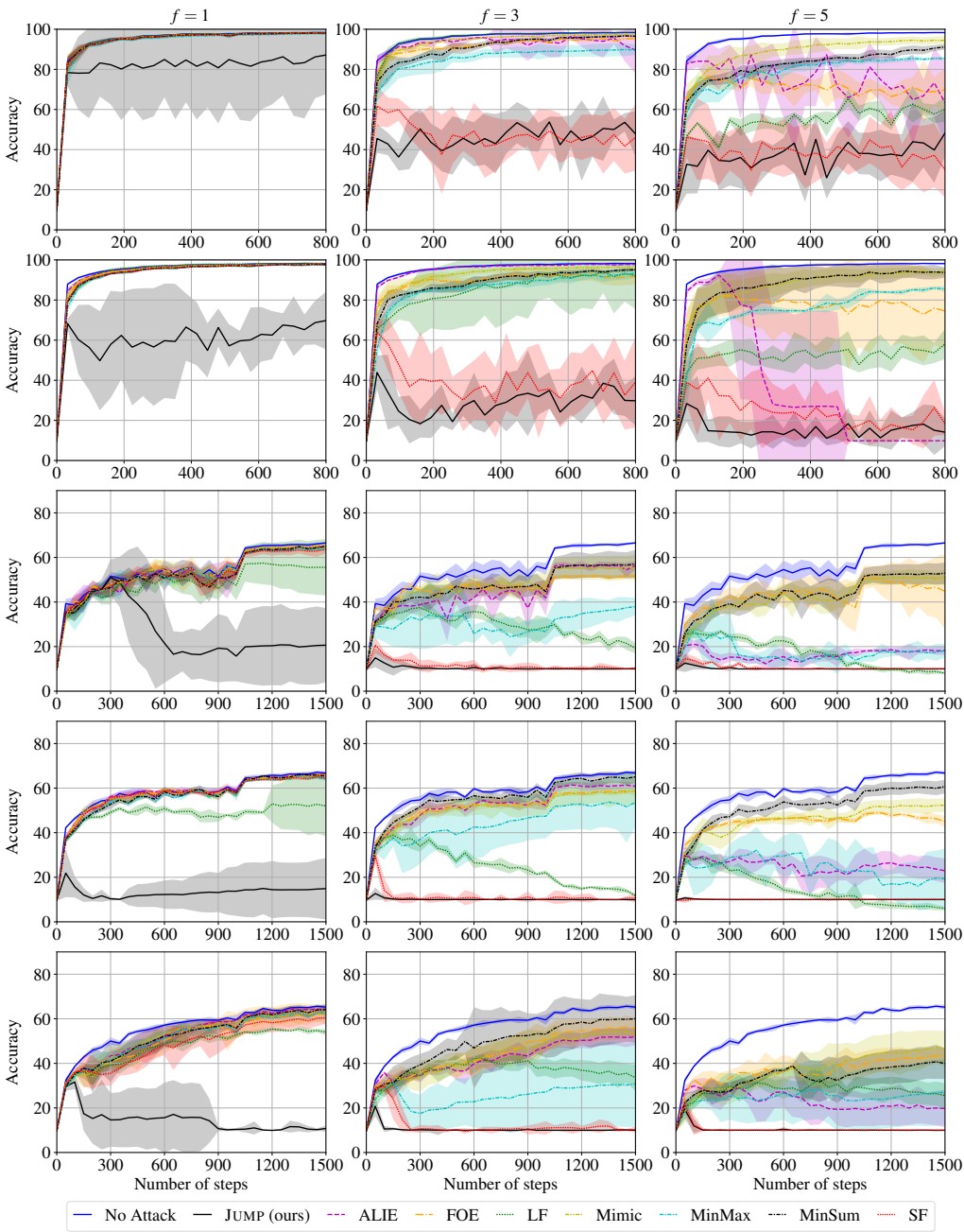

Figure 14: Accuracy evolution of Robust D-SGD with the geometric median aggregation on MNIST (1st and 2nd row with momentum parameter 0.0 and 0.9 respectively) and CIFAR-10 (3rd, 4th, and 5th row with momentum parameter 0.0, 0.9, and 0.99 respectively), with high data heterogeneity ($\alpha = 0.1$), under JUMP and other representative attacks for $f = 1$ (1st column), $f = 3$ (2nd column), and $f = 5$ (3rd column) Byzantine workers. Each experiment is run 5 times and we show the 95% confidence interval.

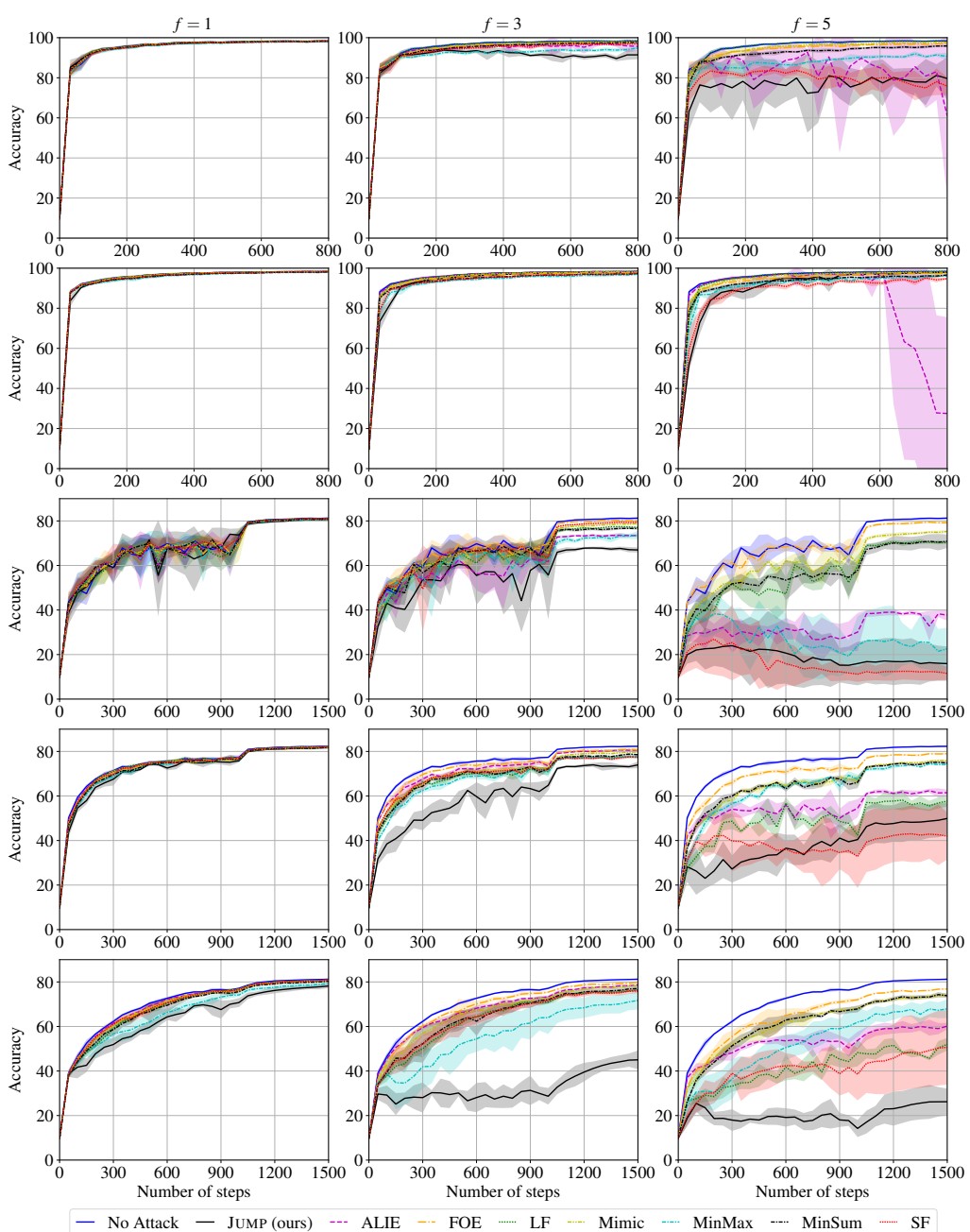

Figure 15: Accuracy evolution of Robust D-SGD with the geometric median aggregation on MNIST (1st and 2nd row with momentum parameter 0.0 and 0.9 respectively) and CIFAR-10 (3rd, 4th, and 5th row with momentum parameter 0.0, 0.9, and 0.99 respectively), with moderate data heterogeneity ($\alpha = 1.0$), under JUMP and other representative attacks for $f = 1$ (1st column), $f = 3$ (2nd column), and $f = 5$ (3rd column) Byzantine workers. Each experiment is run 5 times and we show the 95% confidence interval.

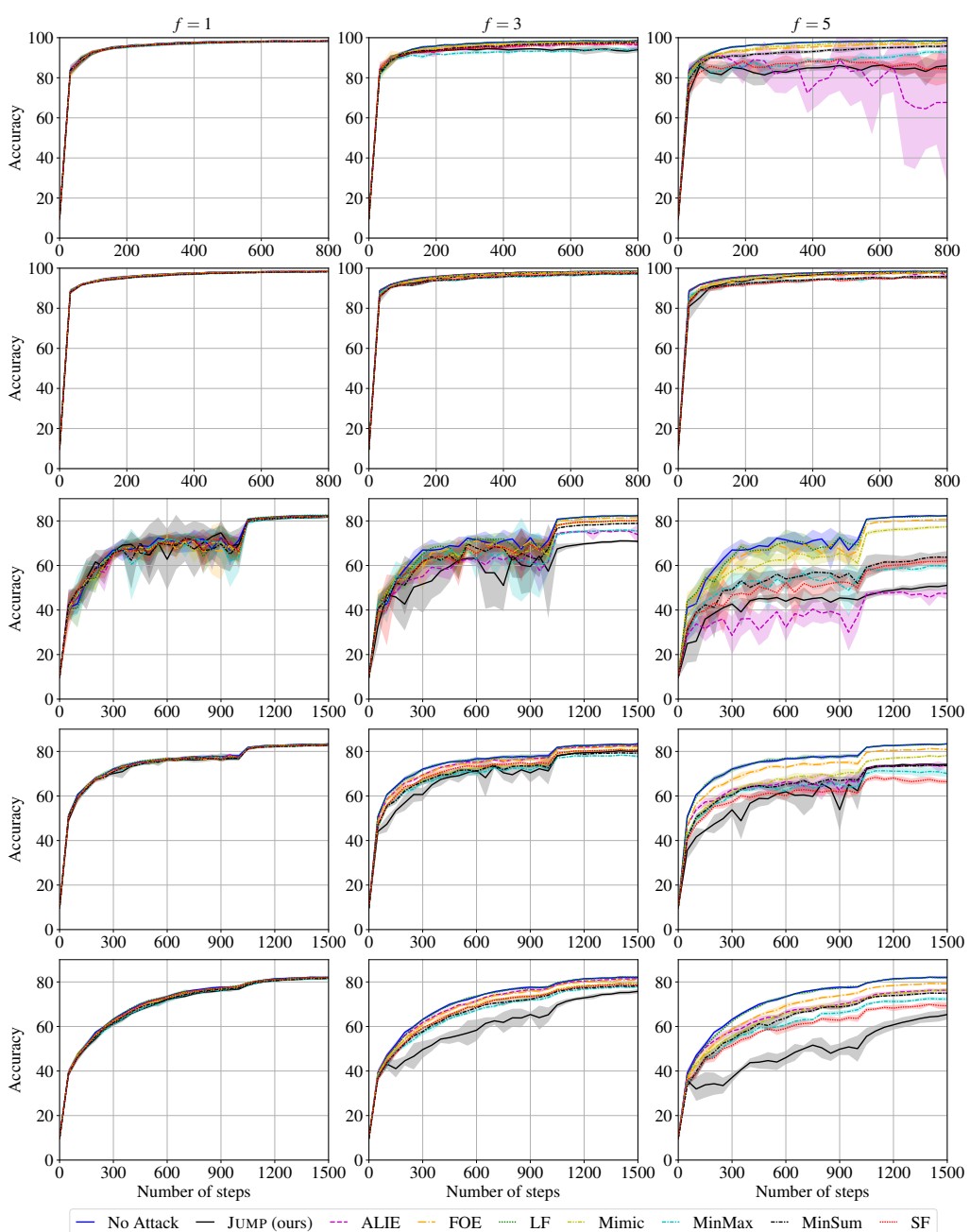

Figure 16: Accuracy evolution of Robust D-SGD with the geometric median aggregation on MNIST (1st and 2nd row with momentum parameter 0.0 and 0.9 respectively) and CIFAR-10 (3rd, 4th, and 5th row with momentum parameter 0.0, 0.9, and 0.99 respectively), with low data heterogeneity ($\alpha = 10.0$), under JUMP and other representative attacks for $f = 1$ (1st column), $f = 3$ (2nd column), and $f = 5$ (3rd column) Byzantine workers. Each experiment is run 5 times and we show the 95% confidence interval.

**Results on CM.** The results with the coordinate-wise median aggregation are in figures 17, 18, and 19.

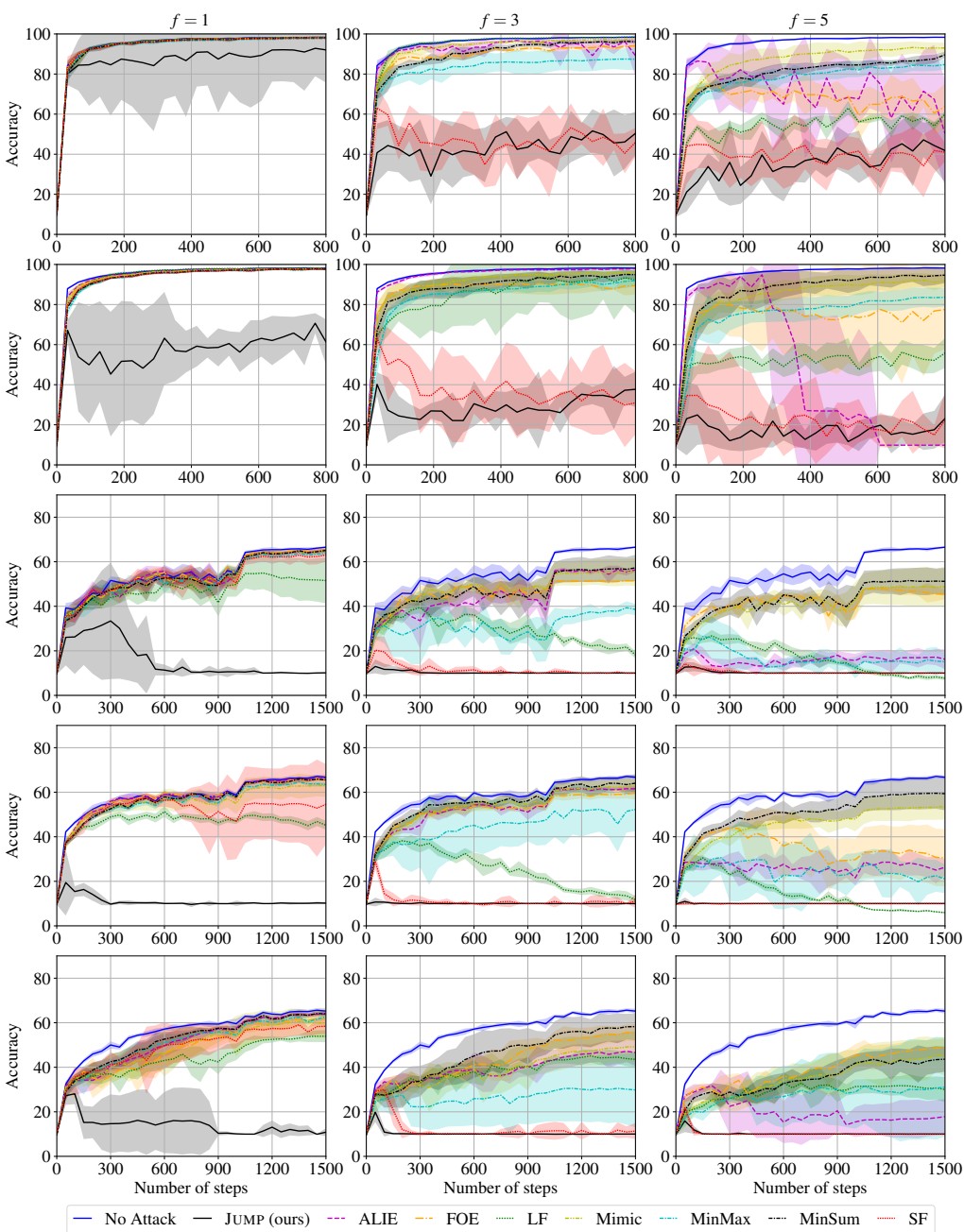

Figure 17: Accuracy evolution of Robust D-SGD with the coordinate-wise median aggregation on MNIST (1st and 2nd row with momentum parameter 0.0 and 0.9 respectively) and CIFAR-10 (3rd, 4th, and 5th row with momentum parameter 0.0, 0.9, and 0.99 respectively), with high data heterogeneity ($\alpha = 0.1$), under JUMP and other representative attacks for $f = 1$ (1st column), $f = 3$ (2nd column), and $f = 5$ (3rd column) Byzantine workers. Each experiment is run 5 times and we show the 95% confidence interval.

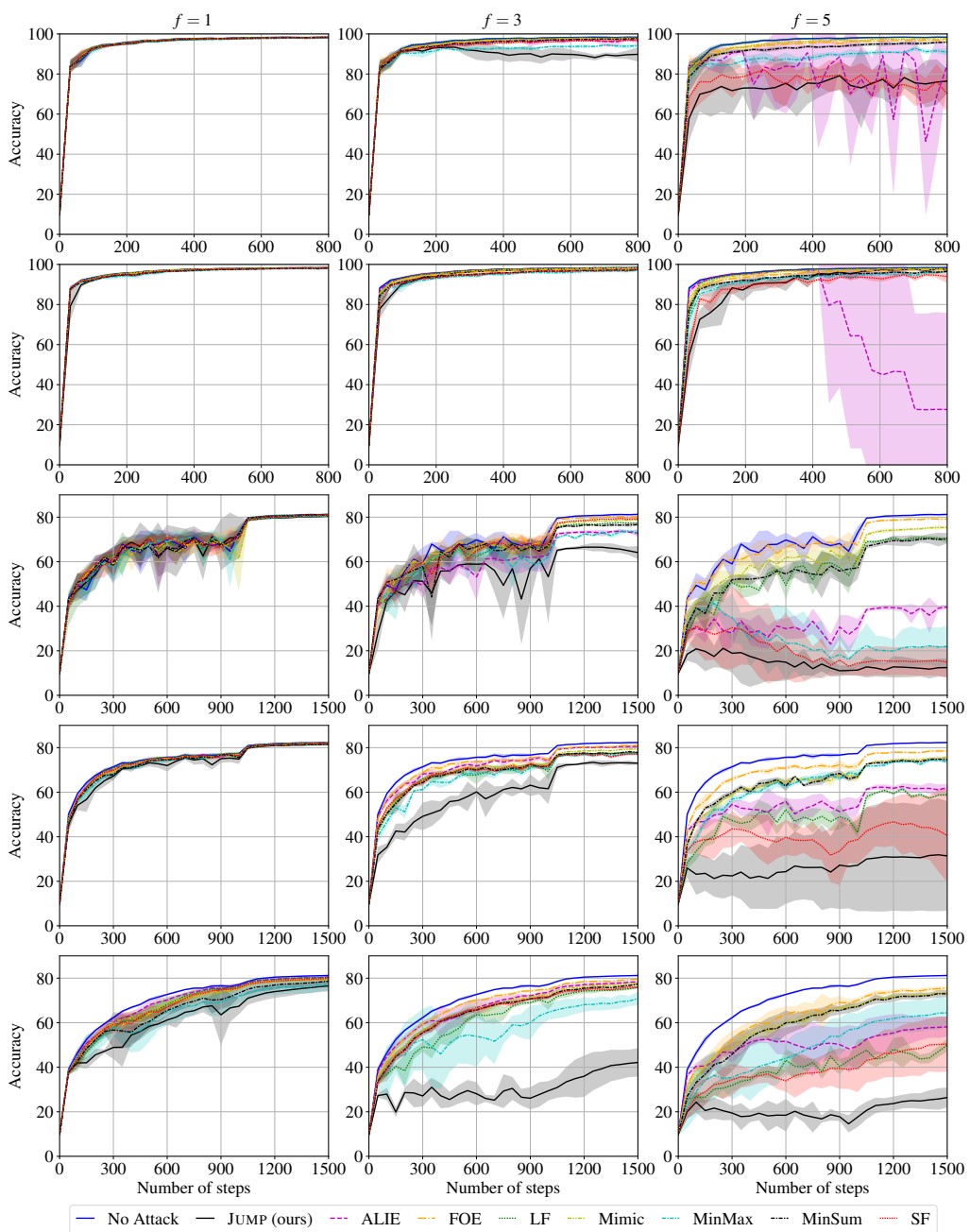

Figure 18: Accuracy evolution of Robust D-SGD with the coordinate-wise median aggregation on MNIST (1st and 2nd row with momentum parameter 0.0 and 0.9 respectively) and CIFAR-10 (3rd, 4th, and 5th row with momentum parameter 0.0, 0.9, and 0.99 respectively), with moderate data heterogeneity ($\alpha = 1.0$), under JUMP and other representative attacks for $f = 1$ (1st column), $f = 3$ (2nd column), and $f = 5$ (3rd column) Byzantine workers. Each experiment is run 5 times and we show the 95% confidence interval.

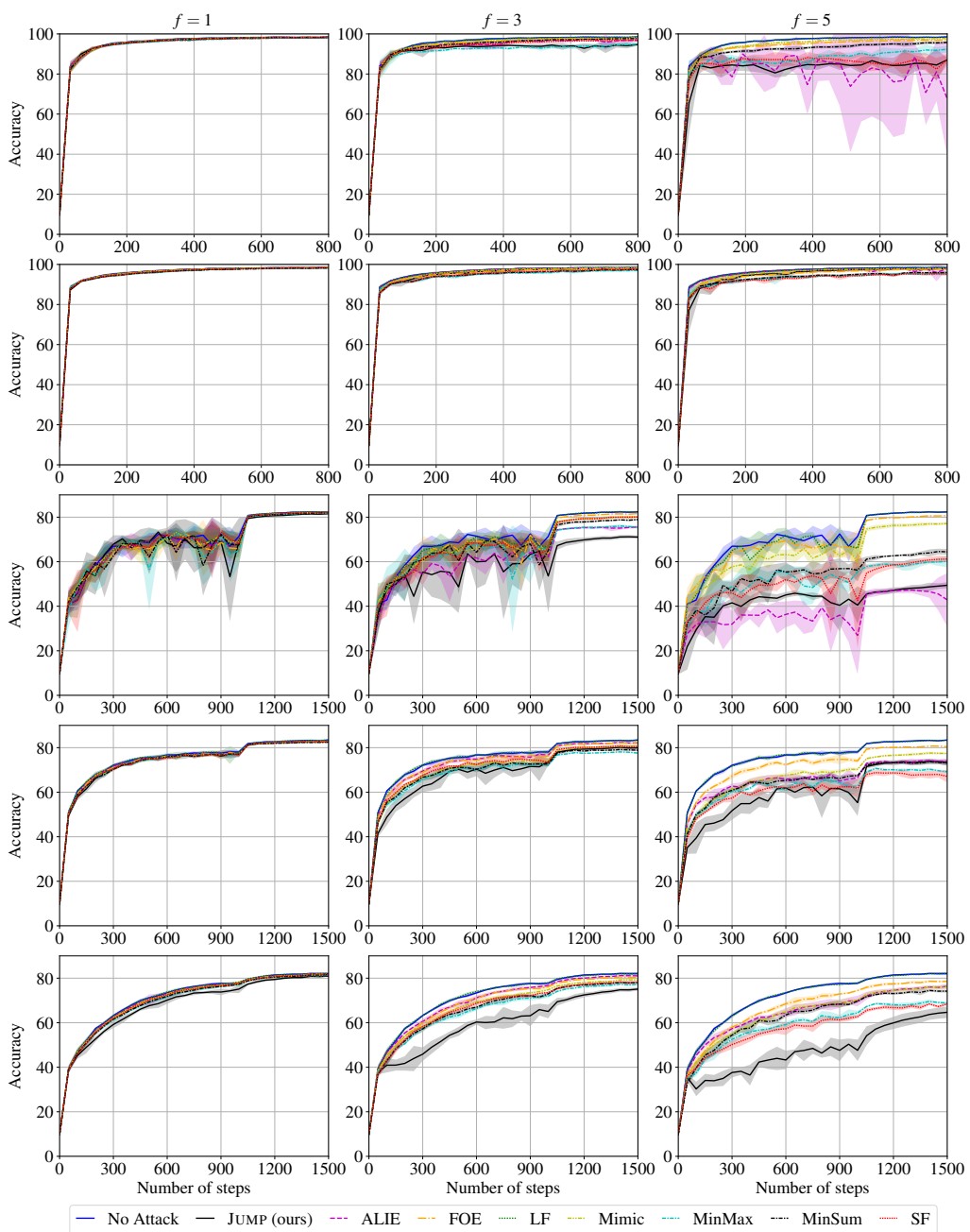

Figure 19: Accuracy evolution of Robust D-SGD with the coordinate-wise median aggregation on MNIST (1st and 2nd row with momentum parameter $0.0$ and $0.9$ respectively) and CIFAR-10 (3rd, 4th, and 5th row with momentum parameter $0.0$, $0.9$, and $0.99$ respectively), with low data heterogeneity ($\alpha = 10.0$), under JUMP and other representative attacks for $f = 1$ (1st column), $f = 3$ (2nd column), and $f = 5$ (3rd column) Byzantine workers. Each experiment is run 5 times and we show the 95% confidence interval.

**Results on TM.** The results with the trimmed mean aggregation are in figures 20, 21, and 22.

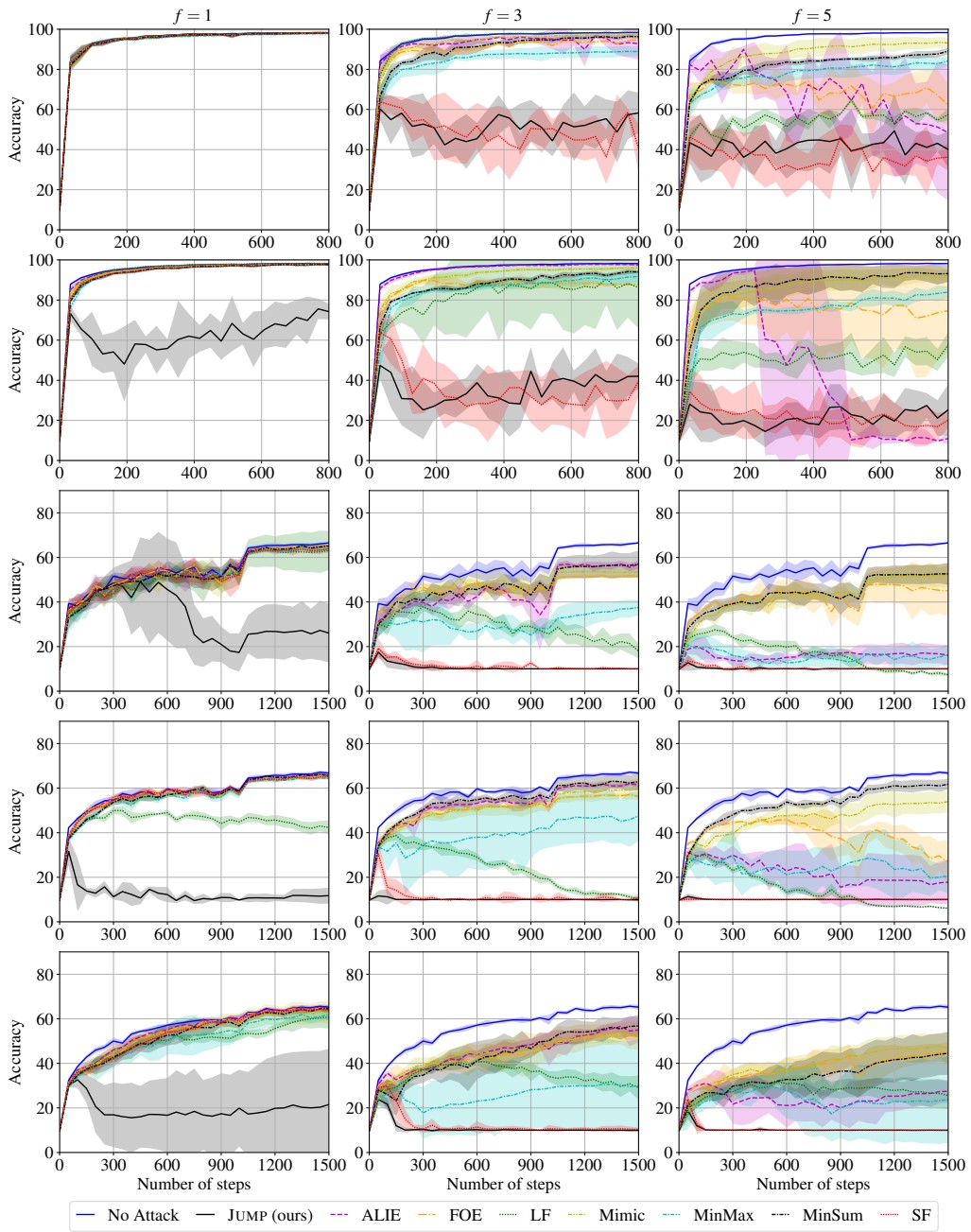

Figure 20: Accuracy evolution of Robust D-SGD with the trimmed mean aggregation on MNIST (1st and 2nd row with momentum parameter 0.0 and 0.9 respectively) and CIFAR-10 (3rd, 4th, and 5th row with momentum parameter 0.0, 0.9, and 0.99 respectively), with high data heterogeneity ($\alpha = 0.1$), under JUMP and other representative attacks for $f = 1$ (1st column), $f = 3$ (2nd column), and $f = 5$ (3rd column) Byzantine workers. Each experiment is run 5 times and we show the 95% confidence interval.

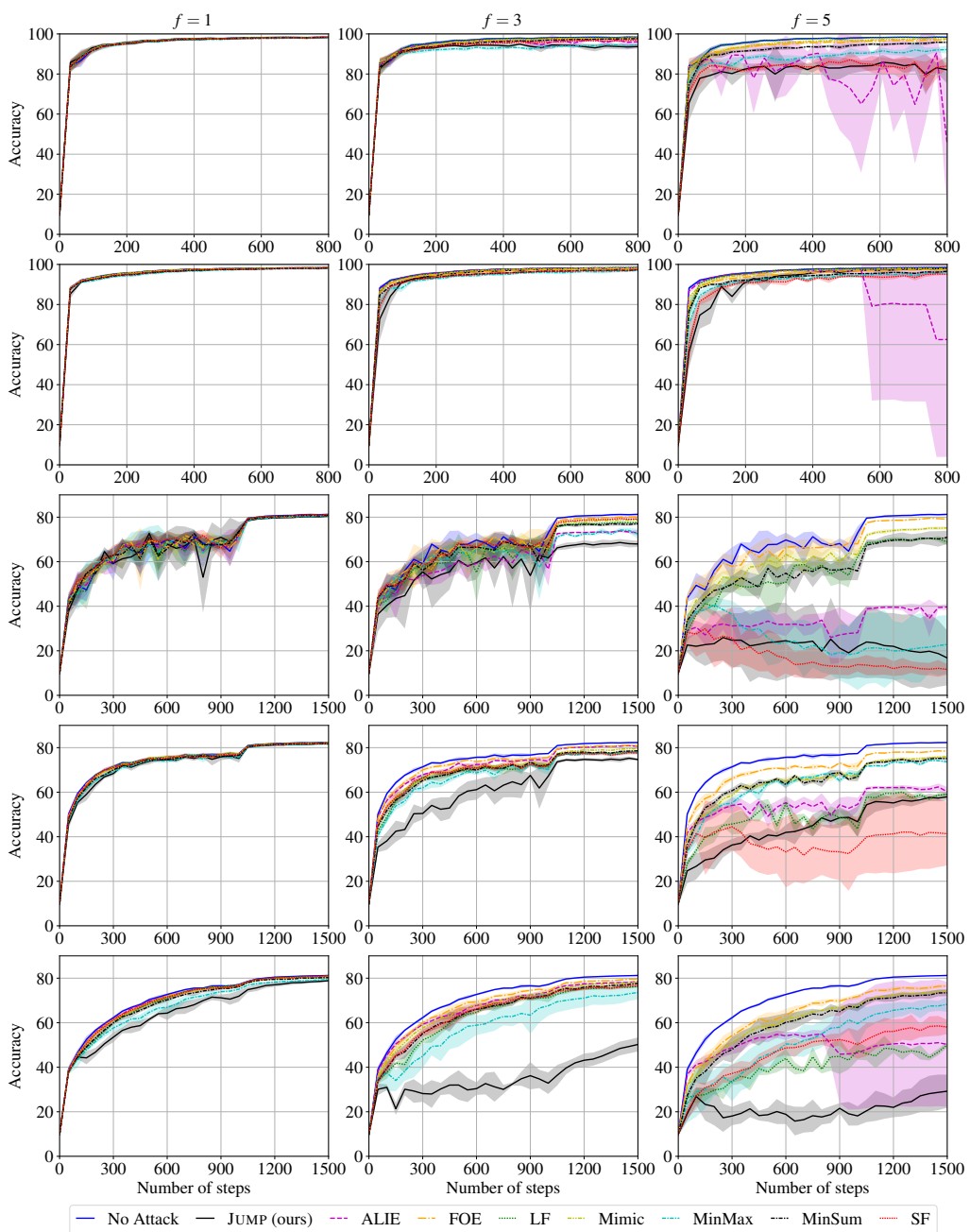

Figure 21: Accuracy evolution of Robust D-SGD with the trimmed mean aggregation on MNIST (1st and 2nd row with momentum parameter 0.0 and 0.9 respectively) and CIFAR-10 (3rd, 4th, and 5th row with momentum parameter 0.0, 0.9, and 0.99 respectively), with moderate data heterogeneity ($\alpha = 1.0$), under JUMP and other representative attacks for $f = 1$ (1st column), $f = 3$ (2nd column), and $f = 5$ (3rd column) Byzantine workers. Each experiment is run 5 times and we show the 95% confidence interval.

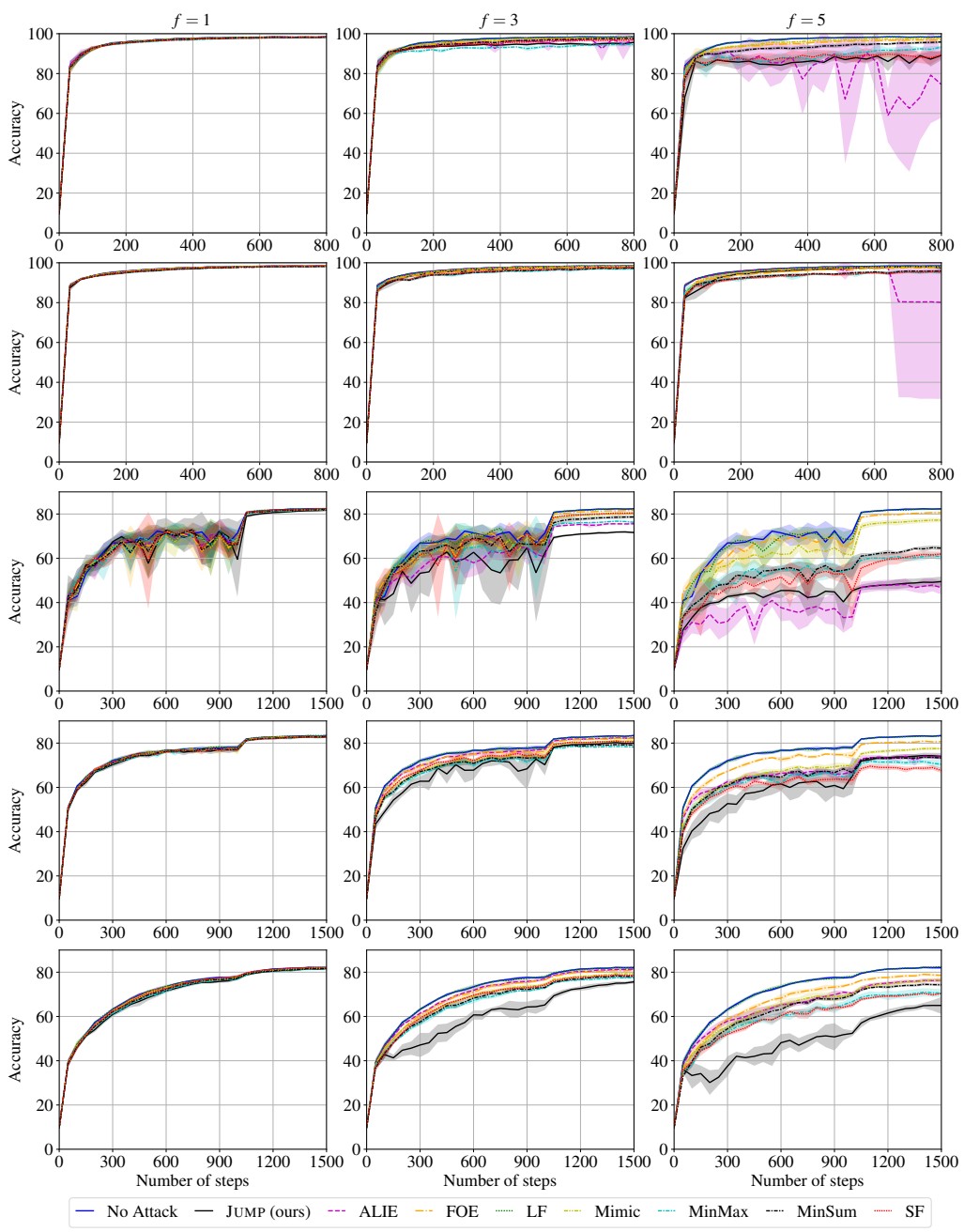

Figure 22: Accuracy evolution of Robust D-SGD with the trimmed mean aggregation on MNIST (1st and 2nd row with momentum parameter 0.0 and 0.9 respectively) and CIFAR-10 (3rd, 4th, and 5th row with momentum parameter 0.0, 0.9, and 0.99 respectively), with high data heterogeneity ($\alpha = 10.0$), under JUMP and other representative attacks for $f = 1$ (1st column), $f = 3$ (2nd column), and $f = 5$ (3rd column) Byzantine workers. Each experiment is run 5 times and we show the 95% confidence interval.

**Results on Krum.** The results with the Krum aggregation are in figures 23, 24, and 25.

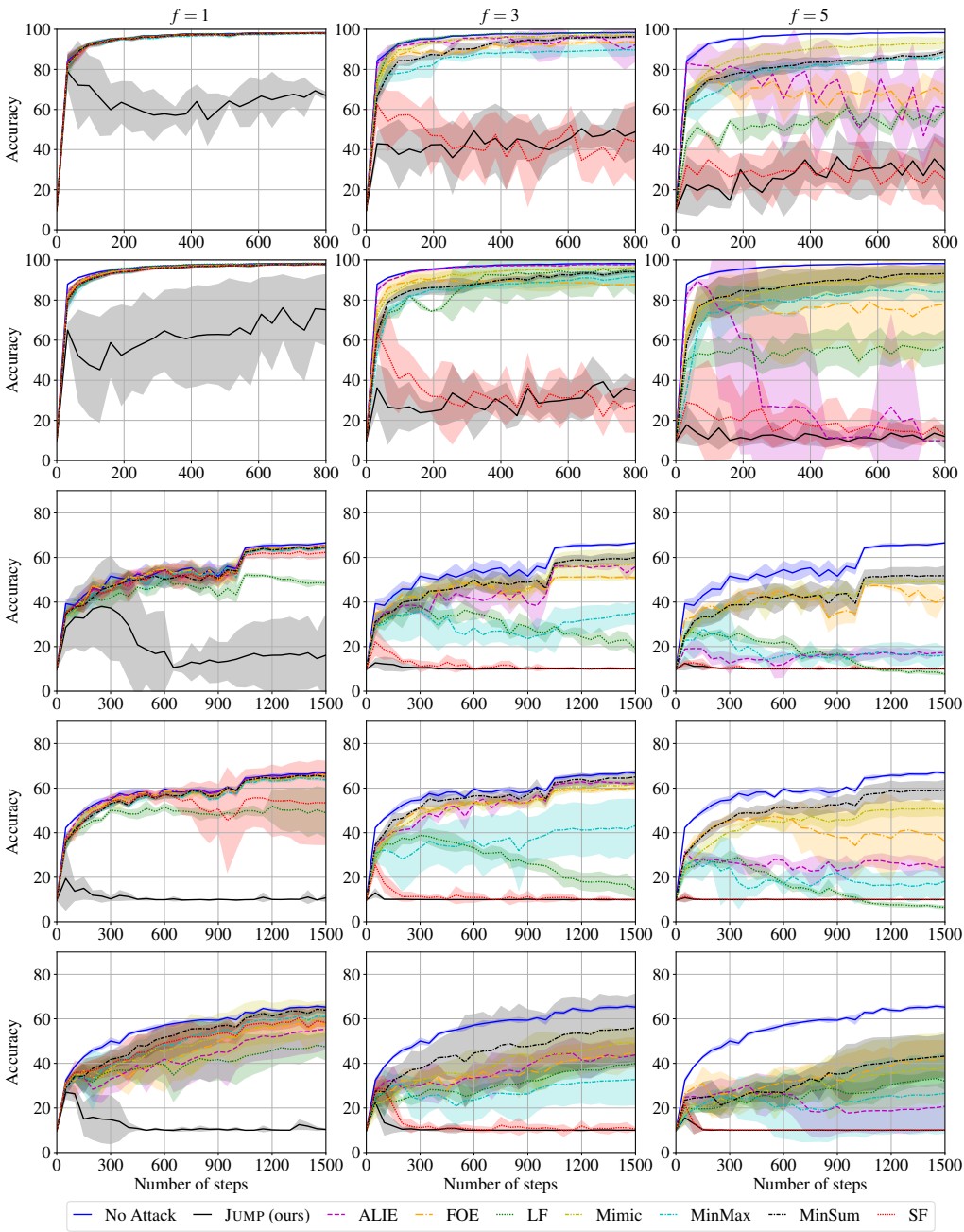

Figure 23: Accuracy evolution of Robust D-SGD with the Krum aggregation on MNIST (1st and 2nd row with momentum parameter 0.0 and 0.9 respectively) and CIFAR-10 (3rd, 4th, and 5th row with momentum parameter 0.0, 0.9, and 0.99 respectively), with high data heterogeneity ($\alpha = 0.1$), under JUMP and other representative attacks for $f = 1$ (1st column), $f = 3$ (2nd column), and $f = 5$ (3rd column) Byzantine workers. Each experiment is run 5 times and we show the 95% confidence interval.

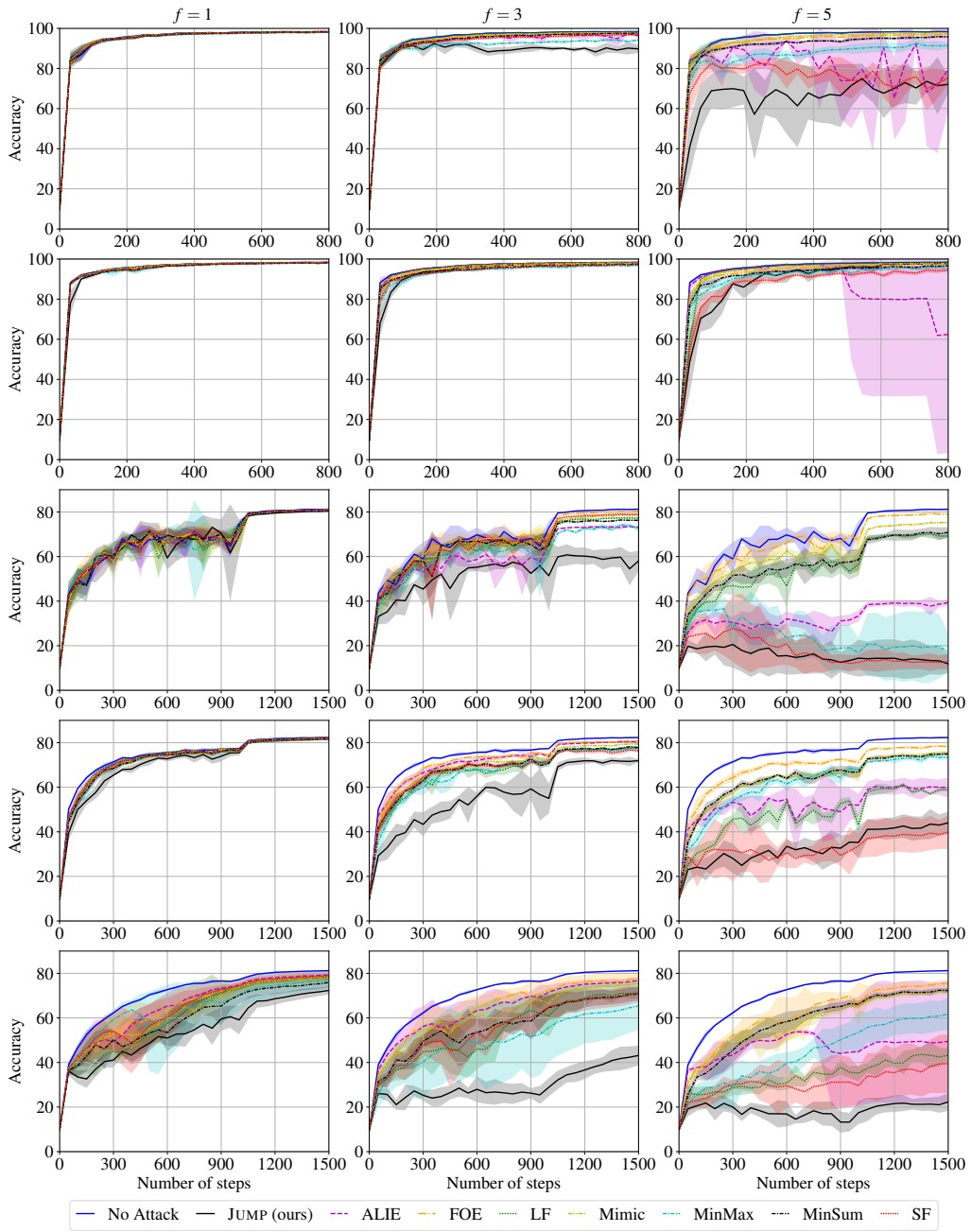

Figure 24: Accuracy evolution of Robust D-SGD with the Krum aggregation on MNIST (1st and 2nd row with momentum parameter $0.0$ and $0.9$ respectively) and CIFAR-10 (3rd, 4th, and 5th row with momentum parameter $0.0$, $0.9$, and $0.99$ respectively), with moderate data heterogeneity ($\alpha = 1.0$), under JUMP and other representative attacks for $f = 1$ (1st column), $f = 3$ (2nd column), and $f = 5$ (3rd column) Byzantine workers. Each experiment is run 5 times and we show the 95% confidence interval.

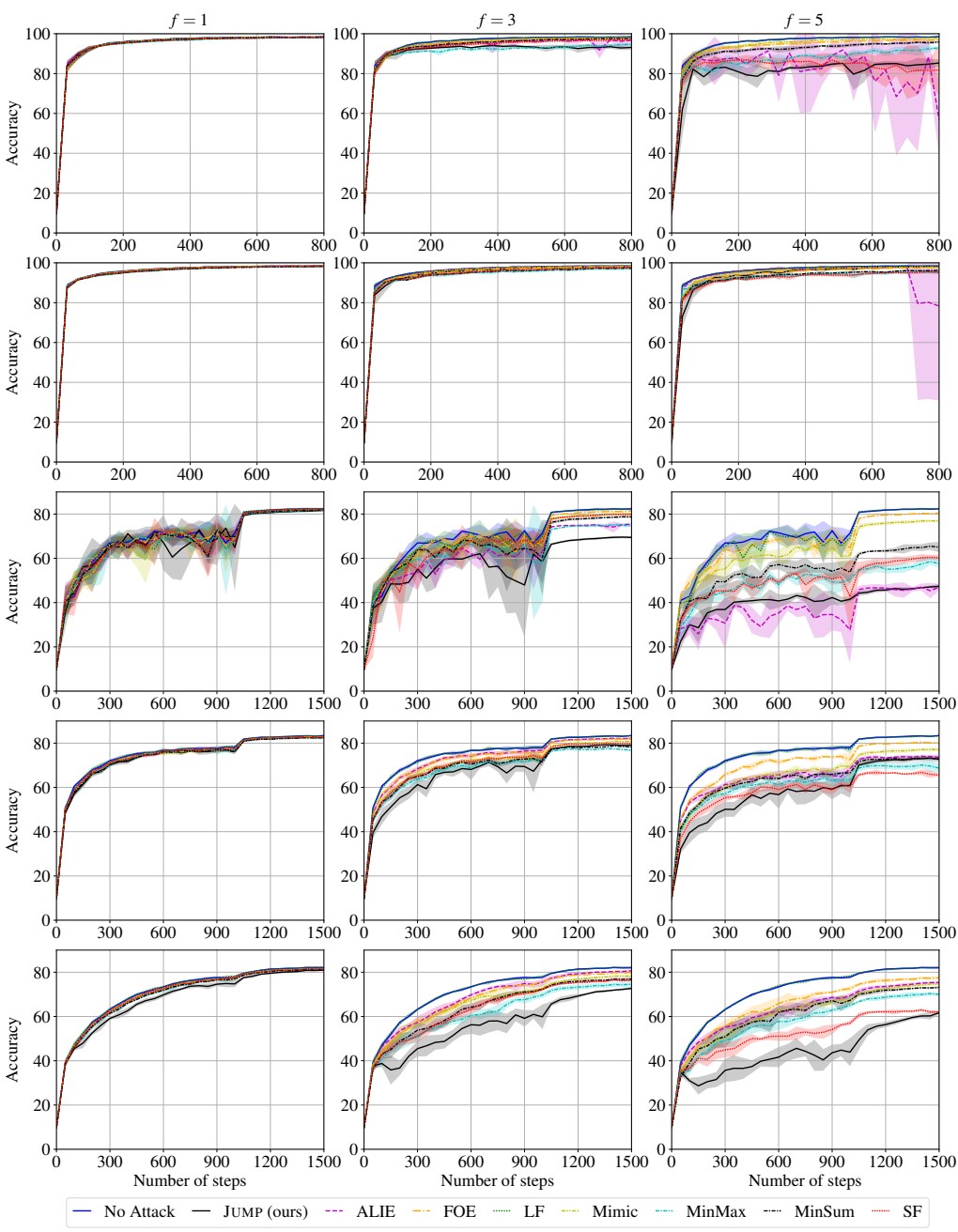

Figure 25: Accuracy evolution of Robust D-SGD with the Krum aggregation on MNIST (1st and 2nd row with momentum parameter 0.0 and 0.9 respectively) and CIFAR-10 (3rd, 4th, and 5th row with momentum parameter 0.0, 0.9, and 0.99 respectively), with low data heterogeneity ($\alpha = 10.0$), under JUMP and other representative attacks for $f = 1$ (1st column), $f = 3$ (2nd column), and $f = 5$ (3rd column) Byzantine workers. Each experiment is run 5 times and we show the 95% confidence interval.

