# OpenReview forum: "Stress Testing Byzantine Robustness in Distributed Learning"
_ICLR.cc/2024/Conference — Submitted to ICLR 2024_

### Official Review · Reviewer_Z2HT · 2023-10-31

**Soundness:** 2 fair
**Presentation:** 2 fair
**Contribution:** 2 fair
**Rating:** 3
**Confidence:** 4

**Summary:**

The authors propose a model of the optimal Byzantine adversary and derive a novel long-term attack strategy called JUMP based on the simplified version of the proposed model. Experiments on two image classification tasks show that JUMP performs better than or comparably to existing attacks.

**Strengths:**

The paper is not hard to understand. The presented adversary model (P) looks reasonable to me. Experimental results show that JUMP performs better than existing attacks on degrading model accuracy.

**Weaknesses:**

Despite the strengths, there are also some concerns.

1. Although the adversary model (P) looks reasonable, it seems oversimplified in section 3. Specifically, after the simplification (I) and (ii) in section 3, the adversary vector is restricted to be colinear with the average of the true gradients. Moreover, considering that in the FOE attack (Xie et al., 2020), the adversary vector is $-\epsilon$ times the average of the true gradients. Although I understand that the hyper-parameter $\epsilon$ in FOE is a pre-fixed constant, while $\lambda_t$ in JUMP is obtained by solving the optimization problem ($P_l$), the novelty of JUMP is limited due to the oversimplification. It seems that simplification (ii) is to simplify the optimization problem but greatly weakens the generalization of the proposed model. The reason why $a_t$ is restricted to be colinear with the average of true gradients is not well specified. I would greatly appreciate it if the authors could comment on this.

2. The proposed attack requires much more information than existing attacks. Other than the clients' local gradients (or momentum) that are required in ALIE (Baruch et al., 2019) and FOE (Xie et al., 2020), the proposed attack JUMP also assumes that the adversary has access to the global honest loss $\mathcal{L}_\mathcal{H}(\cdot)$ and the robust aggregation rule $F(\cdot)$ on the server. The extra assumption needs to be specified. Moreover, could the proposed method JUMP deal with the case where there is randomness in the robust aggregation $F(\cdot)$ and the aggregated result is not a deterministic value?

3. JUMP seems to have a much higher computation cost than existing attacks such as ALIE and FOE. However, there is neither a theoretical analysis of the time complexity nor the experimental results of the running time in the main text.

**Questions:**

Please comment on my concerns above.

---

> ### Author Response · Authors · 2023-11-16
>
> We thank you for your comments and address your concerns below. We hope that you re-evaluate our paper in this light, and we are happy to answer any further questions.
>
> >Although the adversary model (P) looks reasonable, it seems oversimplified in section 3
>
> We agree that the simplifications may seem to degrade the power of the adversary. In fact, we dedicate Section 4 to explaining why these simplifications $\textit{only marginally}$ degrade the strength of the attacker.
> For example, regarding the collinearity constraint brought up by the reviewer, we show in Section 4.1 that removing this constraint does not improve the attack by much, since the main ingredient of the attack is to overshoot minima, i.e., going in the same direction as the average of honest gradient, which is allowed by our collinearity constraint.
>
> >The proposed attack requires much more information than existing attacks
>
> We acknowledge that our attack is open-box and has full knowledge of the system and defense. This is currently stated clearly in sections 1.1 and 2.
> The purpose of our paper is not to propose a realistic or black-box attack. Rather, we (i) formulate and study the strategy followed by the optimal attack (i.e., solving (P)) in sections 2 and 4, respectively, and (ii) propose a tractable attack which retains most of the strength of the worst-case attack by building on the same strategy, as explained in Section 4.
> We recall that the aforementioned strategy consists in overshooting minima by amplifying the direction of the average honest gradient, at the right moment of training by being aware of the loss landscape.
> Furthermore, based on our result, we provide researchers with a tool to stress test the strength of their defenses in the worst case, using one attack, since it is theoretically claimed by Byzantine-robust defenses (Allouah et al. 2023, Karimireddy et al. 2021, 2022, Farhadkhani et al. 2022) to be able to withstand any attack.
> For example, even with the full knowledge of our attack, the SOTA defenses all perform well under low data heterogeneity, which actually confirms existing theory in the i.i.d. setting (Farhadkhani et al. 2022).
>
> Besides, our attack does not need to know all the data of honest clients; it only needs to know the average of honest gradients (see Algorithm 1) and some proxy data of the same (global) distribution. Although this is not a concern in this paper, but in practice it can be done by eavesdropping and data reconstruction attacks (e.g., gradient inversion).
> We will emphasize this point further in the paper.
>
> Finally, Shejwalkar \& Houmansadr (2021) analyze state-of-the-art attacks which possess the same level of knowledge as Jump, i.e., knowledge of the system, honest updates and global distribution. Our work belongs to the same line of research.
>
> >JUMP seems to have a much higher computation cost than existing attacks such as ALIE and FOE. However, there is neither a theoretical analysis of the time complexity nor the experimental results of the running time in the main text
>
> We provide an analysis in Section A of the computational cost and also its trade-off with the power of the attack for various segment lengths.
> Notably, the greedy version of Jump $(\tau=1)$ has a reasonable runtime, especially when considering that future benchmarks will only need to run Jump instead of several other attacks.
> We could not include this discussion in the main body due to lack of space, and because we admittedly focus on a worst-case unbounded adversary.
>
> ### Clarifying remarks
> Our attack is not simply an optimization of FOE. We show in Section 4.2, among other factors, that allowing the attack vector to be in the \emph{same direction} as the true gradient is essential to Jump, and is a key reason to its superiority over FOE. Also, directly maximizing the honest loss grants adversaries the ability to circumvent at the right moment during training. Finally, Jump enables adversaries to plan ahead when $\tau>1$ and maximize the training loss across iterations, which is not possible with FOE and other attacks.

---

> ### Comment · Reviewer_Z2HT · 2023-11-23
>
> Thank you for the detailed response. However, the concerns listed below have not been adequately addressed.
>
> 1. My concern about the oversimplification of the model (P) remains. The simplified adversary model $(P_l)$ loses many possibilities of potential attack strategies. Moreover, the greedy version ($\tau=1$) of Jump is quite close to the existing attack FOE, while the run time significantly increases when $\tau$ increases as shown in Section A.
>
> 2. The proposed attack requires the knowledge of the workers' average gradient, the server's defense strategy, and some proxy data of the global distribution, which is unrealistic. The authors claim they are to provide researchers with a tool to stress test the strength of their defenses. However, the contribution is quite limited since the attack in the test can be rarely operated in the real world.
>
> Given the reasons above, my rating remains.

---

### Official Review · Reviewer_9W2c · 2023-11-01

**Soundness:** 3 good
**Presentation:** 3 good
**Contribution:** 3 good
**Rating:** 6
**Confidence:** 3

**Summary:**

The paper proposes JUMP, a new adaptive attack strategy for Byzantine learning algorithm. The key idea is to formulate the attack problem into an optimization problem and solve it with off-the-shelf solvers. Tis problem is highly challenging to solve directly so some simplifications are made. Experiments show that JUMP can significantly outperform existing attacks.

**Strengths:**

1. The proposed algorithm is significantly better than baselines.
2. The optimization problem formulation is new and novel given most existing attacks are heuristics-based.

**Weaknesses:**

This is an interesting work for which I did not find major weakness, a few minor points that can improve.
1. The JUMP optimization problem seems to assume realized gradient sampling which might be a bit strong in practice, it can be possibly formulated into an robust or expectation optimization problem without assuming noise is realized.
2. More datasets in experiments can always help validate the algorithm more.

**Questions:**

Could the authors clarify more on if the same gradient sampling noise is used in the solver and true algorithm update or different realizations are used?

---

> ### Author Response · Authors · 2023-11-16
>
> We thank you for your encouraging comments and address your concerns below. We hope that you re-evaluate our paper in this light, and we are happy to answer any further questions.
>
> >Could the authors clarify more on if the same gradient sampling noise is used in the solver and true algorithm update or different realizations are used?
>
> In the results shown in our paper, the attack uses the same noise realizations as those of the honest workers.
> This is in line with our threat model, where the adversary has full knowledge, including which batches are used by honest workers.
> Interestingly, even with this knowledge, the SOTA defenses all perform well under low data heterogeneity, which confirms existing theory in the i.i.d. setting (Farhadkhani et al. 2022).
> We will include a discussion on this point in the paper.
>
> >The JUMP optimization problem seems to assume realized gradient sampling which might be a bit strong in practice, it can be possibly formulated into an robust or expectation optimization problem without assuming noise is realized.
>
> We appreciate your advice on loosening our assumption by formulating our problem into a robust optimization problem without assuming noise is realized.
> We agree this could yield more realistic attacks, but we recall that our goal is to propose a worst-case attack of a truly Byzantine adversary, i.e., has full knowledge and controls all malicious workers.
> Indeed, theoretical works in our line of research (Allouah et al. 2023, Karimireddy et al. 2021, 2022, Farhadkhani et al. 2022) claim to defend against such a worst-case adversary, yet we argue that they do not actually implement the latter in their validation experiments.

---

### Official Review · Reviewer_6Sfh · 2023-11-04

**Soundness:** 3 good
**Presentation:** 3 good
**Contribution:** 2 fair
**Rating:** 5
**Confidence:** 5

**Summary:**

This paper proposes an untargetted attack on FL called, Jump attack, which aims to indiscriminately reduce the accuracy of the FL global model. Jump attack first computes the average of honest gradients and scales it by \lambda to get the final malicious gradient that all the malicious clients (attackers) then send to the server. Jump computes \lambda at certain intervals, e.g., in each FL round or after each \tau rounds. The paper shows how Jump attack can overshoot the good minima of a simple non-convex problem and force the model to a bad minima, while other attacks only slow down the convergence of model to the good local minima.  The experimental results show that the attack outperforms current SOTA attacks for certain FL settings.

**Strengths:**

- Jump attack seems very simple and potentially effective
- Section 4 is very useful and something that I haven’t seen in other works

**Weaknesses:**

- Experimental evaluation is insufficient and unfair to SOTA attacks
- It is not clear what is the solver used in the Jump attack
- Some of the relevant attacks are not evaluated
- Some very relevant works are not cited

**Questions:**

Jump attack is quite smart and I like the idea primarily because it might be very easy to implement in practice. However I have multiple concerns about the current draft.

- Shejwalkar & Houmansadr (2021) also use something similar to Jump’s malicious gradient in (4); e.g., in equation (6) of Shejwalkar & Houmansadr (2021) attack, if we substitute \eta_t = (1 - \lambda_t) while p_t = -\bar{g}^H_t we get the Jump attack. Given this I think authors should discuss the distinction between the two attacks more explicitly.
- Experimental evaluation is on very small settings: total number of clients is 12 which is very small for the FL settings that are vulnerable  to model poisoning in practice [1]. This does not discount the utility  of the work, but authors should be clear in terms of the goals of the work and where it is useful, especially positioning it with the conclusions of [1] so that readers will understand  how to use this work.
- Next, I think the experiments performed are not fair: Jump attack uses the knowledge of the robust aggregation but the attacks it is compared with are aggregation-agnostic attacks; authors should provide a fair comparison, i.e., compare with aggregation-tailored attacks of Shejwalkar & Houmansadr (2021).
- I did not understand what is the solver used in the work and how it solved the Jump’s objective; please add this to the main paper.
- Experimental setup is an important part of conveying the utility of the work. Please check [3] for details and consider more practical settings, e.g., challenging datasets, more clients, etc.


[1] Shejwalkar et al., Back to the Drawing Board: A Critical Evaluation of Poisoning Attacks on Production Federated Learning, IEEE S&P 2022
[2] Fang et al., Local Model Poisoning Attacks to Byzantine-Robust Federated Learning, USENIX 2019
[3] Khan et al., On the Pitfalls of Security Evaluation of Robust Federated Learning, IEEE S&P WSP 2023

---

> ### Author Response · Authors · 2023-11-16
>
> We thank you for your encouraging comments and address your concerns below. We hope that you re-evaluate our paper in this light, and we are happy to answer any further questions.
>
> >Shejwalkar \& Houmansadr (2021) also use something similar to Jump’s malicious gradient in (4); e.g., in equation (6) of Shejwalkar & Houmansadr (2021) attack, if we substitute $\eta_t = (1 - \lambda_t)$ while $p_t = -\bar{g}^H_t$ we get the Jump attack. Given this I think authors should discuss the distinction between the two attacks more explicitly.
>
> We have included in Section 4.2 a comparison with the attacks developed by Shejwalkar \& Houmansadr (2021).
>
> There, among other factors, we explain that a key difference is that Jump allows the perturbation vector to be in the $\textit{same direction}$ as the average honest gradient, in order to overshoot minima. For Shejwalkar \& Houmansadr's (2021) attacks, because of the optimization scheme they use, the adversary can only try to deviate from the direction of the average honest gradient (the scale factor is positive).
>
> Furthermore, Shejwalkar \& Houmansadr (2021) focus on the i.i.d. setting, which is arguably not the norm in real-world scenarios, and do not extensively investigate the heterogeneous case.
> In our work, we consider different levels of heterogeneity, using Dirichlet sampling.
> In terms of threat model, the strongest one considered by Shejwalkar \& Houmansadr (2021) also assumes full knowledge. Yet, their attacks only leverage the knowledge of the aggregation and honest updates, while our attack also utilizes the global honest loss function.
> We will also include this discussion in the paper.
>
>
> >Experimental evaluation is on very small settings: total number of clients is 12 which is very small for the FL settings that are vulnerable to model poisoning in practice [1]. This does not discount the utility of the work, but authors should be clear in terms of the goals of the work and where it is useful, especially positioning it with the conclusions of [1] so that readers will understand how to use this work.
>
> We acknowledge that our experiments are not large-scale, since our work is directed towards researchers who need strong attacks to verify their theoretical worst-case Byzantine robustness claims. We will emphasize this point further in the paper.
>
> Besides, regarding how challenging the datasets are, we use heterogeneous variants of MNIST and CIFAR-10, where the extreme heterogeneity scenario is challenging and is common in Byzantine-robustness works (Allouah et al. 2023, Karimireddy et al. 2022). In any case, note that the more challenging the task, the harder it is for the defender and not the attacker, which is why we focus on moderate heterogeneity, or low fraction of Byzantine workers.
>
> >Next, I think the experiments performed are not fair: Jump attack uses the knowledge of the robust aggregation but the attacks it is compared with are aggregation-agnostic attacks; authors should provide a fair comparison, i.e., compare with aggregation-tailored attacks of Shejwalkar \& Houmansadr (2021).
>
> We believe that it would be inadequate to compare Jump to aggregation-tailored attacks of Shejwalkar \& Houmansadr (2021), given that these attacks choose perturbation vectors depending on the defense, unlike Jump.
> Now, to automatically choose a perturbation vector for any aggregation rule, Shejwalkar \& Houmansadr (2021) propose to simulate a smaller FL system and evaluate aggregation-tailored attacks with all possible perturbation vectors, then pick the best.
> However, it is unclear how this can be implemented outside the i.i.d. setting they were intended for, since reproducing the data heterogeneity of the original system is not straightforward.
> We will include this discussion in the paper.
>
> For completeness, $\textbf{we updated our paper with results}$ of our experiments with the aggregation-tailored attack of Shejwalkar \& Houmansadr (2021) that uses the inverse unit perturbation vector as an example (see Section 4.2).
> The main change in our results of Table 1 (Section 5) is that when $f=3$ using Krum with NNM, our attack becomes comparable to aggregation-tailored attack (``AGRT'' in Table 1). Otherwise, our conclusions still hold.

---

> > ### Comment · Reviewer_6Sfh · 2023-11-19
> > **Thanks for the response!**
> >
> > 1. Comparison with SH2021:
> > - Thanks for adding the discussions and comparisons with .
> > - About the evaluation setup comparison: I agree that i.i.d. FL is not practical and I appreciate that the current work sticks to non-i.i.d. setting. However, it is *more* difficult to attack i.i.d. settings than to attack non-i.i.d. settings as the latter allows attacker much more leeway to craft malicious vectors while remaining undetected. Hence, if an attack works in i.i.d. settings, it will definitely work in non-i.i.d. settings.
> > - SH2021 also consider partial knowledge attacks where attacker does not use knowledge of honest updates. Nevertheless, if you really want to stress-test FL (as your next answer mentions) you should consider *both* full and partial knowledge attacks as in SH21. Although I think this is generally not needed as it is close to impossible for attackers to access honest updates or losses (check [1] in reference I provided before).
> >
> > 2. Evaluation setup:
> > - The current work is definitely very useful. But IMO, it will be much more useful if you increase the total number of clients from 13/15/17 to 1000 for MNIST or CIFAR10, consider cross-device setting (along with cross-silo). In my experience, running FedAvg or similar algorithms on small datasets like MNIST, CIFAR10 with 1000 clients is possible with minimal resources, e.g., a single 1080ti GPU; you can even run FEMNIST with 3400 clients.
> > - The extreme heterogeneity you consider is dirichlet with \alpha=0.1 which is great, only if you have a very large number of clients. With small number of clients considered here, this distribution will *not be extremely heterogeneous*.
> > - I agree that to stress-test defenses, one should make settings in favor of attacker, and that is exactly why I suggest you consider larger numbers of clients and cross-device settings.
> >
> > 3. Fairness of comparisons:
> > - AGRT of SH2021 are not defense dependent; at best, it is dataset dependent, e.g., it is suggested to use inverse standard deviation for CIFAR10.
> > - The root of unfairness is the fact that JUMP attack uses all the available information, including honest local losses, defense, etc., while the baseline attacks do not, e.g., ALIE, Min-max, Mimic are defense-agnostic, LF, SF are very weak in general.
> > - Emulating FL locally as in SH21: You know that the data is dirichlet distributed with some \alpha. You can use the same to emulate FL locally.
> > -Thanks for adding new comparisons, but please find the correct perturbation as discussed above.
> >
> > My concerns 4 and 5 in weakness in original review remain unaddressed.

---

> ### Author Response · Authors · 2023-11-20
>
> We thank you for your response and follow up on them below.
> We hope that you re-evaluate our paper in this light.
>
> >About the evaluation setup comparison: I agree that i.i.d. FL is not practical and I appreciate that the current work sticks to non-i.i.d. setting. However, it is more difficult to attack i.i.d. settings than to attack non-i.i.d. settings as the latter allows attacker much more leeway to craft malicious vectors while remaining undetected.
>
> We agree with both statements. This is why we focus in the main paper on a moderate heterogeneity setting ($\alpha=1$), which is both practical and challenging to attackers.
> In addition, we have results of experiments on low, moderate, and high data heterogeneity in the appendix.
>
> >Nevertheless, if you really want to stress-test FL (as your next answer mentions) you should consider both full and partial knowledge attacks as in SH21.
>
> We stress test Byzantine robustness in distributed learning against an open-box adversary. We agree that the partial knowledge setting is interesting on its own, but is out of the scope of our work.
>
> >But IMO, it will be much more useful if you increase the total number of clients from 13/15/17 to 1000 for MNIST or CIFAR10, consider cross-device setting (along with cross-silo).
>
> We thank the reviewer for their suggestion.
> We agree that a larger scale experiment can be an interesting addition.
> However, we chose to focus on FL settings that are reasonably easy for the defender, and where theoretical defenses have tight guarantees.
> In contrast, cross-device settings, where client subsampling and local steps are used, have considerably weaker and fewer theoretical guarantees.
>
> >The extreme heterogeneity you consider is dirichlet with $\alpha=0.1$ which is great, only if you have a very large number of clients. With small number of clients considered here, this distribution will not be extremely heterogeneous.
>
> For the experiments we show, most clients have only one label in their dataset, which we believe is an extreme form of heterogeneity.
>
> >I agree that to stress-test defenses, one should make settings in favor of attacker, and that is exactly why I suggest you consider larger numbers of clients and cross-device settings.
>
> For clarity, we would like to reemphasize that our goal is to stress test the worst-case guarantees of Byzantine-robust defenses, and thus we chose our FL settings to be reasonably $\underline{\text{in favor of the defender}}$ (not the attacker): low fraction of Byzantine workers, moderate heterogeneity, SOTA defenses for non-i.i.d. settings, etc...
>
>
> >AGRT of SH2021 are not defense dependent; at best, it is dataset dependent.
>
> We believe that there is a misunderstanding:
> in Section IV.B, Shejwalkar \& Houmansadr (2021) propose using different optimization formulations following the aggregation used.
> This led us to think that aggregation-tailored attacks are indeed defense-dependent.
>
> Also, in Section VI.C, Shejwalkar \& Houmansadr (2021) also utilize the knowledge of server’s aggregations to choose perturbation vectors. Figure 2 also shows that even with the same dataset, different aggregations sometimes have different best perturbation (like in CIFAR10, Krum's best perturbation vector is inverse unit vector, but Trimmed-mean's best perturbation is standard deviation).
>
>
> >The root of unfairness is the fact that JUMP attack uses all the available information, including honest local losses, defense, etc., while the baseline attacks do not, e.g., ALIE, Min-max, Mimic are defense-agnostic, LF, SF are very weak in general.
>
> We have added results for AGRT, which has full knowledge, and is still outperformed by Jump.
> Besides, the fact that previous attacks have not utilized the open-box adversary's information is arguably not ``unfairness''.
> A major contribution of our work is precisely on how to leverage the full information to break defenses via the overshooting strategy of Jump.

---

> > ### Author Response · Authors · 2023-11-20
> >
> > >Emulating FL locally as in SH21: You know that the data is dirichlet distributed with some $\alpha$. You can use the same to emulate FL locally. -Thanks for adding new comparisons, but please find the correct perturbation as discussed above.
> >
> > While an open-box attacker could know the distribution of the data, it would be unreasonable to assume that the defender (e.g., server) knows this information.
> > In turn, we note that the method shown in the original paper of Shejwalkar \& Houmansadr (2021), which assumes an i.i.d. setting, is possible in practice; the attacker can access a corrupted worker's data, which is representative of the global distribution given the i.i.d. assumption.
> >
> > Moreover, the data distribution obtained with Dirichlet sampling depends on the number of workers, which can be problematic for the method proposed by the reviewer to extend the perturbation vector search method of SH2021.
> > Once again, finding a correct perturbation vector does not seem straightforward in the non-i.i.d. setup.
> >
> > In any case, we still showed the performance of AGRT with unit perturbation vector, which we saw performed well with MinMax and MinSum, the results of which are using the same perturbation vector.
> >
> > >My concerns 4 and 5 in weakness in original review remain unaddressed.
> >
> >
> > Regarding concern 5, we have justified our experimental setup in our previous response: we chose our FL settings to be reasonably in favor of the defender (not the attacker); low fraction of Byzantine workers, moderate heterogeneity, SOTA defenses for non-i.i.d. settings, no client subsampling/local steps, etc...
> >
> > Regarding concern 4, we apologize that our answer was not included in our initial response by error. Here is our response:
> >
> > >I did not understand what is the solver used in the work and how it solved the Jump’s objective.
> >
> > In Section 3.2, we discuss appropriate nonlinear solvers for our tasks. We recall  that the constraints of Problem $(P_\ell)$ are non-differentiable in general due to robust aggregation. This motivates the use of derivative-free non-linear solvers such as Powell’s (Powell, 1964), Nelder-Mead’s (Nelder \& Mead, 1965), and Differential Evolution (Storn \& Price, 1997). In fact, we have tested all the aforementioned methods on MNIST and our toy two-dimensional tasks. Both cases showed that Powell's is most effective while Differential Evolution can obtain better solutions but is much slower than Powell's, so we use Powell's in our main experiments.
> > Powell's method, strictly Powell's conjugate direction method, minimizes the objective function by a bi-directional search along each search vector, successively. The bi-directional line search along each search vector can be done by Golden-section search, or Brent's method when search space is boundless. The algorithm iterates an arbitrary number of times until no significant improvement is made.
> > We will add a detailed description in the paper.
> >
> > We will be happy to answer any further questions.

---

### Official Review · Reviewer_hYhe · 2023-11-05

**Soundness:** 3 good
**Presentation:** 2 fair
**Contribution:** 3 good
**Rating:** 5
**Confidence:** 3

**Summary:**

A new Byzantine attack on federated learning is introduced, that purports to significantly improve upon prior techniques.

**Strengths:**

Interesting problem space, a significant number of numerical experiments (although I do have questions regarding if the experiments properly contextualise the performance as the number of byzantine agents grows, as the primary comparisons top out at 29% compromised), and well incorporated toy-models that have been appropriately used to help explain key dynamics.

**Weaknesses:**

The style and quality of the writing is to me a significant issue, as it makes it difficult to parse the authors intent at times (examples of some points of concern in the earlier sections are raised below). I also hold concerns relating to the nature of the contributions, and the claimed performance increases that the authors have achieved, especially as the level of outperformance drops in the results reported in the appendices.

The following is a set of general issues/questions I have with specific parts of the paper, as well as a commentary on areas where the issues regarding the writing style and lack of specificity in the written content cause interpretation issues (with particular focus placed upon the abstract and introduction).

General questions:
- The concept of robust aggregation processes is to filter out likely adversaries. If all adversaries are equal, then detecting this cluster would likely be a trivial inclusion into robust aggregation. How would your work perform if you were unable to impose uniformity among the adversary update vectors?
- How reasonable is the assumption regarding the model dimension? Obviously this is a worst-case assumption, but if the adversary only has access to the local information on the set of compromised agents, does this assumption hold in practice? If an attacker has enough information to know the updates of all agents (both compromised and uncompromised) then they've compromised the central update server, which you've assumed is secure. This seems like an inherent contradiction in the threat model. I will note though that I believe I have seen this in other byzantine works, so this may not be an issue specific to your work.
- You note on page 5 that this approach works for any variant of robust D-SGD - but how do these modifications influence the performance of your technique?
- Could you explain the segment length in additional detail? At the moment problem P_l implies that the attacker only acts every $\tau$ steps - but this would appear to be contradicted by the loss subfigure from Figure 1.
- One of the reasons that JUMP is claimed to work is that it can force trajectory jumps over global minima - however this would seem to be an observation that would be very sensitive to the largest allowable trajectory jumps, and yet this doesn't seem to be a property that has been explored.
- How would your technique perform in the context of a) more agents, or b) when the number of agents is fixed, but the proportion of byzantine agents increased?
- Why does Table 1 show results that are only positive for your technique, whereas the "full numerical results" from Appendix C2 reveal ranges of setups where your technique is outperformed. This feels as if the main body contents have been pruned to show your technique in the best possible light. Could you explain the choices of content included in Table 1 vs C2?

Abstract:
- "consists in ensuring" - generally the idiomatic follow on from "consists" would be of, but more broadly there's no need for consists within this sentence. "Byzantine robustness in distributed learning ensures that distributed...."
- The first two sentences imply that there are other kinds of workers other than Byzantine workers, yet the idea of what a worker is or how they're involved in the process is not defined.
- "so far" implies that they've been defined within this work, whereas "to date" more clearly contextualises this statement relative to historical work.
- "critical points" - undefined and unspecified, relies on a familiarity with the field that the reader may not have. I'm assuming that this is convergence points of the learning process, however this isn't clear.
- "is a solution to a simplified form of the optimal adversary's problem, it is very powerful" - what is the problem? A form in what way? What is powerful a measure of? This sentence doesn't have the contextual scaffolding for anyone to understand it. Moreover the framing is so convoluted that it's impossible to parse intent and meaning here - the only interpretation that would make sense would be that the authors are trying to say that JUMP may be a solution to a simplified problem, but this problem has a high degree of transferability to more complex spaces.
- "even the greedy version" - greedy in what context? What is the non-greedy version of JUMP?
- While I'm not familiar with "accuracy damage" as a metric, the framing of a doubling being going from 66% to 50% vs 81% is again framed in a fashion where the impact of this work is impenetrable.

Introduction:
- What is this new generation of models, relative to traditional machine learning? An introduction like this would imply the use of new architectures like Transformers, which I'm assuming aren't actually going to be present here.
- Typically if you're going to introduce an acronym like DSGD, then that implies that the whole item is a proper noun, so then the in text version should be Distributed SGD or Distributed-SGD, rather than distributed SGD.
- "causes safety and reliability issues" - these issues aren't guaranteed, but it can cause such issues.
- In the framing of the introduction, it's not clear if Byzantine workers are malicious or benign (as in, if the manipulation has intent behind it), and if they're potentially collaborative.
- "consistently circumvents it to worse regions of the loss landscape, where it leverages data heterogeneity to converge to poor models" The subject of the "it" is ambiguous; the involvement of data hereogeneity doesn't seem to add anything to the sentence; the idea of what a poor model is is vague and unspecified; and generally considering that this is an attack on a single, collaboratively derived model then it should be model singular rather than pluarl.
- "segment length that JUMP uses" - what segment length? Unspecified and unclear.

Section 2:
- Eqn 2 contextualises H as the full set of weight updates. Equation P [also, why is this equation P?] introduces $i \notin \mathcal{H}$ - but by this definition, the elements $i \notin \mathcal{H}$ aren't incorporated into the algorithm at all.

Section 4 and appendices:
- The blue/red vectors on the dense blue/green contours are quite difficult to parse visually - a problem that is even worse in the figures contained within the appendices.

Section 5:
- In table 1 it's only implicit that this is referring to accuracy, and accuracy is not even mentioned until 4 lines into the table caption, which is certainly a choice.
- Table talks about heterogeneity - which ahs been unspecified up to this point (there are 5 prior references on pages 1 and 2, and all of these prior references are incredibly abstract. This issue stems from the positioning of Table 1, and how it preceeds the content that is used to describe it. It's very easy to read table 1 as a part of the 'key reasons of jump superiority" subsection.
- Table 1 doesn't include detail over the number of experiments that were used to capture the standard deviation (or range, who knows, because these details aren't specified). [See issue above re: positioning]
- Table isn't contextualised in terms of how much agents are being used in total. [See issue above]
- For the above 3 dot points, the minimum information required to interpret a table really should be included in the table caption.
- "Powerful" is ambiguous, and being "significantly more powerful" or "comprable" is ambiguous. It is possible to perform a hypothesis test to calculate the likelihood that JUMP produces a stronger deleterious impact upon the accuracy than the other experiments - this would be far more meaningful than the current framing.

Appendix C-2, Table 6 - if you're going to ommitt all other compairsons, NaN as your superiority metric could probably just be replaced with a dash as well.

**Questions:**

See above.

---

> ### Author Response · Authors · 2023-11-16
>
> We appreciate your response and detailed comments.
> We will carefully incorporate your writing advice in the paper.
> We believe we address your concerns below and hope that you re-evaluate our paper in this light.
> We are happy to answer any further questions.
>
>
>
> >Why does Table 1 show results that are only positive for your technique, whereas the "full numerical results" from Appendix C2 reveal ranges of setups where your technique is outperformed. This feels as if the main body contents have been pruned to show your technique in the best possible light. Could you explain the choices of content included in Table 1 vs C2?
>
> We recall that the results in the appendices essentially convey the same message as in Section 5, where JUMP outperforms existing attacks across the vast majority of scenarios.
> It is worth mentioning that the results for $\alpha = 10$ (i.e., low data heterogeneity) are slightly different than those of Section 5 because of the low data heterogeneity. Indeed, in the latter regime, the considered defenses all have excellent accuracies across \textit{all} attacks, which is predicted by the theory in i.i.d. settings when using local momentum (Farhadkhani et al., 2022).
> In addition, since real-world federated learning datasets are typically non-i.i.d., we chose the results on CIFAR-10 under moderate heterogeneity setting to show in the table of the main paper.
>
> >The concept of robust aggregation processes is to filter out likely adversaries. If all adversaries are equal, then detecting this cluster would likely be a trivial inclusion into robust aggregation. How would your work perform if you were unable to impose uniformity among the adversary update vectors?
>
> First, we agree that Byzantine gradients could be detected by having the aggregation eliminate identical inputs. Please note that it is standard in existing attacks and their implementation to have the Byzantine gradients be identical (Allouah et al. 2023, Karimireddy et al. 2021, 2022, Farhadkhani et al. 2022), since it is generally understood that adding small noise can make the malicious gradients nonidentical and still remain in a bad region of the space.
> However, such a defense (eliminating dense clusters) does not have theoretical guarantees and can easily be fooled in the presence of data heterogeneity, e.g., Byzantine gradients can gather around points they want to remove, which would be devastating if there is considerable heterogeneity.
> Also, in our threat model, we assume that one adversary controls several workers, or equivalently that Byzantine workers collude. This makes the problem clear, embodies the worst case, and is also standard in Byzantine Machine Learning works (Allouah et al. 2023, Karimireddy et al. 2021, 2022, Farhadkhani et al. 2022).
>
> >How reasonable is the assumption regarding the model dimension? Obviously this is a worst-case assumption, but if the adversary only has access to the local information on the set of compromised agents, does this assumption hold in practice?
>
> We are unsure of what the reviewer meant by ``model dimension''. We assume that they refer to the threat model.
>
> We acknowledge that our attack is open-box and has full knowledge of the system and defense. This is currently stated clearly in sections 1.1 and 2.
> The purpose of our paper is not to propose a realistic or black-box attack. Rather, we (i) formulate and study the strategy followed by the optimal attack (i.e., solving (P)) in sections 2 and 4, respectively, and (ii) propose a tractable attack which retains most of the strength of the worst-case attack by building on the same strategy, as explained in Section 4.
> We recall that the aforementioned strategy consists in overshooting minima by amplifying the direction of the average honest gradient, at the right moment of training by being aware of the loss landscape.
> Furthermore, based on our result, we provide researchers with a tool to stress test the strength of their defenses in the worst case, using one attack, since it is theoretically claimed by Byzantine-robust defenses (Allouah et al. 2023, Karimireddy et al. 2021, 2022, Farhadkhani et al. 2022) to be able to withstand any attack.
> For example, even with the full knowledge of our attack, the SOTA defenses all perform well under low data heterogeneity, which actually confirms existing theory in the i.i.d. setting (Farhadkhani et al. 2022).
>
> Besides, our attack does not need to know all the data of honest clients; it only needs to know the average of honest gradients (see Algorithm 1) and some proxy data of the same (global) distribution. Although this is not a concern in this paper, but in practice it can be done by eavesdropping and data reconstruction attacks (e.g., gradient inversion).
> We will emphasize this point further in the paper.

---

> ### Author Response · Authors · 2023-11-16
>
> >You note on page 5 that this approach works for any variant of robust D-SGD - but how do these modifications influence the performance of your technique?
>
> This is a good question.
> First, note that the SOTA defense in Byzantine ML is robust D-SGD with local momentum with pre-aggregation (Allouah et al. 2023, Karimireddy et al. 2022), so our experimental results are shown for this defense.
> We also test on other variants in Appendix C.2 with less or no local momentum, where we explain that our conclusions regarding the performance of Jump still hold.
>
>
> >Could you explain the segment length in additional detail? At the moment problem $P_l$ implies that the attacker only acts every  steps - but this would appear to be contradicted by the loss subfigure from Figure 1.
>
> We explain the steps of the \textsc{Jump} attack in Section 3.2. Segment length $\tau$ is the size of each subproblem $P_\ell$. In other words, we split the whole training process of $T$ iterations into $\lceil \frac{T}{\tau} \rceil$ segments with length $\tau$. Each time we solve a subproblem $P_\ell$, we obtain $\lambda_{(\ell-1)\tau+1}, \ldots, \lambda_{\ell \tau}$ which can be applied to training iterations $(\ell-1)\tau+1, \ldots, \ell \tau$. Hence, after we solve all subproblems, we obtain $\lambda_{1}, \ldots, \lambda_{T}$ which are used to generate the Byzantine vectors for the whole training process.
>
> >One of the reasons that JUMP is claimed to work is that it can force trajectory jumps over global minima - however this would seem to be an observation that would be very sensitive to the largest allowable trajectory jumps, and yet this doesn't seem to be a property that has been explored.
>
> This is a very good question.
> An $\textit{optimal}$ jump is indeed sensitive to the largest allowable trajectory jumps and to the loss landscape.
> Furthermore, the quality of the landing point influences the effectiveness of our attack, as we explain in Section 4.
> However, note that the learning rate used is tuned on the Byzantine-free case, and thus is not too large. In fact, we have tested with multiple learning rates, but the strategy followed by Jump, and the optimal attack obtained by solving Problem (P), does not change.
> This is because we use nonlinear programming (Powell's method in our experiments) to find an optimal jump strategy, including finding the optimal point to do a wise and precise jump, as explained in Section 4.
>
>
> >How would your technique perform in the context of a) more agents, or b) when the number of agents is fixed, but the proportion of byzantine agents increased?
>
> To showcase the strength of our attack, we focus our experiments on small-scale tasks and a low fraction of Byzantine workers.
> Indeed, the more complex the task, and the larger the Byzantine fraction, the harder it is for the defender and not the attacker.
> This is why, given that we propose an attack, we focus on this harder setting for the attacker.
> Finally, we recall that we already conducted a larger number of experiments, for different levels of heterogeneity, on different datasets, and Byzantine fractions.
>
> ### Clarifying remarks
> ``critical points'' are also called stationary/saddle points in non-convex optimization, and refer to model parameters where the gradient of the loss is zero.
>
> ``greedy'' refers to the attack obtained by setting $\tau$, the segment length, to $1$ in Jump (see Section 3.2). When $\tau>1$, the attack plans malicious gradients for the next $\tau$ iterations, which makes it a non-greedy approach.
>
> ``accuracy damage'' refers to the decrease in accuracy (i.e., accuracy with best existing attack accuracy minus accuracy with our attack) relative to the Byzantine-free baseline.
> We will make these clear in the paper.

---

> > ### Comment · Reviewer_hYhe · 2023-11-19
> >
> > Thank you for your detailed response to my somewhat unwieldy review - and I apologise for the comment re model dimensions, which from memory I was using instead as a mental shorthand for both the fraction of Byzantine workers and the number of total workers (a poor mental shorthand, I will admit).
> >
> > I will state though that the idea of the attacker still requiring access to the average of the honest gradients still seems like it would necessitate a level of access that would negate the need for an attacker to introduce Byzantine workers in the first place. Obviously white box threat models in a range of contexts permit strong degrees of access, but I would still contend that typically a white-box attacker doesn't imply so much access that the attack itself isn't needed. However, I will also acknowledge that my knowledge of attacks against federated learning is weaker than for other adversarial contexts, and I may be raising issue with a point that is well accepted within the Byzantine robustness community.
> >
> > While I appreciate the authors work, the time they placed into their rebuttal, and the promises to add additional detail to the paper, based upon the manuscript as it currently stands I maintain my review, although I also will definitely spend more time digesting your responses to the other reviewers over this period.

---

> > > ### Author Response · Authors · 2023-11-20
> > >
> > > We thank you for your response and follow up on them below.
> > > We hope that you re-evaluate our paper in this light.
> > >
> > > >I will state though that the idea of the attacker still requiring access to the average of the honest gradients still seems like it would necessitate a level of access that would negate the need for an attacker to introduce Byzantine workers in the first place.
> > >
> > > We believe that accessing the average of honest gradients does not negate the need for Byzantine attacks.
> > > The goal of the latter is to hurt the accuracy of the model during training. Thus, only accessing the honest average gradient, without utilizing it, does not fulfill this goal.
> > > It may violate privacy and enable data reconstruction, which can make Byzantine attacks more powerful, but does not directly impact training accuracy.

---

### Official Review · Reviewer_y6Q8 · 2023-11-06

**Soundness:** 3 good
**Presentation:** 4 excellent
**Contribution:** 3 good
**Rating:** 3
**Confidence:** 4

**Summary:**

This paper studies Byzantine attacks in distributed learning. The authors first restrict the search space of Byzantine attacks. Based on the restriction, they propose an omniscient Byzantine attack called JUMP. In particular, JUMP solves a non-convex optimization problem that aims to maximize the global losses in several following epochs. Extensive experiments validate the performance of the proposed JUMP.

**Strengths:**

- The idea of performing stress testing on Byzantine defenses is interesting.
- The evaluation is through. The authors compare their method against different baselines. The ablations are helpful.
- The writing is clear and easy to follow.

**Weaknesses:**

- The proposed JUMP attack assumes that Byzantine clients have access to all data of honest clients. This assumption is almost impossible to hold in real-world applications. I understand that the authors try to explore the worst-case behavior of Byzantine attacks. Given the unpractical assumption of the proposed attack, I still doubt whether these worst-case results make any sense.
- JUMP needs to repeatedly compute the *gradients* of all honest clients, which is computationally expensive. The computation complexity also increases *linearly* with the number of honest clients and segment length.
- Based on the two aforementioned reasons, though the proposed JUMP attack demonstrates high attack effectiveness, I doubt whether the improvement is meaningful given the unrealistic assumptions and restrictions: have access to data of all honest clients, have knowledge of Byzantine defenses, and high computation cost.

**Questions:**

Please refer to the weaknesses.

---

> ### Author Response · Authors · 2023-11-16
>
> We thank you for your encouraging comments and address your concerns below. We hope that you re-evaluate our paper in this light, and we are happy to answer any further questions.
>
> >The proposed JUMP attack assumes that Byzantine clients have access to all data of honest clients. This assumption is almost impossible to hold in real-world applications. I understand that the authors try to explore the worst-case behavior of Byzantine attacks. Given the unpractical assumption of the proposed attack, I still doubt whether these worst-case results make any sense.
>
> We acknowledge that our attack is open-box and has full knowledge of the system and defense. This is currently stated clearly in sections 1.1 and 2.
> The purpose of our paper is not to propose a realistic or black-box attack. Rather, we (i) formulate and study the strategy followed by the optimal attack (i.e., solving (P)) in sections 2 and 4, respectively, and (ii) propose a tractable attack which retains most of the strength of the worst-case attack by building on the same strategy, as explained in Section 4.
> We recall that the aforementioned strategy consists in overshooting minima by amplifying the direction of the average honest gradient, at the right moment of training by being aware of the loss landscape.
>
> Furthermore, based on our result, we provide researchers with a tool to stress test the strength of their defenses in the worst case, using one attack, since it is theoretically claimed by Byzantine-robust defenses (Allouah et al. 2023, Karimireddy et al. 2021, 2022, Farhadkhani et al. 2022) to be able to withstand any attack.
> For example, even with the full knowledge of our attack, the SOTA defenses all perform well under low data heterogeneity, which actually confirms existing theory in the i.i.d. setting (Farhadkhani et al. 2022).
>
> Besides, our attack does not need to know all the data of honest clients; it only needs to know the average of honest gradients (see Algorithm 1) and some proxy data of the same (global) distribution. Although this is not a concern in this paper, but in practice it can be done by eavesdropping and data reconstruction attacks (e.g., gradient inversion).
> We will emphasize this point further in the paper.
>
> >JUMP needs to repeatedly compute the gradients of all honest clients, which is computationally expensive. The computation complexity also increases linearly with the number of honest clients and segment length.
>
> The greedy version of Jump ($\tau=1$, which is the one we recommend) does not need to recompute honest gradients. It only needs to know honest gradients' average of current round, which is common in existing attacks.
>
> We provide an analysis in Section A of the computational cost and also its trade-off with the power of the attack for various segment lengths.
> Notably, the greedy version of Jump $(\tau=1)$ has a reasonable runtime, especially when considering that future benchmarks will only need to run Jump instead of several other attacks.
> We could not include this discussion in the main body due to lack of space, and because we admittedly focus on a worst-case unbounded adversary.

---

> ### Comment · Reviewer_y6Q8 · 2023-11-22
> **Thanks for the responses**
>
> Thank you for the detailed responses.
>
> As the author claimed in the response, the recommended greedy version of JUMP attack is effective and does not violate any privacy principle.
>
> However, in this case, the proposed JUMP attack becomes an omniscient version of IPM attack proposed in [1]. And it is well-known that the omniscient version of different attacks can greatly improve their effectiveness [2]. Therefore, the novelty is quite limited.
>
> Therefore, I still maintain my rating.
>
> [1] Xie, Cong, Oluwasanmi Koyejo, and Indranil Gupta. "Fall of empires: Breaking byzantine-tolerant sgd by inner product manipulation." Uncertainty in Artificial Intelligence. PMLR, 2020.
>
> [2] Shejwalkar, Virat, and Amir Houmansadr. "Manipulating the byzantine: Optimizing model poisoning attacks and defenses for federated learning." NDSS. 2021.

---

> > ### Author Response · Authors · 2023-11-22
> >
> > Thank you for your response. We argue below that our work is novel and delivers important contributions and insights.
> >
> > > However, in this case, the proposed JUMP attack becomes an omniscient version of IPM attack proposed in [1].
> >
> > Our attack is not simply an omniscient version of IPM; the design is different, and we give insights on why our attack works better. We show in Section 4.2, among other factors, that allowing the attack vector to be in the $\textit{same direction}$ as the true gradient is essential to Jump, and is a key reason to its superiority over IPM. Also, directly maximizing the honest loss grants adversaries the ability to circumvent at the right moment during training. Finally, Jump enables adversaries to plan ahead when
> >  and maximize the training loss across iterations, which is not possible with IPM and other attacks.
> >
> > > And it is well-known that the omniscient version of different attacks can greatly improve their effectiveness [2].
> >
> > While we agree with the general statement above, for any attack and scenario, we believe that it does not negate the need to study worst-case attacks in the specific problem we consider. Our attack Jump improves over existing attacks, including [2], and we deliver insights on why Jump is superior, i.e., the overshooting strategy inherited from the optimal attack (see Section 4). In this sense, our work is novel both because we propose a new and stronger attack (which will benefit research on defenses as well), and also explain how it works in the same way as the optimal attack.

---

### Official Review · Reviewer_DW8P · 2023-11-07

**Soundness:** 2 fair
**Presentation:** 2 fair
**Contribution:** 2 fair
**Rating:** 5
**Confidence:** 3

**Summary:**

This paper presents an attack in the distributed setting, in which different workers provide the gradients of their data to the server that aggregates them to train the model.

The main question of the paper is to study how robust the aggerated model is to so-called Byzantine attackers who might send arbitrary values instated of their true gradients.

The paper focuses on a special type of attack, which has 3 properties (called simplifications) in how it attacks the protocol:
1. they limit the byzantine attackers to share the same vector of gradients.
2. they limit the adversary's vector to be co-linear with (i.e., to be a multiplicative factor of) the average honest gradient.
3. finally, they break the number of rounds T of the protocol into intervals of length $\tau$, and let the adversary optimize (i.e., maximize) the average loss (on the non-corrupted parties) for each interval.

With the above simplifications (that make the attack feasible to optimize) the paper is able to launch attacks and experiments with its power in comparison with certain other attacks and show that their attack does better in maximizing the loss.

**Strengths:**

The paper studies an important question.

The unification of *some* of the previous attacks as done in Section 4.2 is interesting. Though this framework is clearly not capturing all attacks on distributed learners.

**Weaknesses:**

The main weakness of the paper is that its attacks is not really "adaptive". Namely, with the assumptions made about the attack's performance (to make it feasible) it is rather easy to detect the Byzantine workers. (e.g., Property 1 and 2 make the adversary's shared values detectable).

I think at this stage the standard of how attacks/defenses are studied in robust learning is higher than the early years, and (rather trivially) being not adaptive is a major weakness.

You claim that your attack breaks current defenses, but certain defenses in the distributed setting come with *proofs* (e.g., those based on robust aggregation that use robust statistics and or bagging that even comes with certification). So, how can you "break" a provable defense?

**Questions:**

Regarding the 2nd point in the "Strengths" section above: does your formulation of previous attacks in Section 4.2 really capture all previous attacks? It seems not to be the case, e.g., attack of https://proceedings.mlr.press/v97/mahloujifar19a.html
Please clarify.

---

> ### Author Response · Authors · 2023-11-16
>
> We thank you for your encouraging comments and address your concerns below. We hope that you re-evaluate our paper in this light, and we are happy to answer any further questions.
>
> >You claim that your attack breaks current defenses, but certain defenses in the distributed setting come with proofs (e.g., those based on robust aggregation that use robust statistics and or bagging that even comes with certification). So, how can you "break" a provable defense?
>
> This is a very good question.
> The state-of-the-art convergence theory (Allouah et al. 2023, Karimireddy et al. 2022) under data heterogeneity only shows convergence to a neighborhood of a stationary point/minimum, where the radius of the neighborhood is proportional to a bound on gradient heterogeneity.
> Typically, these bounds may be extremely loose, and there is a considerable gap between the theoretical claims and empirical performance.
> On the other hand, in the homogeneous setting where the convergence neighborhood size can be made arbitrarily small (Farhadkhani et al. 2022), even our strong attack does not significantly hinder SOTA defenses, which in turn confirms existing theory.
>
> >The unification of some of the previous attacks as done in Section 4.2 is interesting. Though this framework is clearly not capturing all attacks on distributed learners.
>
> We believe that there is a misunderstanding: we do not claim to unify existing attacks.  In Section 4.2, we simply describe existing attacks in a generic way. It is true however that \textsc{Jump} captures several attacks as special cases, e.g., FOE and SF, but this is only a byproduct.
> We will make this clear in the paper.
>
> >The main weakness of the paper is that its attacks is not really "adaptive". Namely, with the assumptions made about the attack's performance (to make it feasible) it is rather easy to detect the Byzantine workers. (e.g., Property 1 and 2 make the adversary's shared values detectable).
>
> First, we agree that Byzantine gradients could be detected by having the aggregation eliminate identical inputs. Please note that it is standard in existing attacks and their implementation to have the Byzantine gradients be identical (Allouah et al. 2023, Karimireddy et al. 2021, 2022, Farhadkhani et al. 2022), since it is generally understood that adding small noise can make the malicious gradients nonidentical and still remain in a bad region of the space.
> However, such a defense (eliminating dense clusters) does not have theoretical guarantees and can easily be fooled in the presence of data heterogeneity, e.g., Byzantine gradients can gather around points they want to remove, which would be devastating if there is considerable data heterogeneity.
>
> Second, we acknowledge that strong attacks ought to be adaptive, as is Jump and other mentioned attacks.
> This is not the only advantage of our attack. As we explain in Section 4, among other factors, the main advantage of Jump is its ability to overshoot minima. This requires planning, which is arguably a stronger form of adaptivity, and is enabled by collinearity and also the maximization of the loss across time within a certain segment length.
>
> ### Clarifying remarks
> We would like to clarify that the simplifications made to obtain Jump are not properties. We essentially go from a natural Byzantine attack formulation in (P) to a simpler optimization formulation. In Section 4, we show that each simplification only marginally reduces the strength of the ``strongest" attack obtained by solving (P) directly.

---

> > ### Comment · Reviewer_DW8P · 2023-11-22
> > **Ack**
> >
> > Thanks for the responses. I keep my score, and still encourage the authors to look into the adaptivity nature of their attack. Ideally, one wants to argue that an attack is "universal" and can be applied to every scheme, to understand the limitations of what is possible. (E.g., check the paper "Universal Multi-Party Poisoning Attacks", for an example). Simply saying that adding noise makes things indistinguishable is not formal enough.

---

> > > ### Author Response · Authors · 2023-11-22
> > >
> > > We thank the reviewer for their response. We recall that we do not claim "universality" nor subsuming existing attacks, although the mentioned work seems interesting. Rather, we develop a $\textbf{stress test}$; that is, a worst-case attack that validates theoretical Byzantine robustness claims. Naturally, it is only interesting to test our attack against defenses that come with theoretical guarantees and that do not fail in trivial scenarios, which is why we do not consider the defense based on eliminating redundant inputs (as we explain in our initial response).

---

> > > > ### Comment · Reviewer_DW8P · 2023-11-23
> > > > **Re:**
> > > >
> > > > When you say "worst-case attack", it naturally means "adaptive" and hence "universal". These are very close notions. In other words, if your attack cannot be guaranteed to work against every scheme, then it is not "worst case" (and not universal / fully adpative either).

---

### Official Review · Reviewer_p8Sy · 2023-11-11

**Soundness:** 3 good
**Presentation:** 3 good
**Contribution:** 3 good
**Rating:** 5
**Confidence:** 5

**Summary:**

This paper considered the problem of designing Byzantine attack schemes for distributed learning systems, where the goal is to manipulate the workers' feedback to the server to maximize the training loss function. The authors proposed a new adaptive attack scheme (i.e., the attacker has full knowledge of the model and aggregation method in the distributed learning system) called Jump. The main idea of Jump is to restrict the solution space of the Byzantine attack problem to a rank-1 solution that is colinear with the average gradient of the honest workers. The authors empirically showed the effectiveness of the Jump method on CNN with MNIST and CIFAR-10 datasets.

**Strengths:**

1. The authors proposed a new adaptive Byzantine attack scheme by simplifying the original Byzantine attack problem.

2. The authors demonstrated that it outperforms several well-known baseline methods.

**Weaknesses:**

1. The Byzantine attack and defense problem in distributed learning systems is a well-studied problem. This paper's novelty and contributions to this area are limited.

2. Some key references in this area are missing.

3. Experimental studies are inadequate.

Please see the detailed comments and questions below.

**Questions:**

1. The Byzantine attack and defense problem in distributed learning has been studied extensively in recent years and there are many variants of this problem (e.g., by allowing the server to have an auxiliary dataset to boost trust [R1]). This paper remains focused on the most basic and standard problem setting, hence the novelty of this paper is limited in this sense.

[R1] X. Cao, M. Fang, J. Liu, and N. Gong, FLTrust: Byzantine-robust Federated Learning via Trust Bootstrapping, in Proc. NDSS 2021.

2. This paper simplifies the problem by restricting the solution space to rank-1 vectors that are co-linear with the average of honest workers. Although this idea is interesting, this paper missed many ideas and methods developed in recent years, e.g., by treating the original attack problem as a bilevel optimization problem and employing more sophisticated bilevel optimization algorithms. It remains unclear how well the simplified strategy in this paper will perform when compared to these methods. Also, how about other simplified and restricted solution spaces, e.g., rank-2, rank-3, etc.?

3. Although the authors conducted extensive numerical studies, most of the experiments were conducted on CNN with MNIST and CIFAR-10, which is now considered relatively easy datasets. Also, the authors only compared MinMax and MinSum in the adaptive attack category. The authors should conduct more experiments with other adaptive attack schemes.

4. Also related to the previous bullet, the solution quality of the Jump method clearly depends on the nonlinear optimization solver. The authors only used Powell's method. This paper can benefit from testing and comparing more nonlinear solvers, and analyzing the impact of different nonlinear solvers on the proposed Jump method.

---

> ### Author Response · Authors · 2023-11-16
>
> We thank you for your encouraging comments and address your concerns below. We hope that you re-evaluate our paper in this light, and we are happy to answer any further questions.
>
> >This paper remains focused on the most basic and standard problem setting, hence the novelty of this paper is limited in this sense.
>
> We acknowledge that the Byzantine attack and defense problem in distributed learning systems is a well-studied problem. However, given the lack of works investigating worst-case Byzantine attacks in the standard Byzantine ML problem, we believe that our contribution is valuable to the community without the need to explore other variants of the problem.
> We will add the reference mentioned by the reviewer in our paper.
>
> >This paper simplifies the problem by restricting the solution space to rank-1 vectors that are co-linear with the average of honest workers. Although this idea is interesting, this paper missed many ideas and methods developed in recent years, e.g., by treating the original attack problem as a bilevel optimization problem and employing more sophisticated bilevel optimization algorithms. It remains unclear how well the simplified strategy in this paper will perform when compared to these methods. Also, how about other simplified and restricted solution spaces, e.g., rank-2, rank-3, etc.?
>
> We recall that the approach leading to the simplified optimization problem of Jump is principled and does not necessitate exploring, e.g., rank-two solution spaces. Indeed, in Section 4, we show that the successive simplifications do not significantly degrade the power of the attack.
>
> >The solution quality of the Jump method clearly depends on the nonlinear optimization solver. The authors only used Powell's method. This paper can benefit from testing and comparing more nonlinear solvers, and analyzing the impact of different nonlinear solvers on the proposed Jump method.
>
> In Section 3.2, we discuss appropriate nonlinear solvers for our tasks. We recall  that the constraints of Problem $(P_\ell)$ are in general non-differentiable due to robust aggregation. This motivates the use of derivative-free non-linear solvers such as Powell’s (Powell, 1964), Nelder-Mead’s (Nelder \& Mead, 1965), and Differential Evolution (Storn \& Price, 1997). In fact, we have tested all the aforementioned methods on MNIST and our two-dimensional tasks. Our experiments have shown that Powell's method solver is most effective while Differential Evolution can obtain better solutions but is much slower than Powell's, so we use Powell's in our main experiments. We will add a detailed description in the paper.
>
> >Although the authors conducted extensive numerical studies, most of the experiments were conducted on CNN with MNIST and CIFAR-10, which is now considered relatively easy datasets.
>
> We use heterogeneous variants of MNIST and CIFAR-10, where the extreme heterogeneity scenario is challenging and is common in Byzantine-robustness works (Allouah et al. 2023, Karimireddy et al. 2022). In any case, note that the more challenging the task, the harder it is for the defender and not the attacker, which is why we focus on moderate heterogeneity, or low fraction of Byzantine workers.

---

### Meta-Review · Area_Chair_7EsW · 2023-12-01

**Metareview:**

This paper provides a novel algorithm JUMP to address the problem relating to Byzantine attacks. Despite the important problem setting in distributed learning and potentially interesting observations, several concerns raised by the reviewers were not yet resolved; as a consequence, the promising submission is not ready to be accepted in its current form.

**Justification For Why Not Higher Score:**

The reviewer's concerns are not cleared and consensus is reached among reviewers.

**Justification For Why Not Lower Score:**

N/A

---

### Decision · Program_Chairs · 2024-01-16

Reject